# On Robustness of Principal Component Regression

## Abstract

Consider the setting of Linear Regression where the observed response variables, in expectation, are linear functions of the $p$-dimensional covariates. Then to achieve vanishing prediction error, the number of required samples scales faster than $p\sigma^2$, where $\sigma^2$ is a bound on the noise variance. In a high-dimensional setting where $p$ is large but the covariates admit a low-dimensional representation (say $r \ll p$), then Principal Component Regression (PCR), cf. [36], is an effective approach; here, the response variables are regressed with respect to the principal components of the covariates. The resulting number of required samples to achieve vanishing prediction error now scales faster than $r\sigma^2 (\ll p\sigma^2)$. Despite the tremendous utility of PCR, its ability to handle settings with noisy, missing, and mixed (discrete and continuous) valued covariates is not understood and remains an important open challenge, cf. [24]. As the main contribution of this work, we address this challenge by rigorously establishing that PCR is robust to noisy, sparse, and possibly mixed valued covariates. Specifically, under PCR, vanishing prediction error is achieved with the number of samples scaling as $r \max(\sigma^2, \rho^{-4} \log^5(p))$, where $\rho$ denotes the fraction of observed (noisy) covariates. We establish generalization error bounds on the performance of PCR, which provides a systematic approach in selecting the correct number of components $r$ in a data-driven manner. The key to our result is a simple, but powerful equivalence between (i) PCR and (ii) Linear Regression with covariate pre-processing via Hard Singular Value Thresholding (HSVT). From a technical standpoint, this work advances the state-of-the-art analysis for HSVT by establishing stronger guarantees with respect to the $\|\cdot\|_{2,\infty}$-error for the estimated matrix rather than the Frobenius norm/mean-squared error (MSE) as is commonly done in the matrix estimation / completion literature.

## 1 Introduction

### 1.1 Background

In this paper, we are interested in developing a better understanding of a popular prediction method known as Principal Component Regression (PCR). In a typical prediction problem setup, we are given access to a labeled dataset $\{(Y_i, \boldsymbol{A}_{i,\cdot})\}$ over $i \geq 1$; here, $Y_i \in \mathbb{R}$ represents the response variable (also known as the label or target) we wish to predict and $\boldsymbol{A}_{i,\cdot} \in \mathbb{R}^{1 \times p}$ represents the associated covariate (or feature) to be utilized in the prediction process.

**Linear Regression.** In Linear Regression, the data is believed to be generated as per a latent linear model and the goal is to learn the linear predictor. More precisely, for some $\beta^* \in \mathbb{R}^p$ and each $i \geq 1$, $Y_i = \boldsymbol{A}_{i,\cdot}\beta^* + \epsilon_i$, where $\epsilon_i$ denotes independent, zero-mean idiosyncratic noise with variance bounded by $\sigma^2$. Under generic noise distributions, the Ordinary Least Squares (OLS) estimator learnt using such observations yields an in-sample (or training) prediction error that vanishes to zero as long as the number of observations scales faster than $p\sigma^2$ (e.g., see [53] and references therein). The same result holds true for the generalization prediction error under reasonable restrictions on the model class (e.g., see [53] and references therein).

**Principal Component Regression.** In the high-dimensional setting, the required number of samples may be too great since it scales with the number of features $p$, which is large. However, this problem

can be circumvented when the covariates have a latent, low-dimensional representation. In particular, PCR, cf. [36], has been precisely designed to address such a setting. Using all observed covariates, PCR first finds an $r \ll p$ dimensional representation for each feature using the method of Principal Component Analysis (PCA); specifically, PCA projects every covariate $A_{i,\cdot}$ onto the subspace spanned by the top $r$ right singular vectors of the covariate matrix, the concatenation of all observed covariates. PCR then uses the $r$-dimensional features to perform linear regression. If the covariate matrix is indeed of rank $r$, then by the theory of Linear Regression, it follows that the number of samples required to achieve vanishing in- and out-of-sample prediction error scales faster than $r\sigma^2$, which is significantly smaller when $r \ll p$.

**Noisy, missing, and mixed valued covariates.** In many practical scenarios of interest (including high-dimensional settings where $p$ is large), the covariates are not fully observed. Specifically, a common thread of many modern datasets is that only a small fraction of the covariates are observed, and the observations themselves are noisy versions of the true covariates. Moreover, as is standard in most real-world datasets, the observations may also be mixed (discrete and continuous) valued covariates. Despite the tremendous success of PCR in a variety of applications, its ability to handle such scenarios remains unknown, as noted in a recent survey [24].

In the context of Linear Regression, this scenario fits under the error-in-variable regression framework, where there has been exciting recent advancement, particularly in the high dimensional setting (see Section 1.3 for details). However, the current inventory of methods fall short in addressing the key challenge of handling noisy, sparse, and mixed valued covariates as the proposed estimators require detailed knowledge of the underlying noise model of the covariates.

## 1.2 Contributions

**Summary of results.** As the main contribution of this work, we argue that PCR, without any change, is robust to noise and missing values in the observed covariates. In particular, we demonstrate that PCR does not require *any* knowledge about the underlying noise model that corrupts the covariates in order to generate predicted responses. Formally, we argue that the (training) error decays to zero as long as the number of samples scales faster than $r \max(\sigma^2, \rho^{-4} \log^5(p))$, where $\rho$ denotes the fraction of observed (noisy) covariates. For a precise statement of the result, please see Theorem 4.1.

We also define an appropriate notion of generalization error for the particular setting of PCR. With respect to this notion, we establish that the testing prediction error of PCR scales similarly to that of the training error, i.e., the testing (or generalization) prediction error is bounded above by the training prediction error plus a term that scales as $r^{3/2}/\sqrt{n}$, where $n$ denotes the number of training samples; hence, the testing prediction error vanishes as long as the number of observations scales faster than $r \max(\sigma^2, \rho^{-4} \log^5(p))$. For a precise statement of the result, see Theorem 4.2.

We extend our results for PCR's in- and out-of-sample prediction error even when the ground-truth covariates are not low-rank and the linear model itself may be misspecified (see Theorem 5.1 and Corollary 5.1). This result further suggests the robustness of PCR, reinforcing the utility of applying it in practice. Further, our result on generalization error provides a systematic way to select the correct number of principal components in a data-driven manner, i.e., to choose the value of $r$ that minimizes the training error plus the generalization penalty term $r^{3/2}/\sqrt{n}$.

Finally, we describe various applications of our results, including synthetic control, time series analysis, privacy preserving regression, and regression with mixed valued covariates. Please refer to Section 6 for details.

**Overview of techniques.** To prove our results, we establish a simple, but powerful equivalence between (i) PCR and (ii) Linear Regression with covariate pre-processing via Hard Singular Value Thresholding (HSVT) (see Proposition 3.1). The HSVT algorithm is commonly analyzed in literature, cf. [25], for matrix estimation / completion. In fact, there is significant literature establishing that HSVT is a noise model agnostic method that recovers the ground-truth matrix from its noisy observations. However, the current results concerning HSVT establish its estimation accuracy in terms of the mean-squared error or expected squared Frobenius norm of the error matrix. To establish our above mentioned results, we bound the expected squared $\|\cdot\|_{2,\infty}$ of the error matrix (see Lemma 5.1), which is a stronger guarantee than what is known in literature (note $\frac{1}{np}\|\boldsymbol{E}\|_F^2 = \frac{1}{np}\sum_{i=1}^n \sum_{j=1}^p e_{ij}^2 \leq \frac{1}{n}\max_{j\in[p]}\sum_{i=1}^n e_{ij}^2 = \frac{1}{n}\|\boldsymbol{E}\|_{2,\infty}^2$). Thus, this result for HSVT may be of interest in its own right.

Our generalization error result utilizes the standard framework of Rademacher complexity, cf. [16] and references therein. However, there are two crucial differences that we need to overcome in order to obtain sharp, meaningful bounds. First, the notion of generalization we utilize to analyze PCR is slightly different from the traditional setup as the noisy test covariates (but not responses) are included in the training process, which complicates the analysis (see Section 2.3 for details); we relate this modified notion of generalization to that of the classical notion. Second, to obtain sharp bounds, we argue that the Rademacher complexity under PCR scales with the dimensionality of the number of principle components utilized rather than the ambient dimension $p$. To achieve this bound, we identify the Rademacher complexity of PCR with implicit $\ell_0$-regularization.

## 1.3 Related works

We primarily focus on the literature pertaining to error-in-variable regression and PCR, but also include a brief discussion on the literature for matrix estimation / completion in Appendix A.

**Error-in-variable regression.** There exists a rich body of work regarding high-dimensional error-in-variable regression (cf. [41], [29], [46], [47], [17], [18], [26], [27], [37]). Two common threads of these works include: (1) a sparsity assumption on $\beta^*$; (2) error bounds with convergence rates for estimating $\beta^*$ under different norms, i.e., $\|\widehat{\beta} - \beta^*\|_q$ where $\|\cdot\|_q$ denotes the $\ell_q$-norm. In all of these works, the goal is to recover the underlying model, $\beta^*$. In contrast, as discussed, the goal of PCR is to primarily provide good prediction. Some notable works closest to our setup include [41], [29], [47], which are described in some more detail next.

In [41], a non-convex $\ell_1$-penalization algorithm is proposed based on the plug-in principle to handle covariate measurement errors. This approach requires explicit knowledge of the unobserved noise covariance matrix $\Sigma_H = \mathbb{E} H^T H$ and the estimator designed *changes* based on their assumption of $\Sigma_H$. They also require explicit knowledge of a bound on $\|\beta^*\|_2$, the object they aim to estimate. In contrast, PCR does not require any such knowledge about the distribution of the noise matrix $H$ (i.e., the algorithm does not explicitly use this information to make predictions).

[29] builds upon [41], but propose a convex formulation of Lasso. Although the algorithm introduced does not require knowledge of $\|\beta^*\|_2$, similar assumptions on $Z$ and $H$ (e.g., sub-gaussianity and access to $\Sigma_H$) are made. This renders their algorithm to be not model agnostic. In fact, many works (e.g., [46], [47], [17]) require either $\Sigma_H$ to be known or the structure of $H$ is such that it admits a data-driven estimator for its covariance matrix. This is so because these algorithms rely on correcting the bias for the matrix $Z^T Z$, which PCR does not need to compute.

It is worth noting that all these works in error-in-variable regression focus only on learning $\beta^*$, and not explicitly de-noising the noisy covariates. Thus even with the knowledge of $\beta^*$, it is not clear how to use it for producing predictions of response variable when given noisy covariates.

**Principal Component Regression.** The formal literature providing an analysis of PCR is surprisingly sparse, especially given its ubiquity in practice. A notable work is that of [15], which suggests a variation of PCR to infer the direction of the principal components. However, it stops short of providing meaningful finite sample analysis beyond what is naturally implied by that of standard Linear Regression. The regularization property of PCR is also well known due to its ability to reduce the variance. As a contribution, we provide rigorous finite sample guarantees of PCR: (i) under noisy, missing covariates; (ii) when the linear model is misspecified; (iii) when the covariate matrix is not exactly low-rank (see Theorem 5.1 and Corollary 5.1).

As a further contribution, we argue that the resulting regression model from PCR has sparse support (this is established using the equivalence between PCR and Linear Regression with covariate pre-processing via HSVT); this sparsity allows for improved generalization error as the Rademacher complexity of the resulting model class scales with this sparsity parameter (i.e., the rank of the covariate matrix pre-processed with HSVT). Hence, PCR not only addresses the challenge of noisy and missing covariates, but also, in effect, performs multiple implicit regularization.

## 2 Problem Setup

The standard formulation for regression considers the setting where the covariates are noiseless and fully observed. In this work, our interest is in a more realistic setting where we observe a noisy and sparse version of the covariates. In particular, our interest is in the high-dimensional framework where the number of observations may be far fewer than the ambient dimension of the covariates.

## 2.1 Model

We describe the model in terms of the structural assumptions on the covariates and the generative process for the response variables. Let $N \geq 1$ denote the total number of observations of interest.

**Covariates.** Let $\boldsymbol{A} \in \mathbb{R}^{N \times p}$ denote the matrix of true covariates, where the number of predictors $p$ is assumed to exceed $N$, i.e., $N \leq p$. We assume the entries of $\boldsymbol{A}$ are bounded:

**Property 2.1** *There exists an absolute constant $\Gamma \geq 0$ such that $|A_{ij}| \leq \Gamma$ for all $(i, j) \in [N] \times [p]$.*

Additionally, we shall assume that the covariates have a lower-dimensional representation, which is formalized as follows:

**Property 2.2** *The covariates matrix $\boldsymbol{A} \in \mathbb{R}^{N \times p}$ has rank $r < N \leq p$.*

**Response Variables.** For each $i \in [N]$, we let $Y_i$ denote the random response variable that is linearly associated with the covariate $\boldsymbol{A}_{i,\cdot} \in \mathbb{R}^{1 \times p}$, i.e.,

$$Y_i = \boldsymbol{A}_{i,\cdot}\beta^* + \epsilon_i \tag{1}$$

where $\beta^* \in \mathbb{R}^p$ is the unknown model parameter and $\epsilon_i \in \mathbb{R}$ denotes the noise.

**Property 2.3** *The response noise $\epsilon = [\epsilon_i] \in \mathbb{R}^N$ is a random vector with independent, mean zero entries such that each of its components has variance bounded above by $\sigma^2$.*

## 2.2 Observations

Rather than observing $\boldsymbol{A}$, we are given access to its corrupted version $\boldsymbol{Z}$, which contains noisy and missing values. Additionally, the observed response variables are restricted to a subset of the $N$ observations.

**Noisy covariates with missing values.** We observe $\boldsymbol{Z} \in \mathbb{R}^{N \times p}$, which is assumed to satisfy the following property.

**Property 2.4** *For all $(i, j) \in [N] \times [p]$, the $(i, j)$-th entry of $\boldsymbol{Z}$, denoted as $Z_{ij}$, is defined as $A_{ij} + \eta_{ij}$ with probability $\rho$ and $\star$ with probability $1 - \rho$, for some $\rho \in (0, 1]$; here, $\star$ denotes a missing value and $\eta_{ij}$ denotes the noise in the $(i, j)$-th entry.*

In words, Property 2.4 states that each entry $Z_{ij}$ is observed with probability $\rho$, independently of other entries; however, even under observation, $Z_{ij}$ is a noisy instantiation of the true covariate $A_{ij}$.

Let $\boldsymbol{H} = [\eta_{ij}] \in \mathbb{R}^{N \times p}$ denote the covariate noise matrix. For ease of notation, let us define $\boldsymbol{X} = \boldsymbol{A} + \boldsymbol{H}$ as the noisy perturbation of covariate matrix, without missing values. We assume the following property about the noise matrix $\boldsymbol{H}$ (see Appendix C for the definition of the following $\psi_\alpha$-random variables/vectors).

**Property 2.5** *Let $\boldsymbol{H}$ be a matrix of independent, mean zero $\psi_\alpha$ -rows for some $\alpha \geq 1$, i.e., there exists an $\alpha \geq 1$ and $K_\alpha < \infty$ such that $\|\eta_{i,\cdot}\|_{\psi_\alpha} \leq K_\alpha$ for all $i \in [N]$. As a consequence, there exists a $\gamma^2 > 0$ such that $\|\mathbb{E}\eta_{i,\cdot}^T \eta_{i,\cdot}\| \leq \gamma^2$ for all $i \in [N]$.*

**Response Variables.** Let $\Omega \subset [N]$ with $|\Omega| = n < N$. We observe $Y_i$, where $i \in \Omega$.

## 2.3 Objective

Given noisy observations of all $N$ covariates $\{\boldsymbol{Z}_{1,\cdot}, \ldots, \boldsymbol{Z}_{N,\cdot}\}$ and a subset of response variables $\{Y_i : i \in \Omega\}$, our aim is to produce an estimate $\widehat{Y} \in \mathbb{R}^N$ so that the prediction error is minimized. Specifically, we measure performance in terms of the *training error*, $\text{MSE}_\Omega(\widehat{Y}) = (1/n) \cdot \mathbb{E}[\sum_{i \in \Omega}(\widehat{Y}_i - \boldsymbol{A}_{i,\cdot}\beta^*)^2]]$ and *testing error*, i.e., $\text{MSE}(\widehat{Y}) = (1/N) \cdot \mathbb{E}[\sum_{i=1}^N(\widehat{Y}_i - \boldsymbol{A}_{i,\cdot}\beta^*)^2]$.

It is worth remarking that in our definition of test performance, the algorithm is given access to the observations associated with the covariates for both *training* and *testing* data during the training procedure; of course, however, the algorithm does not access the test response variables! Traditionally, the algorithm only has access to the training covariates and response variables during the training process. The reason for this difference is a simple consequence of the nature of the algorithm of interest, PCR. Specifically, PCR pre-processes the covariates using PCA, which changes the training procedure if only a subset of the covariates are utilized. Therefore, to allow for a meaningful evaluation, it is a natural to allow the algorithm to have access to *all* available covariate information. Indeed, as discussed in Section 6, having access to all covariates is entirely reasonable in many important real-world applications.

## 2.4 Notations, definitions, and a summary of assumptions.

For any matrix $\boldsymbol{B} \in \mathbb{R}^{N \times p}$ and index set $\Omega \subset [N]$, let $\boldsymbol{B}^\Omega$ denote the $|\Omega| \times p$ submatrix of $\boldsymbol{B}$ formed by stacking the rows of $\boldsymbol{B}$ according to $\Omega$, i.e., $\boldsymbol{B}^\Omega$ is the concatenation of $\{\boldsymbol{B}_{i,\cdot} : i \in \Omega\}$. The superscript $\Omega$ is sometimes omitted if the matrix representation is clear from context. $\text{poly}(\alpha_1, \ldots, \alpha_k)$, denotes a function that scales at most polynomially in its arguments. Let $x \vee y = \max(x, y)$ and $x \wedge y = \min(x, y)$ for any $x, y \in \mathbb{R}$. Lastly, let $\mathbb{1}$ denote the indicator function.

# 3 Algorithm

We recall the description of PCR, cf. [36]. In particular, we suggest a minor modification of PCR in the presence of missing data. Specifically, PCR is modified by simply rescaling the observed covariate matrix by the inverse of the fraction of observed data. We also describe Linear Regression with covariate pre-processing via Hard Singular Value Thresholding (HSVT). We observe that these two algorithms produce identical estimates of the response variable. This simple, but powerful equivalence will allow us to study the robustness property of PCR through the lens of HSVT.

## 3.1 Principal Component Regression

Let $\widehat{\rho}$ denote the fraction of observed entries of $\boldsymbol{Z}$, i.e., $\widehat{\rho} = \frac{1}{Np} \sum_{i=1}^N \sum_{j=1}^p \mathbb{1}(Z_{ij} \neq \star) \vee \frac{1}{Np}$. Let $\widetilde{\boldsymbol{Z}} \in \mathbb{R}^{N \times p}$ represent the rescaled version of $\boldsymbol{Z}$, where every unobserved value $\star$ is replaced by $0$, i.e., for all $i \in [N]$ and $j \in [p]$, $\widetilde{Z}_{ij} = Z_{ij}/\widehat{\rho}$ if $Z_{ij} \neq \star$ and $0$ otherwise.

The Singular Value Decomposition (SVD) of $\widetilde{\boldsymbol{Z}}$ is denoted as $\widetilde{\boldsymbol{Z}} = \boldsymbol{U}\boldsymbol{S}\boldsymbol{V}^T = \sum_{i=1}^N s_i u_i v_i^T$ where $\boldsymbol{U} \in \mathbb{R}^{N \times N}$, $\boldsymbol{S} \in \mathbb{R}^{N \times p}$, and $\boldsymbol{V} \in \mathbb{R}^{p \times p}$. Without loss of generality, assume that the singular values $s_i$'s are arranged in decreasing order, i.e., $s_1 \geq \ldots \geq s_N \geq 0$. Note that $\boldsymbol{U} = [u_1, \ldots, u_N]$ and $\boldsymbol{V} = [v_1, \ldots, v_p]$ are orthonormal matrices, i.e., the $u_i$'s and $v_j$'s are orthonormal vectors.

For any $k \in [N]$, let $\boldsymbol{U}_k = [u_1, \ldots, u_k]$, $\boldsymbol{V}_k = [v_1, \ldots, v_k]$, and $\boldsymbol{S}_k = \text{diag}(s_1, \ldots, s_k)$. Then, the $k$-dimensional representation of $\widetilde{\boldsymbol{Z}}$, as per PCA, is given by $\boldsymbol{Z}^{\text{PCR},k} = \widetilde{\boldsymbol{Z}}\boldsymbol{V}_k$. Let $\beta^{\text{PCR},k} \in \mathbb{R}^k$ be the solution to the Linear Regression problem under $\boldsymbol{Z}^{\text{PCR},k}$, i.e., $\beta^{\text{PCR},k}$ solves minimize $\sum_{i \in \Omega}(Y_i - \boldsymbol{Z}_{i\cdot}^{\text{PCR},k}w)^2$ over $w \in \mathbb{R}^k$. Then, the estimated $N$-dimensional response vector $\widehat{Y}^{\text{PCR},k} = \boldsymbol{Z}^{\text{PCR},k}\beta^{\text{PCR},k}$.

## 3.2 Linear Regression and Hard Singular Value Thresholding

Here, we describe Linear Regression with covariate pre-processing via Hard Singular Value Thresholding (HSVT). To that end, given any $\lambda > 0$, we define the map $\text{HSVT}_\lambda : \mathbb{R}^{N \times p} \to \mathbb{R}^{N \times p}$, which simply shaves off the input matrix's singular values that are below the threshold $\lambda$. Precisely, given $\boldsymbol{B} = \sum_{i=1}^N \sigma_i x_i y_i^T$, let $\text{HSVT}_\lambda(\boldsymbol{B}) = \sum_{i=1}^N \sigma_i \mathbb{1}(\sigma_i \geq \lambda)x_i y_i^T$.

For any $k \in [N]$, given $\widetilde{\boldsymbol{Z}}$ as before, define $\boldsymbol{Z}^{\text{HSVT},k} = \text{HSVT}_{s_k}(\widetilde{\boldsymbol{Z}})$. Let $\beta^{\text{HSVT},k} \in \mathbb{R}^p$ be a solution of Linear Regression under $\boldsymbol{Z}^{\text{HSVT},k}$, i.e., $\beta^{\text{HSVT},k}$ solves minimize $\sum_{i \in \Omega}(Y_i - \boldsymbol{Z}_{i\cdot}^{\text{HSVT},k}w)^2$ over $w \in \mathbb{R}^p$. Then, the estimated $N$-dimensional response vector $\widehat{Y}^{\text{HSVT},k} = \boldsymbol{Z}^{\text{HSVT},k}\beta^{\text{HSVT},k}$.

## 3.3 Equivalence

We now state a key relation between the above two algorithms. Precisely, the two algorithms produce identical estimated response vectors. Refer to Appendix E.1 for a proof of Proposition 3.1.

**Proposition 3.1** *For any $k \leq N$, $\widehat{Y}^{\text{PCR},k} = \widehat{Y}^{\text{HSVT},k}$.*

# 4 PCR Prediction Error: Low-Rank Covariates

We now state our main results in terms of the training and testing error for PCR. For a review on vector and matrix norms, see Appendix C.

## 4.1 Theorem Statements

**Training prediction error.** We state the following result for PCR when the covariate matrix is low-rank, i.e., $\boldsymbol{A}$ admits a low-dimensional representation, and PCR chooses the correct number of principal components. Refer to Appendix J for a proof of Theorem 4.1.

**Theorem 4.1 (Training Error of PCR)** *Let Properties 2.1, 2.2, 2.3, 2.4, and 2.5 hold with $r$ denotes the rank of $\boldsymbol{A}$. Suppose PCR chooses the correct number of principal components $k = r$. Let $\rho \geq \frac{64 \log(Np)}{Np}$ and $n = \Theta(N)$. Then for any given $\Omega \subset [N]$,*

$$\mathrm{MSE}_\Omega(\widehat{Y}) \leq \frac{4\sigma^2 r}{n} + C(\alpha) \frac{C' \log^2(np)}{n\rho^2} \|\beta^*\|_1^2 \cdot \left( r + \frac{(n^2\rho + np)\log^3(np)}{\rho^2 \tau_r^2} \right), \tag{2}$$

*where $C' = (1 + \gamma + \Gamma + K_\alpha)^4$, $\tau_r$ is $r$th singular value of true covariate matrix $\boldsymbol{A}$, and $C(\alpha) > 0$ a constant that may depend on $\alpha \geq 1$.*

**Test prediction error.** We describe the test error to evaluate the generalization performance of PCR. Here, the model class under consideration is described by $\mathcal{F} = \{\beta \in \mathbb{R}^p : \|\beta\|_2 \leq B, \|\beta\|_0 \leq r\}$, where $B > 0$ is a positive constant. As previously mentioned, the emphasis of this work is to provide a rigorous analysis on the prediction properties of the PCR algorithm through the lens of HSVT (due to the equivalence of their predictions, see Proposition 3.1).

To that end, we consider candidate vectors $\beta^{\mathrm{HSVT},r} = \boldsymbol{V}_r \cdot \beta^{\mathrm{PCR},r} \in \mathbb{R}^p$ that have bounded $\ell_2$-norm (as is commonly assumed in generalization error analysis for linear regression). Further, as argued in Proposition 3.1, PCR with parameter $r$ is equivalent to Linear Regression with pre-processing of the noisy covariates using HSVT (i.e., retaining the top $r$ singular values). In light of this observation, we establish the following result that suggests restricting our model class to sparse linear models only.

**Proposition 4.1** *Let $\boldsymbol{X} \in \mathbb{R}^{n \times p}$ and $M = \boldsymbol{X}v$ for some $v \in \mathbb{R}^p$. If $\mathrm{rank}(\boldsymbol{X}) = r$, then there exists a $v^* \in \mathbb{R}^p$ such that $M = \boldsymbol{X}v^*$ and $\|v^*\|_0 = r$.*

By Proposition 4.1, for any $\boldsymbol{Z}^{\mathrm{HSVT},r}$ and $\beta^{\mathrm{HSVT},r} = \boldsymbol{V}_r \cdot \beta^{\mathrm{PCR},r}$, there exists a $\beta' \in \mathbb{R}^p$ such that $\boldsymbol{Z}^{\mathrm{HSVT},r} \cdot \beta^{\mathrm{HSVT},r} = \boldsymbol{Z}^{\mathrm{HSVT},r} \cdot \beta'$, where $\|\beta'\|_0 \leq r$. Thus, for analyzing the generalization properties of PCR with parameter $r$, or equivalently Linear regression with covariate pre-processing using HSVT with rank $r$ thresholding, we can restrict our model class to linear predictors with sparsity $r$.

Hence, given the above observations, we consider the collection of candidate regression vectors within $\mathcal{F}$, i.e., those subset of vectors in $\mathbb{R}^p$ that have bounded $\ell_2$-norm and are $r$-sparse. Refer to Appendix K for a proof of Theorem 4.2 and a more rigorous theoretical justification of our model class of interest.

**Theorem 4.2 (Test Error of PCR)** *Let the conditions of Theorem 4.1 hold. Further, let $\widehat{\beta} \in \mathcal{F}$. Then, $\mathbb{E}_\Omega[\mathrm{MSE}(\widehat{Y})] \leq \mathbb{E}_\Omega[\mathrm{MSE}_\Omega(\widehat{Y})] + \frac{C' r^{3/2} \widehat{\alpha}^2}{\sqrt{n}} \|\beta^*\|_1$, where $C' = CB^2\Gamma$ with $C > 0$ a universal constant; $\widehat{\alpha}^2 = \mathbb{E}[\|\widehat{\boldsymbol{A}}\|_{\max}^2]$; and $\mathbb{E}_\Omega$ denotes the expectation taken with respect to $\Omega \subset [N]$ (of size $n$), which is chosen uniformly at random without replacement.*

Since Theorem 4.1 holds for any $\Omega$, we note that $\mathbb{E}_\Omega[\mathrm{MSE}_\Omega(\widehat{Y})]$ is also bounded above by the right-hand side of (2).

**Implications.** The statement of Theorem 4.1 requires that the *correct* number of principal components are chosen in PCR. In settings where all $r$ singular values of $\boldsymbol{A}$ are roughly equal (see the discussion below for such an example), i.e., $\tau_1 \approx \tau_2 \approx \ldots \approx \tau_r = \Theta(\sqrt{Np/r})$, the training prediction error vanishes as long as $n$ scales faster than $\max(\sigma^2 r, \rho^{-4} r \log^5 p)$. Further, as long as $r = O(\log^{\frac{1}{4}} p)$, the testing error also vanishes with the same scaling of $n$, with $n = \Theta(N)$.

## 4.2 Example

**Embedded random Gaussian features.** We now present a classical example that justifies algorithms such as PCR (or PCA). Consider the setting where the matrix of interest $\boldsymbol{A} \in \mathbb{R}^{N \times p}$ is generated by sampling its rows from a distribution on $\mathbb{R}^p$, which in turn, is an embedding of some underlying latent distribution on $\mathbb{R}^r$. Specifically, consider the example in Proposition 4.2, which describes how the rows of $\boldsymbol{A}$ are generated; this is similar in spirit to the probabilistic model for PCA, cf. [19, 49].

**Proposition 4.2** *Let $\boldsymbol{A} = \tilde{\boldsymbol{A}}\tilde{\boldsymbol{R}}$ where $\tilde{\boldsymbol{A}} \in \mathbb{R}^{N \times r}$ is a random matrix whose entries are independent standard normal random variables, i.e., $\tilde{A}_{ij} \sim \mathcal{N}(0,1)$ and $\tilde{\boldsymbol{R}} \in \mathbb{R}^{r \times p}$ is another random matrix with independent entries such that $\tilde{R}_{ij} = 1/\sqrt{r}$ with probability $1/2$ and $\tilde{R}_{ij} = -1/\sqrt{r}$ with probability $1/2$. Suppose, $r \leq \frac{\sqrt{p}}{4\sqrt{2}\log p} + 1$. Then, $\mathrm{MSE}_\Omega(\widehat{Y}) \leq \frac{r}{n}(4\sigma^2 + C''\|\beta^*\|_1^2 \frac{\log^7(np)}{\rho^4})$, where $C'' > 0$ is a constant that may depend on model parameters $\gamma, \alpha \geq 1$, and $K_\alpha$.*

# 5 PCR Prediction Error: Beyond Low-Rank Covariates, Mismatched Model

We state a bound on the prediction error for PCR in the general setting where the covariates are not necessarily low-rank. We also consider the scenario where the response variables may satisfy the linear model but with error, i.e., the linear model is mismatched. Precisely, rather than satisfying (1), we assume the response variables are generated in the following manner: for each $i \in [N]$, the random response $Y_i$ is associated with the covariate $\boldsymbol{A}_{i,\cdot} \in \mathbb{R}^{1 \times p}$ such that

$$Y_i = \boldsymbol{A}_{i,\cdot} \beta^* + \phi_i + \epsilon_i, \tag{3}$$

where $\beta^* \in \mathbb{R}^p$ remains the unknown model parameter, $\epsilon_i \in \mathbb{R}$ again denotes the zero mean response noise satisfying Property (2.3), and $\phi_i \in \mathbb{R}$ is the arbitrary mismatch error; for simplicity, we assume the mismatch error is deterministic. In contrast to Property 2.2, we do not assume the covariate matrix $\boldsymbol{A}$ is necessarily low-rank. However, as in Section 2, we assume the other properties hold, i.e., the conditions on the observed (noisy) covariate matrix $\boldsymbol{Z}$ and training subset $\Omega \in [N]$ of size $n$. As before, our interest is in bounding the prediction error of PCR, but we now do so in the general setting where $\boldsymbol{A}$ is not necessarily low-rank and there exists a mismatch error in the linear model.

## 5.1 Theorem Statements

**Training Prediction Error.** We first state a somewhat abstract result, Theorem 5.1 (proof in Appendix F). Next, we state a technical property of HSVT, Lemma 5.1 (proof in Appendix H). Together, they yield a concrete result, Corollary 5.1. For definitions on vector/matrix norms, see Appendix C.

**Theorem 5.1 (Training Error of PCR: Generic Result)** *Consider PCR with parameter $k \geq 1$. Suppose Property 2.3 holds. Then, under the model described by* (3),

$$\mathrm{MSE}_\Omega(\widehat{Y}) \leq \frac{4\sigma^2 k}{n} + \frac{3\|\beta^*\|_1^2 \mathbb{E}\|\boldsymbol{A}^\Omega - \widehat{\boldsymbol{A}}^\Omega\|_{2,\infty}^2}{n} + 20\|\phi\|_\infty^2. \tag{4}$$

*Interpretation.* The bound in (4) has three terms on the right hand side: (a) $\sigma^2 k/n$ represents the standard "regression" prediction error, which scales with the model complexity $k$ and inversely with number of samples $n$; (b) $\|\beta^*\|_1^2 \cdot \mathbb{E}\|\boldsymbol{A}^\Omega - \widehat{\boldsymbol{A}}^\Omega\|_{2,\infty}^2/n$, which is a consequence of the corruption of $\boldsymbol{A}$ (if $\boldsymbol{A}$ was fully observed and rank $k$, then this error term would vanish); (c) $\|\phi\|_\infty^2$ represents the impact of the model mismatch.

*Quantification.* To quantify (4), we need to evaluate $\mathbb{E}[\|\boldsymbol{Z}^{\mathrm{HSVT},k,\Omega} - \boldsymbol{A}^\Omega\|_{2,\infty}^2]$ under HSVT. In effect, HSVT produces the estimate $\boldsymbol{Z}^{\mathrm{HSVT},k}$ of $\boldsymbol{A}$ from its noisy and sparse instantiation $\boldsymbol{Z}$. Our interest is in evaluating the estimation error with respect to the $\ell_{2,\infty}$-error. It is worth remarking that the estimation error for HSVT is typically evaluated with respect to the Frobenius norm; hence, this quantity is well understood, cf. [25]. On the other hand, the error bound with respect to $\ell_{2,\infty}$-norm is unknown. To that end, we provide a novel characterization of this error in Lemma 5.1 below.

Let the SVD of the covariate matrix be $\boldsymbol{A} = \sum_{i=1}^N \tau_i u_i v_i^T$ with the singular values $\tau_i$ arranged in descending order. Let $\boldsymbol{A}^k = \sum_{i=1}^k \tau_i u_i v_i^T$ denote the truncation of $\boldsymbol{A}$ obtained by retaining the top $k$ components. Then for $C(\alpha) > 0$, an absolute constant that depends only on $\alpha$, we define the quantity,

$$\Delta = \sqrt{N\rho}\sqrt{\rho\gamma^2 + (1-\rho)\Gamma^2} + 2C(\alpha)\sqrt{p}(K_\alpha + \Gamma)\left(1 + 9\log(Np)\right)^{\frac{1}{\alpha}}\sqrt{\log(Np)}. \tag{5}$$

**Lemma 5.1 ($\|\cdot\|_{2,\infty}$ error bound for HSVT)** *Let Properties 2.1, 2.3, 2.4, and 2.5 hold. Let $\tau_k$ and $\tau_{k+1}$ denote the $k$-th and $(k+1)$-st singular values of $\boldsymbol{A}$, respectively. Suppose $\rho \geq \frac{64\log(Np)}{Np}$. Then, for $C > 0$, a universal constant.*

$$\mathbb{E}[\|\boldsymbol{Z}^{\mathrm{HSVT},k} - \boldsymbol{A}\|_{2,\infty}^2] \leq \frac{C(K_\alpha^2 + \Gamma^2)}{\rho^2}\left(k + \frac{N\Delta^2}{\rho^2(\tau_k - \tau_{k+1})^2}\right)\log^{\frac{2}{\alpha}} Np + 2\|\boldsymbol{A}^k - \boldsymbol{A}\|_{2,\infty}^2.$$

**Corollary 5.1 (Training Error of PCR: Generic Result)** *Let the conditions of Theorem 5.1 and Lemma 5.1 hold. Let $n = \Theta(N)$. Then, for $C' = (1 + \gamma + \Gamma + K_\alpha)^4$ and $C(\alpha) > 0$ is a constant that may depend on $\alpha \geq 1$, we have*

$$\mathrm{MSE}_\Omega(\widehat{Y}) \leq \frac{4\sigma^2 k}{n} + \frac{C(\alpha)C'\|\beta^*\|_1^2 \log^2 np}{n\rho^2}\left(\frac{(n^2\rho + np)\log^3 np}{\rho^2(\tau_k - \tau_{k+1})^2} + k\right) + \frac{6\|\beta^*\|_1^2}{n}\|\boldsymbol{A}^k - \boldsymbol{A}\|_{2,\infty}^2 + 20\|\phi\|_\infty^2. \tag{6}$$

**Test Prediction Error.** Theorem 4.2 holds with $r$ replaced by a general $k$.

*How do we pick a good $k$ in practice?* The purpose of test prediction error, such as that implied by Theorem 4.2, is to precisely resolve such a question. Specifically, Theorem 4.2 suggests that the overall error is at most the training error plus a term that scales as $k^{3/2}/\sqrt{n}$. Therefore, one should choose the $k$ that minimizes this bound. Naturally, as $k$ increases, the training error is likely to decrease, but the additional term $k^{3/2}/\sqrt{n}$ will increase; therefore, a unique minima in terms of the value of $k$ exists and can be found in a data-driven manner.

## 5.2 Example

To explain the utility of Theorem 5.1 and Corollary 5.1, we consider a setting where $\boldsymbol{A}$ is an approximately low-rank matrix with geometrically decaying singular values. To that end, let $e_{\cdot,j} \in \mathbb{R}^p$ denote the $j$-th canonical basis vector. Let $u_i$, $v_i$, and $\tau_i$ denote the left singular vectors, right singular vectors, and singular values of $\boldsymbol{A}$, respectively. Let $\tau_1 = C_1\sqrt{Np}$ for some constant $C_1 > 0$. Further, suppose $\tau_k = \tau_1 \theta^{k-1}$ for all $k \in [N]$ with $\theta \in (0,1)$, and let $v_i^T e_j = O(1/\sqrt{p})$ for all $i, j \in [p]$. The conditions stated above are self-explanatory with potentially one exception: $v_i^T e_j = O(1/\sqrt{p})$. In effect, this assumption states that the right singular vectors of $\boldsymbol{A}$ satisfy an "incoherence" condition, cf. [22], with the natural basis; or, equivalently, all elements of the right singular vectors are roughly of the same magnitude, $O(1/\sqrt{p})$. Under this setting, we state the following (proof in Appendix M):

**Proposition 5.1** *Let $\boldsymbol{A}$ be generated as above and let conditions of Corollary 5.1 hold. Suppose PCR chooses parameter $k = C_2 \frac{\log \log(np)}{\log(1/\theta)}$ for absolute constant $C_2 > 0$. Then, for $C' = (1+\gamma+\Gamma+K_\alpha)^4$, $C'(\alpha, \theta) > 0$ a constant dependent only on $\alpha$ and $\theta$; and $C'' > 0$, a universal constant, we have*

$$\mathrm{MSE}_\Omega(\widehat{Y}) \leq \frac{2C_2\sigma^2 \log\log np}{n\log(1/\theta)} + \frac{C'(\alpha,\theta)C'\|\beta^*\|_1^2 \log^{5+2C_2} np}{n\rho^4} + \frac{C''\|\beta^*\|_1^2}{\log^{2C_2} np} + 20\|\phi\|_\infty^2, \quad (7)$$

From Proposition 5.1, it follows that if the number of principal components is chosen as $\Theta\left(\frac{\log\log(np)}{\log(1/\theta)}\right)$ and $n = \Omega(\rho^{-4}\mathrm{poly}(\log p))$, then the training prediction error is effectively $\|\phi\|_\infty^2$ for sufficiently large $n, p$. This is precisely the unavoidable model mismatch error.

**Existence of such a matrix.** Here, we show that there exists a matrix with exponentially decaying singular values that also satisfies the required properties of our theorem. We will construct an example based on the incoherence between the canonical basis and the Discrete Fourier Transform (DFT) basis. Suppose that $\boldsymbol{A} = \boldsymbol{U\Sigma V}^T$, where (i) $\boldsymbol{\Sigma}$ is a diagonal matrix such that $\Sigma_{11} = C\sqrt{Np}$ for some $C > 0$ and the diagonal entries of $\boldsymbol{\Sigma}$ satisfy $0 \leq (\Sigma_{i+1,i+1})(\Sigma_{i,i}) \leq \theta$ for all $i \in [N \wedge p - 1]$ and for some $\theta \in (0,1)$; (ii) $\boldsymbol{U} \in \mathbb{R}^{N \times N}$ is a DFT matrix such that $U_{ij} = (1/\sqrt{N}) \cdot e^{\boldsymbol{i}\frac{2\pi}{N}(i-1)(j-1)}$ for all $i, j \in [N]$, where $\boldsymbol{i}$ denotes the imaginary unit; (iii) $\boldsymbol{V} \in \mathbb{R}^{p \times p}$ is a DFT matrix such that $V_{ij} = (1/\sqrt{p}) \cdot e^{\boldsymbol{i}\frac{2\pi}{p}(i-1)(j-1)}$ for all $i, j \in [p]$.

The entries of the resulting matrix $\boldsymbol{A}$ are complex numbers, but one could also construct $\boldsymbol{A}$ by taking $\boldsymbol{U}$ and $\boldsymbol{V}$ as discrete cosine (or sine) transform matrices. Further, observe that $\boldsymbol{U}$ and $\boldsymbol{V}$ are orthonormal matrices; hence, $\sigma_i(\boldsymbol{A}) = \sigma_i(\boldsymbol{\Sigma})$ for all $i \in [N \wedge p]$. Next, we argue that $\|\boldsymbol{A}\|_{\max} \leq C'$ for some constant $C' > 0$ (the proof of which can be found in Appendix L.2).

**Lemma 5.2** *Let $\boldsymbol{A}$ be generated as above. Then, $\|\boldsymbol{A}\|_{\max} \leq \frac{C}{1-\theta}$ Here, $C > 0$ and $\theta \in (0,1)$ are the constants that appear in the description of $\boldsymbol{\Sigma}$.*

# 6 Applications

Given the ubiquity of PCR in practice, we describe four concrete, important applications that are enabled (and theoretically justified) by our formulation and the associated finite sample analyses results: (i) causal inference (synthetic control); (ii) time series forecasting; (iii) privacy preserving learning; (iv) regression with mixed valued covariates. We choose these examples as they showcase the broad meaning of "error" with respect to the covariates. (i)-(ii) are related to measurement error (as is commonly assumed with temporal data); (iii) is when noise is added to the covariates by design (in this example, to ensure differential privacy); (iv) is when the structure of the covariates restricts our observations to only its noisy instantiations (in this example, the latent covariate of interest is a "continuous" Bernoulli parameter, but we only observe its discrete $0/1$ categorical instantiation). Due to space constraints, we defer the detailed description of these applications to Appendix B.

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
