[Supplementary Material · neurips_appendix_2019.pdf]

## A  Matrix Estimation Literature Review

Over the past decade, matrix estimation has spurred tremendous theoretical and empirical research across numerous fields, including recommendation systems (cf. [38, 39, 42, 28, 25, 40, 23, 44, 30]), social network analysis (cf. [3, 4, 5, 12, 34]), and graph learning (graphon estimation) (cf. [8, 55, 21, 20]). Traditionally, the end goal is to recover the underlying mean matrix from an incomplete and noisy sampling of its entries; the quality of the estimate is often measured through the Frobenius norm. Further, entry-wise independence and sub-gaussian noise is typically assumed. A key property of many matrix estimation methods is that they are model agnostic (i.e., the de-noising procedure does not change with the noise assumptions). We advance state-of-art for HSVT, a specific matrix estimation method, by: (i) analyzing its error with respect to the $\|\cdot\|_{2,\infty}$ matrix norm; (ii) allow for a broader class of noise distributions (e.g., sub-exponential). Such generalizations are necessary to enable the various applications detailed in Section 6.

## B  Applications

Throughout this section, $N, n, p, Y, \beta^*, \boldsymbol{A}, \boldsymbol{X}, \boldsymbol{Z}$, and $\Omega$, will be as in Section 2. Recall that $\boldsymbol{A} \in \mathbb{R}^{N \times p}$ denotes the underlying covariate matrix, where its entries are assumed to be bounded by $\Gamma$ (Property 2.2). Recall, $\boldsymbol{X} \in \mathbb{R}^{N \times p}$ is a random matrix with independent rows and the $(i, j)$-th element $X_{ij}$ has expected value equal to $A_{ij}$ (Property 2.5). $\boldsymbol{Z}$ is then the masked version of $\boldsymbol{X}$, denoting the observed, corrupted covariate matrix; here, the $(i, j)$-th element $Z_{ij} = X_{ij}$ is observed with probability $\rho$ and $Z_{ij} = \star$ is unobserved with probability $1 - \rho$ (Property 2.4). The (noisy) response vector is denoted by $Y \in \mathbb{R}^N$ and is equal in expectation to $\boldsymbol{A}\beta^*$ (Property 2.3). Finally, $\Omega \in [N]$ represents the training set of size $n$; thus, $Y^\Omega \in \mathbb{R}^n$ denotes the elements of $\boldsymbol{Y}$ that are observed, while $\boldsymbol{Z}^\Omega \in \mathbb{R}^{n \times p}$ represents the observed covariates corresponding to the observed responses.

For each application, we first introduce the required the mathematical notation to formally describe the problem and second, show how the results in the preceding sections immediately provide meaningful finite sample analyses.

### B.1  Synthetic Control

**Problem formulation.** Synthetic control is a popular method for comparative case studies and policy evaluation in econometrics to predict a counterfactual for a unit of interest after its exposure to a treatment. To do so, a synthetic treatment unit is constructed using a combination of so-called "donor" units. Proposed by [2], it has been analyzed in [11], [1], [32], [35], [14], [13]. A canonical example is in [1], where the unit of interest is California, the donor pool is all other states in the U.S., and the treatment is Proposition 99; the goal is to isolate the effect of Proposition 99 on cigarette consumption in California. In other words, to evaluate the treatment effect, synthetic control methods estimate the unobservable counterfactual (tobacco consumption in the absence of Proposition 99 in the example above) for the target unit using observations from the donor units, which are assumed to be unaffected by the treatment.

More generally, both the target unit and each donor unit is associated with a time series over $N$ periods (e.g., a time series of monthly cigarette consumption for each U.S. state in the example above). Let $p$ denote the number of donors. Suppose the intervention occurs at time $n$, where $1 \leq n < N$. We will refer to the pre- and post- intervention periods as the time periods prior to and after the intervention point, and denote $\Omega = [n]$ as the pre-treatment indices.

Let $\boldsymbol{A} \in \mathbb{R}^{N \times p}$ represent the true utilities of the $p$ donor units across the entire time horizon $N$; hence, $\boldsymbol{A}_{\cdot,j} \in \mathbb{R}^N$ represents the time series over $N$ periods for donor $j \in [p]$. Rather than observing $\boldsymbol{A}$, we are given access to $\boldsymbol{Z} \in \mathbb{R}^{N \times p}$, a sparse, noisy instantiation of $\boldsymbol{A}$.

For every $i \in [N]$, let $Y_i$ denote the noisy utility of the target unit in the absence of intervention. However, since the target unit experiences treatment for all time instances $n < i \leq N$, we only have access to a noisy version of the target unit's utility for the pre-intervention period, i.e., we only observe $Y^\Omega = [Y_i]$ for $i \in [n]$. We will denote $\mathbb{E}[Y]$ as the true, latent utility for the target unit if the intervention never occurred. Hence, given data $(Y^\Omega, \boldsymbol{Z})$, the aim is to recover $\mathbb{E}[Y]$. Please refer to Figure 1 for a graphical overview of the setup of the problem.

**Figure 1:** Caricature of observed data $(Y^{\Omega}, \boldsymbol{Z})$ for synthetic control (with $\star$ denoting unobserved data in the donor matrix). Here, "?" represents the counterfactual observations for the target unit in the absence of intervention.

**How it fits our framework.** We now motivate applying PCR to this problem by arguing that: (i) $\boldsymbol{A}$ is (approximately) low-rank; (ii) a linear relationship exists between the target units and the donor units (i.e., why linear regression is justified).

*Why is $\boldsymbol{A}$ low-rank?* A natural generalization of the typical *factor model*, which is commonly utilized in the Econometrics literature (cf. [2], [1]), is the generic latent variable model (LVM). LVMs are known to be general nonlinear models that capture complex latent structures in various applications. Let $\boldsymbol{A}' = [A'_{ij}] \in \mathbb{R}^{N \times (p+1)}$ be the concatenation of $\boldsymbol{A}$ with the vector of underlying utilities for the target unit, $\mathbb{E}[Y]$. In particular, let $\boldsymbol{A}'_{\cdot,0} = \mathbb{E}[Y]$ represent the true utility vector for the target unit, and let $\boldsymbol{A}'_{\cdot,j} = \boldsymbol{A}_{\cdot,j}$ for all $j \in [p]$ denote the true utilities for the donor pool. Thus, $\boldsymbol{A}'$ represents the matrix of mean utilities for all $p + 1$ units (target unit and donor pool) in the absence of intervention. If the underlying matrix $\boldsymbol{A}'$ is generated as per a LVM, then

$$A'_{ij} = g(\theta_i, \rho_j). \tag{8}$$

Here, $\theta_i \in \mathbb{R}^{d_1}$ and $\rho_j \in \mathbb{R}^{d_2}$ are latent feature vectors capturing unit and time specific information, respectively, for some $d_1, d_2 \geq 1$; and the latent function $g : \mathbb{R}^{d_1} \times \mathbb{R}^{d_2} \to \mathbb{R}$ captures the model relationship. If $g$ is "well-behaved" (i.e. Lipschitz) and the latent spaces are compact, then it can be seen that $\boldsymbol{A}'$ (and consequently $\boldsymbol{A}$) is approximately low-rank. This is made more rigorous by the following proposition.

**Proposition B.1 (Proposition 2.1 of [10])** *Let $\boldsymbol{A}'$ satisfy* (8). *Let $g$ be an $\mathcal{L}$-Lipschitz function with $\theta_i \in [0,1]^{d_1}$ and $\rho_j \in [0,1]^{d_2}$ for all $i \in [N]$ and $j \in \{0, \ldots, p\}$. Then, for any $\delta > 0$, there exists a low-rank matrix $\boldsymbol{T}$ of rank $r \leq C \cdot \delta^{-\min(d_1, d_2)}$ such that*

$$\|\boldsymbol{A}' - \boldsymbol{T}\|_{\max} \leq \mathcal{L} \cdot \delta.$$

*Here, $C$ is a constant that depends on the latent spaces $([0,1]^{d_1}, [0,1]^{d_2})$, ambient dimensions $(d_1, d_2)$, and Lipschitz constant $\mathcal{L}$.*

Further theoretical justification of the LVM model comes from the celebrated result by cf. [9], which states that a LVM is indued when the underling $\boldsymbol{A}'$ is random and its rows/columns are exchangeable Indeed, it has been shown empirically that many real-world data exhibit low-rank donor pool matrices (e.g., see Figure 1 of [11] and Figure 3 of [10]). Thus this (approximate) low-rank structure induced by LVMs motivates why we make a low-rank assumption on $\boldsymbol{A}$, the donor pool matrix.

*Why use linear regression?* Here, we justify the usage of linear regression by arguing that the target unit is a linear combination of the donor units with high probability. This is made rigorous by the following Proposition.

**Proposition B.2** *(Proposition 4.1 of [10]) Assume Property 2.2 holds and that the rank of $\boldsymbol{A}'$ is bounded by $r$. Suppose the target unit is chosen uniformly at random amongst the $p + 1$ units; equivalently, let the units be re-indexed as per some permutation chosen uniformly at random. Then, with probability $1 - r/(p + 1)$, there exists a $\beta^* \in \mathbb{R}^p$ such that the target unit (represented by index 0) satisfies for all $j \in [N]$,*

$$A'_{j0} = \sum_{k=1}^{p} \beta_k^* \cdot A'_{jk}.$$

Thus, under the low-rank property of $\boldsymbol{A}$, the target unit is shown to be a linear combination of the donor units with high probability. Assuming (without loss of generality) that target unit corresponds to column index $j = 0$ of $\boldsymbol{A}'$ (as described above), we can express the underlying utility of the target unit as

$$\mathbb{E}[Y_i] = \boldsymbol{A}'_{i0} = \boldsymbol{A}_{i,\cdot}\beta^*$$

for all $i \in [N]$ and some $\beta^* \in \mathbb{R}^p$. Hence, in effect, observations generated via a generic latent variable model (as per (8)), which encompass a large class of models, naturally fit within the synthetic control framework.

**Results.** We now present the post-intervention prediction error, which is a corollary of Theorem 4.2. Here, we assume that the underlying covariate matrix $\boldsymbol{A}$ is low-rank.

**Corollary B.1** *Let the conditions of Theorem 4.2 and Proposition B.2 hold. Then with probability at least $1 - r/(p+1)$,*

$$\mathbb{E}\left[\frac{1}{N-n}\sum_{i=n+1}^{N}\left(\widehat{\boldsymbol{A}}_{i,\cdot}\widehat{\beta} - \boldsymbol{A}_{i,\cdot}\beta^*\right)^2\right]$$

$$\leq C_1 \cdot \left(\frac{4\sigma^2 r}{n} + C(\alpha)\frac{C'\log^2(np)}{n\rho^2}\|\beta^*\|_1^2\left(r + \frac{(n^2\rho + np)\log^3(np)}{\rho^2\tau_r^2}\right)\right) + C_2 r^2\widehat{\alpha}^2 \cdot \sqrt{\frac{\log(np)}{n}},$$

*where $C_1 > 0$ is a universal constant; $C' = (1 + \gamma + \Gamma + K_\alpha)^4$; $C_2 = CB^2 \cdot \Gamma\|\beta^*\|_1$ with $C > 0$ a universal constant; $C(\alpha) > 0$ a constant that may depend on $\alpha \geq 1$; $\tau_r$ is $r$th singular value of true covariate matrix $\boldsymbol{A}$; and $\widehat{\alpha}^2 = \mathbb{E}[\|\widehat{\boldsymbol{A}}\|_{\max}^2]$.*

## B.2 Time Series Analysis

**Problem formulation.**

We follow the formulation in [7]. Specifically, consider a discrete-time setting with $t \in \mathbb{Z}$ representing the time index and $f : \mathbb{Z} \to \mathbb{R}$ representing the latent time series of interest. The underlying time series is denoted as $f = [f(t)]$ for all $t \in [T]$. For each $t \in [T]$ and probability $\rho \in (0,1]$, the random variable $X(t)$ such that $\mathbb{E}[X(t)] = f(t)$ is observed. Under this setting, the objective of interest is to accurately interpolate (impute) the $T$ observations and forecast the evolution of the underlying time series at time $T + 1$, i.e., given access to the time series $X(t)$ for $t \in [T]$, our goal is to estimate $f(t)$ for all $t \in [T + 1]$.

We focus on the case where $f$ follows a Linear Recurrent Formulae (LRF)[1]. LRFs have the following form:

$$f(t) = \sum_{l=1}^{r} \alpha_l f(t - l). \tag{9}$$

LRFs admit a rich class of time series as seen by the following proposition.

**Proposition B.3** *(**Proposition 5.2 in [7]**) Let $P_{m_a}$ be a polynomial of degree $m_a$. Then,*

$$f(t) = \sum_{g=1}^{G} \exp(\alpha_g t) \cdot \cos(2\pi\omega_g t + \phi_g) \cdot P_{m_g}(t)$$

*admits a representation as in (9). Further the order $r$ of $f(t)$ is independent of $T$, the number of observations, and is bounded by*

$$r \leq G(m_{\max} + 1)(m_{\max} + 2),$$

*where $m_{\max} = \max_{g \in G} m_g$.*

**How it fits our framework.** We motive why PCR works for this problem by first showing how $N, n, p, Y, \beta^*, \boldsymbol{A}, \boldsymbol{X}, \boldsymbol{Z}$ are induced for the forecasting problem defined above for a LRF (refer to (9)). Let $T = N \times p$ with $p > r$ and $n = T$. We then have

$$Y^\Omega = [X(p), X(2p), \ldots, X(T - p)] \in \mathbb{R}^n$$

**Figure 2:** Caricature of the underlying time series $f(t)$ for $t \in [T]$ generated as per a LRF model (described by (9)) represented as a matrix vector product.

$$\boldsymbol{A} = [A_{ij}] = [f((i-1)p + j)], \text{where } \boldsymbol{A} \in \mathbb{R}^{N \times (p-1)}$$

$$\boldsymbol{Z} = [Z_{ij}] = [X((i-1)p + j) \cdot \pi_{ij}], \text{where } \boldsymbol{Z} \in \mathbb{R}^{N \times (p-1)}, \text{and } \pi_{ij} \sim \text{Bernoulli}(\rho)$$

$$\beta^* = [0, 0 \ldots, \alpha_1, \ldots, \alpha_r] \in \mathbb{R}^{p-1}$$

In words, $\boldsymbol{A}$ is a matrix of non-overlapping entries of the underlying, unobserved time series $f$. Please refer to Figure 2 for a graphical depiction of the matrices/vectors induced by the latent time series $f$. $\boldsymbol{Z}$ is analogously defined with respect to the noisy, sparse observations. $\beta^*$ refers to the (unobserved) coefficients that define the LRF. $Y^{\Omega}$ are observations within the time series we use as response variables.

**Results.** It is easy to see that $\text{rank}(\boldsymbol{A}) \leq r$. We can now immediately apply Theorem 4.2 to bound the overall imputation and forecast error under PCR.

**Corollary B.2** *Let the conditions of Theorem 4.2 hold. Suppose the underlying time series $f$ and rank $r$ are defined as in (9). Then,*

$$\mathbb{E}\left[\frac{1}{T+1} \sum_{t=1}^{T+1} \left(f(t) - \widehat{f}(t)\right)^2\right]$$

$$\leq \frac{4\sigma^2 r}{T} + C(\alpha)\frac{C' \log^2(Tp)}{T\rho^2}\|\beta^*\|_1^2\left(r + \frac{(T^2\rho + Tp)\log^3(Tp)}{\rho^2\tau_r^2}\right) + C_2 r^2 \widehat{\alpha}^2 \cdot \sqrt{\frac{\log(Tp)}{T}},$$

*where $C' = (1 + \gamma + \Gamma + K_\alpha)^4$; $C_2 = CB^2 \cdot \Gamma\|\beta^*\|_1$ with $C > 0$ a universal constant; $C(\alpha) > 0$ a constant that may depend on $\alpha \geq 1$; $\tau_r$ is $r$th singular value of true covariate matrix $\boldsymbol{A}$; and $\widehat{\alpha}^2 = \mathbb{E}[\|\widehat{\boldsymbol{A}}\|_{\max}^2]$.*

## B.3 Regression with Privacy

**Problem formulation.** With the increasing use of machine learning for critical operations, analysts must maximize the accuracy of their predictions and simultaneously protect sensitive information (i.e. covariates). An important notion of privacy is that of differential privacy; this requires that the outcome of a database query cannot greatly change due to the presence or absence of any individual data record (cf. [33] and references therein). More specifically, let $\delta$ be a positive real number, $\mathcal{D}$ be a collection of datasets, and $\mathcal{A} : \mathcal{D} \to \text{im}(\mathcal{A})$ be a randomized algorithm that takes a dataset as input. The algorithm $\mathcal{A}$ is said to provide $\delta$-differential privacy if, for all datasets $\mathcal{D}_1$ and $\mathcal{D}_2$ in $\mathcal{D}$ that differ on a single element, and all subsets $\mathcal{S} \in \text{im}(\mathcal{A})$, the following holds:

$$\mathbb{P}\left(\mathcal{A}(\mathcal{D}_1) \in S\right) \leq \exp(\delta) \cdot \mathbb{P}\left(\mathcal{A}(\mathcal{D}_2) \in S\right), \tag{10}$$

where the randomness lies in the algorithm. Thus, (10) guarantees that little can be learned about any particular record within the database.

One popular mechanism $\mathcal{A}$ to guarantee differential privacy is known as the Laplacian mechanism. In this setting, noise is drawn from a Laplacian distribution and added to query responses. In particular, introducing additive noise $W \sim \text{Laplace}(0, \Delta_f/\delta)$ to any database query guarantees $\delta$-privacy (cf. [33] and references therein); here, $\Delta_f = \max_{\mathcal{D}_1, \mathcal{D}_2 \in \mathcal{D}} |f(\mathcal{D}_1) - f(\mathcal{D}_2)|$, where the maximum is taken over all pairs of datasets $\mathcal{D}_1$ and $\mathcal{D}_2$ in $\mathcal{D}$ differing in at most one element, and $f : \mathcal{D} \to \mathbb{R}^d$ is a vector-valued function denoting the true, latent query response. We now describe how PCR can be applied in the context of a differentially private framework.

**How it fits our framework.** Let $A$ denote the true, fixed database of $N$ sensitive individual records and $p$ covariates. We consider the setting where an analyst is allowed to ask two types of queries of the data: (1) $f_A$ - querying for individual data records, i.e., $A_{i,\cdot}$ for $i \in [N]$; (2) $f_Y$ - querying for a linear combination of an individual's covariates, i.e. $A_{i,\cdot}\beta^*$. A typical example would be where $A_{i,\cdot}$ is the genomic information for patient $i$ and $A_{i,\cdot}\beta^*$ denotes patient $i$'s outcome for a clinical study.

In order to provide $\delta$-differential privacy, the Laplacian mechanism will return query responses with additive Laplacian noise. For query type (1), let $Z_{ij}$ for $j \in [p], i \in [N]$ by the returned response; here, $Z_{ij} = A_{ij} + \eta_{ij}$ with probability $\rho$ and $Z_{ij} = \star$ with probability $1 - \rho$, where $\eta_{i,\cdot} = [\eta_{ij}]$ for $j \in [p]$ is independent Laplacian noise with the variance parameter proportional to $\Delta_{f_A}/\delta^2$. For query type (2), when an analyst queries for the response variable $A_{i,\cdot}\beta^*$, she observes $Y_i = A_{i,\cdot}\beta^* + \epsilon_i$, where $\epsilon_i$ is again independent Laplacian noise with variance parameter proportional to $\Delta_{f_Y}/\delta$. We note that the above setup naturally fits our framework since the Laplacian distribution belongs to the family of sub-exponential distributions i.e. satisfying Property 2.5 with $\alpha = 1$.

Finally, let $Y^\Omega$ denote the $n$ noisy observed responses (e.g., corresponding to the outcomes of $n$ patient clinical trials), and let $Z$ denote the noisy observed covariates (e.g., the collection of genomic information of all $N$ patients). Ultimately, the goal in such a setup is to accurately learn in- and out-of-sample global statistics (e.g., having low $\text{MSE}_\Omega(\widehat{Y})$ and $\text{MSE}(\widehat{Y})$ respectively) about the data, while preserving the individual privacy of the users.

*Why is privacy preserved?* It is worth highlighting that the de-noising step of the PCR algorithm (i.e., applying HSVT to $Z$) does not compromise the security of any single data record. To begin, Lemma 5.1 demonstrates that the estimated covariate matrix $\widehat{A}$ via HSVT achieves small average $\ell_{2,\infty}$-error (column-squared error); hence, for instance, HSVT can accurately learn the *average* age of all patients. However, this does not translate to accurately estimating the age of any particular patient (i.e., corresponding to an entry-wise error bound). Similarly, Corollary B.3 (stated below), establishes that PCR can estimate the vector $A\beta^*$ well on average, but not any particular element of this vector. Hence, the privacy of any individual record is maintained while small average prediction error is achieved.

**Results.** We now state the following theorem, which demonstrates the efficacy of PCR (with respect to prediction) in the context of a differentially private framework.

**Corollary B.3** *Consider PCR with parameter $k \geq 1$. Let conditions of Theorem 5.1 and Lemma 5.1 hold. Let $\eta_{ij}$ be sampled independently from $\sim \text{Laplace}(0, \Delta_{f_A}/\delta)$ for $i \in [N], j \in [p]$. Let $\epsilon_i$ be sampled independently from $\sim \text{Laplace}(0, \Delta_{f_Y}/\delta)$. Let $n = \Theta(N)$. Then, PCR preserves $\delta$-differential privacy of $A$ and $A\beta^*$, and*

$$\text{MSE}_\Omega(\widehat{Y}) \leq \frac{4k}{n}\cdot\left(\frac{\Delta_{f_Y}}{\delta}\right)^2 + \frac{CC'\|\beta^*\|_1^2 \log^2 np}{n\rho^2}\left(\frac{(n^2\rho + np)\log^3 np}{\rho^2(\tau_k - \tau_{k+1})^2} + k\right) + \frac{6\|\beta^*\|_1^2}{n}\|A^k - A\|_{2,\infty}^2,$$

*where $C' = \left(1 + \gamma + \Gamma + \frac{\Delta_{f_A}}{\delta}\right)^4$ and $C > 0$ is an absolute constant.*

Note test prediction error is bounded as in Theorem 4.2 without any change with $r$ replaced by $k$ for PCR with parameter $k \geq 1$.

## B.4 Regression with Mixed Valued Features

**Problem formulation.** Regression models with mixed discrete and continuous covariates are ubiquitous in practice. With respect to discrete covariates, a standard generative model assumes the covariates are generated from a categorical distribution (i.e., a generalized Bernoulli distribution). Formally, a categorical distribution for a random variable $X$ is such that $X$ has support in $[G]$ and the probability mass function (pmf) is given by $\mathbb{P}(X = g) = \rho_g$ for $g \in [G]$ with $\sum_{g=1}^{G}\rho_j g = 1$.

For simplicity, we focus on the case where the regression is being done with a collection of Bernoulli random variables (i.e., each $X$ has support in $\{0, 1\}$). The extension to general categorical random variables, in addition to continuous covariates, is straightforward and discussed below.

A standard model in regression with Bernoulli random variables assumes that the response variable is a linear function of the latent parameters of the observed discrete outcomes. Formally, $\boldsymbol{A}_{i,\cdot} = [\rho_1^{(i)}, \rho_2^{(i)}, \ldots, \rho_p^{(i)}] \in \mathbb{R}^{1 \times p}$, where $\rho_j^{(i)}$ for $j \in [p]$ is the latent Bernoulli parameter for the $j$-th feature and $i$-th measurement. Further, the mean of the response variable satisfies $\mathbb{E}[Y_i] = \sum_{j=1}^p \rho_j^{(i)} \beta_j$. Unfortunately, for each feature, we only get binary observations, i.e., $X_{ij} \in \{0, 1\}$.

As an example, consider $\mathbb{E}[Y_i]$ to be the expected health outcome of patient $i$. Let there be a total of $p$ possible observable binary symptoms (e.g., cold, fever, headache, etc.). Then $\boldsymbol{A}_{i,\cdot}$ denotes the vector of (unobserved) probabilities that patient $i$ has some collection of symptoms (e.g., $A_{i1} = \mathbb{P}(\text{patient } i \text{ has a cold}), A_{i2} = \mathbb{P}(\text{patient } i \text{ has a fever}), \ldots$). However, for each patient, we only observe the binary outcome of these symptoms (i.e., $X_{i1} = \mathbb{1}(\text{patient } i \text{ has a cold}), X_{i2} = \mathbb{1}(\text{patient } i \text{ has a fever})$), even though the response is linearly related with the underlying probabilities of the symptoms. The objective in such a setting is to accurately recover $\boldsymbol{A}\beta^*$ given $Y^\Omega$ and $\boldsymbol{X}$.

*Current practice for mixed valued features.* A common practice for regression with categorical variables is to build a separate regression model for every possible combination of the categorical outcomes (i.e., to build a separate regression model conditioned on each outcome). In the healthcare example above, this would amount to building $2^p$ separate regression models corresponding to each combination of the observed $p$ binary symptoms. This is clearly not ideal for the following two major reasons: (i) the sample complexity is exponential in $p$; (ii) we do not have access to the underlying probabilities $\boldsymbol{A}_{i,\cdot}$ (recall $\boldsymbol{X}_{i,\cdot} \in \{0, 1\}^p$), which is what we actually want to regress $Y^\Omega$ against.

**How it fits our framework.** Recall from Property 2.5 that the key structure we require of the covariate noise $\eta_{ij}$ is that $\mathbb{E}[\eta_{ij}] = 0$. Now even though $X_{ij} \in \{0, 1\}$, it still holds that $\mathbb{E}[X_{ij}] = \rho_j^{(i)} = A_{ij}$, which immediately implies $\mathbb{E}[\eta_{ij}] = \mathbb{E}[X_{ij} - A_{ij}] = 0$. Further, $\eta_{ij}$ is sub-Gaussian ($\alpha = 2$) since $|\eta_{ij}| \leq 1$. Thus, the key conditions on the noise are satisfied for PCR to effectively (in the $\|\cdot\|_{2,\infty}$-norm) de-noise $\boldsymbol{X}$ to recover the underlying probability matrix $\boldsymbol{A}$; this, in turn, allows PCR to produce accurate estimates $\widehat{\boldsymbol{A}}\widehat{\beta}$ through regression, as seen by Theorem 4.2.

Pleasingly, the required sample complexity grows with the rank of $\boldsymbol{A}$ (the inherent model complexity), rather than exponentially in $p$. Further, the de-noising step allows us to regress against the estimated latent probabilities rather than their "noisy", binary outcomes.

*Extension from Bernoulli to general categorical random variables.* Recall from above that a categorical random variable has support in $[G]$ for $G \in \mathbb{N}$. In this case, one can translate a categorical random variable to a a collection of binary random variables using the standard one-hot encoding method. It is worth highlighting that by using one-hot encoding, clearly $\eta_{ij_1}$ will not be independent of $\eta_{ij_2}$ for any $(j_1, j_2)$ pair, which encodes the same categorical variable. However, from Property 2.5, we only require independence of the noise across rows, not within them. Thus this lack of independence is not an issue. Further, the generalization to multiple categorical variables, in addition to continuous covariates, is achieved by simply appending these features to each row and collectively de-noising the entire matrix before the regression step.

# C Matrix and Vector Norm Definitions

In this section, we will define a series of matrix and vector norms. For any vector $v = [v_i] \in \mathbb{R}^n$ and real number $p \geq 1$, we define the $\ell_p$-norm of $v$ as

$$\|v\|_p = \left( \sum_{i=1}^n |v_i|^p \right)^{1/p}.$$

In particular, if $p = 2$, this corresponds to the Euclidean norm, i.e., $\|v\|_2^2 = \sum_{i=1}^n v_i^2$. Similarly, $p = 1$ yields $\|v\|_1 = \sum_{i=1}^n |v_i|$ and $p = \infty$ yields $\max_{i \in [n]} |v_i|$. Further, we define

$$\|v\|_0 = |\{i : v_i \neq 0\}|$$

as the number of nonzero elements of $v$.

For any matrix $\boldsymbol{Q} = [Q_{ij}]^{m \times n}$, we define the Frobenius norm of $\boldsymbol{Q}$ as

$$\|\boldsymbol{Q}\|_F = \left( \sum_{i=1}^m \sum_{j=1}^n Q_{ij}^2 \right)^{1/2},$$

and the spectral (operator) norm as

$$\|\boldsymbol{Q}\| = \sigma_{\max}(\boldsymbol{Q}) = \sup_{v:\|v\|_2=1} \|\boldsymbol{Q}v\|_2,$$

where $\sigma_{\max}(\boldsymbol{Q})$ denotes the largest singular value of $\boldsymbol{Q}$. We define the max-norm of $\boldsymbol{Q}$ as

$$\|\boldsymbol{Q}\|_{\max} = \max_{i \in [m], j \in [n]} |Q_{ij}|.$$

Finally, we denote the $\ell_{2,\infty}$ mixed norm of $\boldsymbol{Q}$ as

$$\|\boldsymbol{Q}\|_{2,\infty} = \max_{j \in [n]} \sum_{i=1}^n Q_{ij}^2.$$

We now define an important class of random variables/vectors.

**Definition C.1** *For any $\alpha \geq 1$, we define the $\psi_\alpha$-norm of a random variable $X$ as $\|X\|_{\psi_\alpha} = \inf\{t > 0 : \mathbb{E}\exp(|X|^\alpha/t^\alpha) \leq 2\}$. If $\|X\|_{\psi_\alpha} < \infty$, we call $X$ a $\psi_\alpha$-random variable. More generally, we say $X$ in $\mathbb{R}^n$ is a $\psi_\alpha$-random vector if all one-dimensional marginals $\langle X, v \rangle$ are $\psi_\alpha$-random variables for any fixed vector $v \in \mathbb{R}^n$. We define the $\psi_\alpha$-norm of the random vector $X \in \mathbb{R}^n$ as $\|X\|_{\psi_\alpha} = \sup_{v \in \mathcal{S}^{n-1}} \|\langle X, v \rangle\|_{\psi_\alpha}$, where $\mathcal{S}^{n-1} := \{v \in \mathbb{R}^n : \|v\|_2 = 1\}$, $\langle \cdot, \cdot \rangle$ usual inner product. Note that $\alpha = 2$ and $\alpha = 1$ represent the class of sub-gaussian and sub-exponential random variables/vectors, respectively.*

# D Useful Theorems

## D.1 Bounding $\psi_\alpha$-norm

**Lemma D.1 Sum of independent sub-gaussians random variables.**
*Let $X_1, \ldots, X_n$ be independent, mean zero, sub-gaussian random variables. Then $\sum_{i=1}^n X_i$ is also a sub-gaussian random variable, and*

$$\left\| \sum_{i=1}^n X_i \right\|_{\psi_2}^2 \leq C \sum_{i=1}^n \|X_i\|_{\psi_2}^2$$

*where $C$ is an absolute constant.*

**Lemma D.2 Product of sub-gaussians is sub-exponential.**
*Let $X$ and $Y$ be sub-gaussian random variables. Then $XY$ is sub-exponential. Moreover,*

$$\|XY\|_{\psi_1} \leq \|X\|_{\psi_2} \|Y\|_{\psi_2}.$$

## D.2 Concentration Inequalities for Random Variables

**Lemma D.3 Bernstein's inequality.**
*Let $X_1, X_2, \ldots, X_N$ be independent, mean zero, sub-exponential random variables. Let $S = \sum_{i=1}^{n} X_i$. Then for every $t > 0$, we have*

$$\mathbb{P}\{|S| \geq t\} \leq 2 \exp\left(-c \min\left[\frac{t^2}{\sum_{i=1}^{N} \|X_i\|_{\Psi_1}^2}, \frac{t}{\max_i \|X_i\|_{\Psi_1}}\right]\right)$$

**Lemma D.4 McDiarmid inequality.**
*Let $x_1, \ldots, x_n$ be independent random variables taking on values in a set $A$, and let $c_1, \ldots, c_n$ be positive real constants. If $\phi : A^n \to \mathbb{R}$ satisfies*

$$\sup_{x_1, \ldots, x_n, x_i' \in A} |\phi(x_1, \ldots, x_i, \ldots, x_n) - \phi(x_1, \ldots, x_i', \ldots, x_n)| \leq c_i,$$

*for $1 \leq i \leq n$, then*

$$\mathbb{P}\left\{|\phi(x_1, \ldots, x_n) - \mathbb{E}\phi(x_1, \ldots, x_n)| \geq \epsilon\right\} \leq \exp\left(\frac{-2\epsilon^2}{\sum_{i=1}^{n} c_i^2}\right).$$

### D.2.1 Upper Bound on the Maximum Absolute Value in Expectation

**Lemma D.5 Maximum of sequence of random variables.**
*Let $X_1, X_2, \ldots, X_n$ be a sequence of random variables, which are not necessarily independent, and satisfy $\mathbb{E}[X_i^{2p}]^{\frac{1}{2p}} \leq K p^{\frac{\beta}{2}}$ for some $K, \beta > 0$ and all $i$. Then, for every $n \geq 2$,*

$$\mathbb{E}\max_{i \leq n} |X_i| \leq CK \log^{\frac{\beta}{2}}(n).$$

**Remark D.1** *Lemma D.5 implies that if $X_1, \ldots, X_n$ are $\psi_\alpha$ random variables with $\|X_i\|_{\psi_\alpha} \leq K_\alpha$ for all $i \in [n]$, then*

$$\mathbb{E}\max_{i \leq n} |X_i| \leq CK_\alpha \log^{\frac{1}{\alpha}}(n).$$

## D.3 Other Useful Lemmas

**Lemma D.6 Perturbation of singular values (Weyl's inequality).**
*Let $\boldsymbol{A}$ and $\boldsymbol{B}$ be two $m \times n$ matrices. Let $k = m \wedge n$. Let $\lambda_1, \ldots, \lambda_k$ be the singular values of $\boldsymbol{A}$ in decreasing order and repeated by multiplicities, and let $\tau_1, \ldots, \tau_k$ be the singular values of $\boldsymbol{B}$ in decreasing order and repeated by multiplicities. Let $\delta_1, \ldots, \delta_k$ be the singular values of $\boldsymbol{A} - \boldsymbol{B}$, in any order but still repeated by multiplicities. Then,*

$$\max_{1 \leq i \leq k} |\lambda_i - \tau_i| \leq \max_{1 \leq i \leq k} |\delta_i|.$$

# E Equivalence

## E.1 Proof of Proposition 3.1

**Proof E.1** *Using the orthonormality of $\boldsymbol{U}, \boldsymbol{V}$, we obtain*

$$\widehat{Y}^{PCR,k} = \widetilde{\boldsymbol{Z}} \cdot \boldsymbol{V}_k \cdot \beta^{PCR,k} \qquad\qquad = \widetilde{\boldsymbol{Z}} \cdot \boldsymbol{V}_k \cdot \left(\boldsymbol{Z}^{PCR,k,\Omega}\right)^\dagger Y^\Omega$$

$$= \boldsymbol{U} \cdot \boldsymbol{S} \cdot \boldsymbol{V}^T \cdot \boldsymbol{V}_k \cdot \left((\widetilde{\boldsymbol{Z}} \cdot \boldsymbol{V}_k)^\Omega\right)^\dagger \cdot Y^\Omega = \boldsymbol{U}_k \cdot \boldsymbol{S}_k \cdot \left((\boldsymbol{U}_k \cdot \boldsymbol{S}_k)^\Omega\right)^\dagger \cdot Y^\Omega$$

$$= \boldsymbol{U}_k \cdot \boldsymbol{S}_k \cdot \left(\boldsymbol{U}_k^\Omega \cdot \boldsymbol{S}_k\right)^\dagger \cdot Y^\Omega \qquad\qquad = \boldsymbol{U}_k \cdot \boldsymbol{T}_k \cdot \boldsymbol{S}_k^{-1} (\boldsymbol{U}_k^\Omega)^T \cdot Y^\Omega$$

$$= \boldsymbol{U}_k \cdot (\boldsymbol{U}_k^\Omega)^T \cdot Y^\Omega. \tag{11}$$

*Similarly,*

$$\widehat{Y}^{HSVT,k} = \boldsymbol{Z}^{HSVT,k} \cdot \beta^{HSVT,k} = \boldsymbol{Z}^{HSVT,k} \cdot \left(\boldsymbol{Z}^{HSVT,k,\Omega}\right)^\dagger \cdot Y^\Omega$$

$$
\begin{aligned}
&= \boldsymbol{U}_k \cdot \boldsymbol{S}_k \cdot \boldsymbol{V}_k^T \cdot \left( (\boldsymbol{U}_k \cdot \boldsymbol{S}_k \cdot \boldsymbol{V}_k^T)^\Omega \right)^\dagger \cdot Y^\Omega \\
&= \boldsymbol{U}_k \cdot \boldsymbol{S}_k \cdot \boldsymbol{V}_k^T \cdot \left( \boldsymbol{U}_k^\Omega \cdot \boldsymbol{S}_k \cdot \boldsymbol{V}_k^T \right)^\dagger \cdot Y^\Omega \\
&= \boldsymbol{U}_k \cdot \boldsymbol{S}_k \cdot \boldsymbol{V}_k^T \cdot \boldsymbol{V}_k \cdot \boldsymbol{S}_k^{-1} \cdot (\boldsymbol{U}_k^\Omega)^\dagger \cdot Y^\Omega \\
&= \boldsymbol{U}_k \cdot (\boldsymbol{U}_k^\Omega)^T \cdot Y^\Omega.
\end{aligned}
\tag{12}
$$

*From (11) and (12), we obtain $\widehat{Y}^{PCR,k} = \widehat{Y}^{HSVT,k}$ for any $k \le N$.*

# F Proof of Theorem 5.1

## F.1 Background

Recall that the $(a,b)$-mixed norm of a matrix $\boldsymbol{B} \in \mathbb{R}^{N \times p}$ is defined as

$$
\|\boldsymbol{B}\|_{a,b} = \left( \sum_{j=1}^{p} \|\boldsymbol{B}_{\cdot,j}\|_a^b \right)^{1/b} = \left( \sum_{j=1}^{p} \left( \sum_{i=1}^{N} \boldsymbol{B}_{ij}^a \right)^{b/a} \right)^{1/b}.
$$

We are interested in the $(2, \infty)$-mixed norm, which corresponds to the maximum $\ell_2$ column norm:

$$
\|\boldsymbol{B}\|_{2,\infty} = \max_{j \in [p]} \|\boldsymbol{B}_{\cdot,j}\|_2 = \max_{j \in [p]} \left( \sum_{i=1}^{N} \boldsymbol{B}_{ij}^2 \right)^{1/2}.
$$

**Lemma F.1** *Let $\boldsymbol{B}$ be a real-valued $n \times p$ matrix and $x$ a real-valued $p$ dimensional vector. Let $q_1, q_2 \in [1, \infty]$ with $1/q_1 + 1/q_2 = 1$. Then,*

$$
\|\boldsymbol{B}x\|_2 \le \|x\|_{q_1} \|\boldsymbol{B}\|_{2,q_2}.
$$

**Proof F.1** *Using Hölder's Inequality, we have*

$$
\|\boldsymbol{B}x\|_2^2 = \sum_{i=1}^{n} \langle \boldsymbol{B}_{i,\cdot}, x \rangle^2 \le \|x\|_{q_1}^2 \sum_{i=1}^{n} \|\boldsymbol{B}_{i,\cdot}\|_{q_2}^2 = \|x\|_{q_1}^2 \cdot \|\boldsymbol{B}\|_{2,q_2}^2.
$$

## F.2 Proof of Theorem 5.1

**Proof F.2** *For simplicity of notation, let us define $\widehat{\boldsymbol{A}} = \boldsymbol{Z}^{HSVT,k}$, $\widehat{\boldsymbol{A}}^\Omega = \boldsymbol{Z}^{HSVT,k,\Omega}$. Due to equivalence relationship of Proposition 3.1 between PCR and performing linear regression using $\widehat{\boldsymbol{A}}^\Omega$, in the remainder of the proof we shall focus on linear regression using $\widehat{\boldsymbol{A}}^\Omega$. Per notation of Section 3.2, let $\beta^{HSVT,k}$ be the solution of linear regression using $\widehat{\boldsymbol{A}}^\Omega$ and predicted response variables $\widehat{Y}^{HSVT,k} = \boldsymbol{Z}^{HSVT,k}\beta^{HSVT,k}$; for simplicity, we will denote $\widehat{\beta} = \beta^{HSVT,k}$ and $\widehat{Y} = \widehat{Y}^{HSVT,k} = \widehat{\boldsymbol{A}}\widehat{\beta}$. Recall, per our model specification in (1), $Y^\Omega = \boldsymbol{A}^\Omega \beta^* + \phi + \epsilon$. Now observe*

$$
\left\| \widehat{\boldsymbol{A}}^\Omega \widehat{\beta} - Y^\Omega \right\|_2^2 = \left\| \widehat{\boldsymbol{A}}^\Omega \widehat{\beta} - \boldsymbol{A}^\Omega \beta^* + \phi \right\|_2^2 + \|\epsilon\|_2^2 - 2\epsilon^T (\widehat{\boldsymbol{A}}^\Omega \widehat{\beta} - \boldsymbol{A}^\Omega \beta^*) - 2\epsilon^T \phi.
\tag{13}
$$

*On the other hand, the optimality of $\widehat{\beta}$ (recall that $\widehat{\beta} \in \arg\min \|\widehat{\boldsymbol{A}}^\Omega \widehat{\beta} - Y^\Omega\|_2^2$) yields*

$$
\begin{aligned}
\left\| \widehat{\boldsymbol{A}}^\Omega \widehat{\beta} - Y^\Omega \right\|_2^2 &\le \left\| \widehat{\boldsymbol{A}}^\Omega \beta^* - Y^\Omega \right\|_2^2 \\
&= \left\| (\widehat{\boldsymbol{A}}^\Omega - \boldsymbol{A}^\Omega)\beta^* + \phi \right\|_2^2 + \|\epsilon\|_2^2 - 2\epsilon^T (\widehat{\boldsymbol{A}}^\Omega - \boldsymbol{A}^\Omega)\beta^* - 2\epsilon^T \phi.
\end{aligned}
\tag{14}
$$

*Combining (13) and (14) and taking expectations, we have*

$$
\mathbb{E} \left\| \widehat{\boldsymbol{A}}^\Omega \widehat{\beta} - \boldsymbol{A}^\Omega \beta^* + \phi \right\|_2^2 \le \mathbb{E} \left\| (\widehat{\boldsymbol{A}}^\Omega - \boldsymbol{A}^\Omega)\beta^* + \phi \right\|_2^2 + 2\mathbb{E}[\epsilon^T \widehat{\boldsymbol{A}}^\Omega (\widehat{\beta} - \beta^*)].
\tag{15}
$$

*Let us bound the final term on the right hand side of* (15). *Under our independence assumptions ($\epsilon$ is independent of $\boldsymbol{H}$), observe that*

$$\mathbb{E}[\epsilon^T \widehat{\boldsymbol{A}}^\Omega]\beta^* = \mathbb{E}[\epsilon^T]\mathbb{E}[\widehat{\boldsymbol{A}}^\Omega]\beta^* = 0.$$

*Recall that $\widehat{\beta} = (\widehat{\boldsymbol{A}}^\Omega)^\dagger Y = (\widehat{\boldsymbol{A}}^\Omega)^\dagger \boldsymbol{A}^\Omega \beta^* + (\widehat{\boldsymbol{A}}^\Omega)^\dagger \epsilon$. Using the cyclic and linearity properties of the trace operator (coupled with similar independence arguments), we further have*

$$
\begin{aligned}
\mathbb{E}[\epsilon^T \widehat{\boldsymbol{A}}^\Omega \widehat{\beta}] &= \mathbb{E}[\epsilon^T \widehat{\boldsymbol{A}}^\Omega (\widehat{\boldsymbol{A}}^\Omega)^\dagger]\boldsymbol{A}^\Omega \beta^* + \mathbb{E}[\epsilon^T \widehat{\boldsymbol{A}}^\Omega (\widehat{\boldsymbol{A}}^\Omega)^\dagger \epsilon] \\
&= \mathbb{E}[\epsilon]^T \mathbb{E}[\widehat{\boldsymbol{A}}^\Omega (\widehat{\boldsymbol{A}}^\Omega)^\dagger]\boldsymbol{A}^\Omega \beta^* + \mathbb{E}\Big[tr\Big(\epsilon^T \widehat{\boldsymbol{A}}^\Omega (\widehat{\boldsymbol{A}}^\Omega)^\dagger \epsilon\Big)\Big] \\
&= \mathbb{E}\Big[tr\Big(\widehat{\boldsymbol{A}}^\Omega (\widehat{\boldsymbol{A}}^\Omega)^\dagger \epsilon \epsilon^T\Big)\Big] = tr\Big(\mathbb{E}[\widehat{\boldsymbol{A}}^\Omega (\widehat{\boldsymbol{A}}^\Omega)^\dagger] \cdot \mathbb{E}[\epsilon\epsilon^T]\Big) \leq \sigma^2 \mathbb{E}\Big[tr\Big(\widehat{\boldsymbol{A}}^\Omega (\widehat{\boldsymbol{A}}^\Omega)^\dagger\Big)\Big] \\
&= \sigma^2 \mathbb{E}[rank(\widehat{\boldsymbol{A}}^\Omega)] \ \leq \ \sigma^2 k, \tag{16}
\end{aligned}
$$

*where the inequality follows from Property 2.3 and the fact that rank of $\widehat{\boldsymbol{A}}^\Omega$ is at most that of $\widehat{\boldsymbol{A}}^\Omega = \boldsymbol{Z}^{HSVT,k}$ and which by definition at most $k$. Consider*

$$\left\|\widehat{\boldsymbol{A}}^\Omega \widehat{\beta} - \boldsymbol{A}^\Omega \beta^* + \phi\right\|_2^2 = \left\|\widehat{\boldsymbol{A}}^\Omega \widehat{\beta} - \boldsymbol{A}^\Omega \beta^*\right\|_2^2 + \|\phi\|_2^2 + 2\phi^T(\widehat{\boldsymbol{A}}^\Omega \widehat{\beta} - \boldsymbol{A}^\Omega \beta^*). \tag{17}$$

*and*

$$\left\|(\widehat{\boldsymbol{A}}^\Omega - \boldsymbol{A}^\Omega)\beta^* + \phi\right\|_2^2 = \left\|(\widehat{\boldsymbol{A}}^\Omega - \boldsymbol{A}^\Omega)\beta^*\right\|_2^2 + \|\phi\|_2^2 + 2\phi^T((\widehat{\boldsymbol{A}}^\Omega - \boldsymbol{A}^\Omega)\beta^*). \tag{18}$$

*From* (16), (17) *and* (18), *the* (15) *becomes*

$$
\begin{aligned}
\mathbb{E}\left\|\widehat{\boldsymbol{A}}^\Omega \widehat{\beta} - \boldsymbol{A}^\Omega \beta^*\right\|_2^2 \leq \ &\mathbb{E}\left\|(\widehat{\boldsymbol{A}}^\Omega - \boldsymbol{A}^\Omega)\beta^*\right\|_2^2 + 2\sigma^2 k \\
&+ 2\mathbb{E}|\phi^T(\widehat{\boldsymbol{A}}^\Omega \widehat{\beta} - \boldsymbol{A}^\Omega \beta^*)| + 2\mathbb{E}|\phi^T((\widehat{\boldsymbol{A}}^\Omega - \boldsymbol{A}^\Omega)\beta^*)|. \tag{19}
\end{aligned}
$$

*Now*

$$|\phi^T(\widehat{\boldsymbol{A}}^\Omega \widehat{\beta} - \boldsymbol{A}^\Omega \beta^*)| \leq \|\phi\|_\infty \|\widehat{\boldsymbol{A}}^\Omega \widehat{\beta} - \boldsymbol{A}^\Omega \beta^*\|_1, \tag{20}$$

$$|\phi^T((\widehat{\boldsymbol{A}}^\Omega - \boldsymbol{A}^\Omega)\beta^*)| \leq \|\phi\|_2 \|(\widehat{\boldsymbol{A}}^\Omega - \boldsymbol{A}^\Omega)\beta^*\|_2 \ \leq \ \sqrt{n}\|\phi\|_\infty \|(\widehat{\boldsymbol{A}}^\Omega - \boldsymbol{A}^\Omega)\beta^*\|_2. \tag{21}$$

*From Lemma F.1 with $q_1 = 1$ and $q_2 = \infty$ to obtain*

$$\left\|(\widehat{\boldsymbol{A}}^\Omega - \boldsymbol{A}^\Omega)\beta^*\right\|_2^2 \leq \|\beta^*\|_1^2 \cdot \max_{j\in[p]}\left\|(\boldsymbol{A}^\Omega - \widehat{\boldsymbol{A}}^\Omega)_{\cdot, j}\right\|_2^2 \ = \ \|\beta^*\|_1^2 \|\boldsymbol{A}^\Omega - \widehat{\boldsymbol{A}}^\Omega\|_{2,\infty}^2. \tag{22}$$

*Using* (20), (21) *and* (22) *in* (19), *we obtain*

$$
\begin{aligned}
\mathbb{E}\left\|\widehat{\boldsymbol{A}}^\Omega \widehat{\beta} - \boldsymbol{A}^\Omega \beta^*\right\|_2^2 \leq \ &2\sigma^2 k + \|\beta^*\|_1^2 \mathbb{E}\|\boldsymbol{A}^\Omega - \widehat{\boldsymbol{A}}^\Omega\|_{2,\infty}^2 + 2\sqrt{n}\|\phi\|_\infty \|\beta^*\|_1 \mathbb{E}\|\boldsymbol{A}^\Omega - \widehat{\boldsymbol{A}}^\Omega\|_{2,\infty} \\
&+ 2\|\phi\|_\infty \mathbb{E}\left\|\widehat{\boldsymbol{A}}^\Omega \widehat{\beta} - \boldsymbol{A}^\Omega \beta^*\right\|_1
\end{aligned}
$$

*Dividing by $1/n$ on both sides, using Jensen's inequality and fact that $\|v\|_1 \leq \sqrt{n}\|v\|_2$ for all $v \in \mathbb{R}^n$, we obtain*

$$
\begin{aligned}
\frac{1}{n}\mathbb{E}\left\|\widehat{\boldsymbol{A}}^\Omega \widehat{\beta} - \boldsymbol{A}^\Omega \beta^*\right\|_2^2 \leq \ &\frac{2\sigma^2 k}{n} + \frac{\|\beta^*\|_1^2 \mathbb{E}\|\boldsymbol{A}^\Omega - \widehat{\boldsymbol{A}}^\Omega\|_{2,\infty}^2}{n} + 2\|\phi\|_\infty \sqrt{\frac{\|\beta^*\|_1^2 \mathbb{E}\|\boldsymbol{A}^\Omega - \widehat{\boldsymbol{A}}^\Omega\|_{2,\infty}^2}{n}} \\
&+ 2\|\phi\|_\infty \sqrt{\frac{1}{n}\mathbb{E}\left\|\widehat{\boldsymbol{A}}^\Omega \widehat{\beta} - \boldsymbol{A}^\Omega \beta^*\right\|_2^2}. \tag{23}
\end{aligned}
$$

*Let*

$$x = \frac{1}{n}\mathbb{E}\left\|\widehat{\boldsymbol{A}}^\Omega \widehat{\beta} - \boldsymbol{A}^\Omega \beta^*\right\|_2^2, \quad y = \frac{2\sigma^2 k}{n} + \frac{\|\beta^*\|_1^2 \mathbb{E}\|\boldsymbol{A}^\Omega - \widehat{\boldsymbol{A}}^\Omega\|_{2,\infty}^2}{n} + 2\|\phi\|_\infty \sqrt{\frac{\|\beta^*\|_1^2 \mathbb{E}\|\boldsymbol{A}^\Omega - \widehat{\boldsymbol{A}}^\Omega\|_{2,\infty}^2}{n}}.$$

Then (23) can be viewed as $x \leq y + 2\|\phi\|_\infty \sqrt{x}$ with both $x, y \geq 0$. Therefore, either $x \leq 4\|\phi\|_\infty \sqrt{x}$ or $x \leq 2y$. That is, $x \leq 16\|\phi\|_\infty^2$ or $x \leq 2y$. That is, $x \leq 2y + 16\|\phi\|_\infty^2$. Replacing values of $x, y$ as above we obtain

$$\frac{1}{n}\mathbb{E}\left\|\widehat{\boldsymbol{A}}^\Omega\widehat{\beta} - \boldsymbol{A}^\Omega\beta^*\right\|_2^2 \leq \frac{4\sigma^2 k}{n} + \frac{2\|\beta^*\|_1^2\mathbb{E}\|\boldsymbol{A}^\Omega - \widehat{\boldsymbol{A}}^\Omega\|_{2,\infty}^2}{n} + 4\|\phi\|_\infty\sqrt{\frac{\|\beta^*\|_1^2\mathbb{E}\|\boldsymbol{A}^\Omega - \widehat{\boldsymbol{A}}^\Omega\|_{2,\infty}^2}{n}} + 16\|\phi\|_\infty^2$$

$$\leq \frac{4\sigma^2 k}{n} + \frac{3\|\beta^*\|_1^2\mathbb{E}\|\boldsymbol{A}^\Omega - \widehat{\boldsymbol{A}}^\Omega\|_{2,\infty}^2}{n} + 20\|\phi\|_\infty^2,$$

where in the last inequality we have used the fact that for any $x, y \in \mathbb{R}$, $2xy \leq x^2 + y^2$.

# G   Bound on Spectral Norm of Random Matrix

Here we state and derive bound on the spectral norm of random matrix whose rows (or columns) are generated independently per $\psi_\alpha$-distribution for $\alpha \geq 1$. The bounds we shall state and derive (Theorem G.1) are not the sharpest possible. But they are sufficient for our purposes. Sharp bound for $\alpha = 1$ and $\alpha \geq 2$ can be found in [6] and [51] respectively. We provide the proof here for completeness as well as ease of exposition.

## G.1   Outline

We begin by presenting Proposition G.1, which holds for general random matrices $\boldsymbol{W} \in \mathbb{R}^{N \times p}$. We note that this result depends on two quantities: (1) $\|\mathbb{E}\boldsymbol{W}^T\boldsymbol{W}\|$ and (2) $\|\boldsymbol{W}_{i,\cdot}\|_{\psi_\alpha}$ for all $i \in [N]$. We then instantiate $\boldsymbol{W} := \boldsymbol{Z} - \rho\boldsymbol{A}$ and present Lemmas G.1 and G.2, which bound (1) and (2), respectively, for our choice of $\boldsymbol{W}$. We state and prove Theorem G.1 that will be crucial in establishing properties of HSVT. The proofs of various results stated on the way will follow near the end of this section.

**Proposition G.1** *Let $\boldsymbol{W} \in \mathbb{R}^{N \times p}$ be a random matrix whose rows $\boldsymbol{W}_{i,\cdot}$ ($i \in [N]$) are independent $\psi_\alpha$-random vectors for some $\alpha \geq 1$. Then for any $\delta_1 > 0$,*

$$\|\boldsymbol{W}\| \leq \left\|\mathbb{E}\boldsymbol{W}^T\boldsymbol{W}\right\|^{1/2} + C(\alpha)\sqrt{(1+\delta_1)p}\max_{i\in[N]}\|\boldsymbol{W}_{i,\cdot}\|_{\psi_\alpha}\left(1 + (2+\delta_1)\log(Np)\right)^{\frac{1}{\alpha}}\sqrt{\log(Np)}$$

*with probability at least $1 - \frac{2}{N^{1+\delta_1}p^{\delta_1}}$. Here, $C(\alpha) > 0$ is an absolute constant that depends only on $\alpha$.*

**Lemma G.1** *Assume Property 2.4 holds. Then,*
$$\left\|\mathbb{E}(\boldsymbol{Z} - \rho\boldsymbol{A})^T(\boldsymbol{Z} - \rho\boldsymbol{A})\right\| \leq \rho(1-\rho)\max_{j\in[p]}\|\boldsymbol{A}_{\cdot,j}\|_2^2 + \rho^2\|\mathbb{E}\boldsymbol{H}^T\boldsymbol{H}\|.$$

**Lemma G.2** *Assume Properties 2.1, 2.5, and 2.4 hold. Then for any $\alpha \geq 1$ with which Property 2.5 holds, we have*
$$\|\boldsymbol{Z}_{i,\cdot} - \rho\boldsymbol{A}_{i,\cdot}\|_{\psi_\alpha} \leq C(K_\alpha + \Gamma) \qquad \text{for all } i \in [N],$$
*where $C > 0$ is an absolute constant.*

## G.2   Key Result: Theorem G.1

Now we state the main result.

**Theorem G.1** *Suppose Properties 2.1, 2.5 for some $\alpha \geq 1$ and 2.4 hold. Then for any $\delta_1 > 0$,*
$$\|\boldsymbol{Z} - \rho\boldsymbol{A}\| \leq \sqrt{N\rho}\sqrt{\rho\gamma^2 + (1-\rho)\Gamma^2}$$
$$+ C(\alpha)\sqrt{1+\delta_1}\sqrt{p}(K_\alpha + \Gamma)\left(1 + (2+\delta_1)\log(Np)\right)^{\frac{1}{\alpha}}\sqrt{\log(Np)}$$

*with probability at least $1 - \frac{2}{N^{1+\delta_1}p^{\delta_1}}$. Here, $C(\alpha)$ is an absolute constant that depends only on $\alpha$.*

**Proof G.1** *The proof follows by plugging the results of Lemmas G.1 and G.2 into Proposition G.1 for $\boldsymbol{W} := \boldsymbol{Z} - \rho\boldsymbol{A}$ and applying Properties 2.1 and 2.5.*

### G.3 Proof of Proposition G.1

**Proof G.2** *We prove the proposition in four steps.*

**Step 1: picking the threshold value.** *Let $e_1, \ldots, e_p \in \mathbb{R}^p$ denote the canonical basis[3] of $\mathbb{R}^p$. Observe that $\|\boldsymbol{W}_{i,\cdot}\|_2^2 = \boldsymbol{W}_{i,\cdot} \boldsymbol{W}_{i,\cdot}^T = \sum_{j=1}^p \left(\boldsymbol{W}_{i,\cdot} e_j\right)^2$ [4]. Therefore, for any $t \geq 0$,*

$$
\mathbb{P}\left\{ \|\boldsymbol{W}_{i,\cdot}\|_2^2 > t \right\} = \mathbb{P}\left\{ \sum_{j=1}^p \left(\boldsymbol{W}_{i,\cdot} e_j\right)^2 > t \right\}
$$

$$
\overset{(a)}{\leq} \sum_{j=1}^p \mathbb{P}\left\{ \left(\boldsymbol{W}_{i,\cdot} e_j\right)^2 > \frac{t}{p} \right\}
$$

$$
\leq \sum_{j=1}^p \mathbb{P}\left\{ |\boldsymbol{W}_{i,\cdot} e_j| > \sqrt{\frac{t}{p}} \right\}
$$

$$
\overset{(b)}{\leq} 2p \exp\left( -C(\alpha) \left( \frac{t}{p \|\boldsymbol{W}_{i,\cdot}\|_{\psi_\alpha}^2} \right)^{\frac{\alpha}{2}} \right),
$$

*where (a) uses the union bound and (b) follows from the definition of $\psi_\alpha$-random vector ($C(\alpha)$ is an absolute constant which depends only on $\alpha \geq 1$). Choosing $t = C^{\frac{2}{\alpha}} C(\alpha)^{-\frac{2}{\alpha}} p \|\boldsymbol{W}_{i,\cdot}\|_{\psi_\alpha}^2 \left( \log(2p) \right)^{\frac{2}{\alpha}}$ for some $C > 1$ gives*

$$
\mathbb{P}\left\{ \|\boldsymbol{W}_{i,\cdot}\|_2^2 > C^{\frac{2}{\alpha}} C(\alpha)^{-\frac{2}{\alpha}} p \|\boldsymbol{W}_{i,\cdot}\|_{\psi_\alpha}^2 \left( \log(2p) \right)^{\frac{2}{\alpha}} \right\} \leq \left( \frac{1}{2p} \right)^{C-1}.
$$

*Applying the union bound, we obtain*

$$
\mathbb{P}\left\{ \max_{i \in [N]} \|\boldsymbol{W}_{i,\cdot}\|_2^2 > C^{\frac{2}{\alpha}} C(\alpha)^{-\frac{2}{\alpha}} p \max_{i \in [N]} \|\boldsymbol{W}_{i,\cdot}\|_{\psi_\alpha}^2 \left( \log(2p) \right)^{\frac{2}{\alpha}} \right\} \leq N \left( \frac{1}{2p} \right)^{C-1}.
$$

*For $\delta_1 > 0$, we define $C(\delta_1) \triangleq 1 + (2 + \delta_1) \log_{2p}(Np)$ and let $C = C(\delta_1)$. Also, we define*

$$
t_0(\delta_1) \triangleq C(\delta_1)^{\frac{2}{\alpha}} C(\alpha)^{-\frac{2}{\alpha}} p \max_{i \in [N]} \|\boldsymbol{W}_{i,\cdot}\|_{\psi_\alpha}^2 \left( \log(2p) \right)^{\frac{2}{\alpha}}.
$$

*We have*

$$
\mathbb{P}\left\{ \max_{i \in [N]} \|\boldsymbol{W}_{i,\cdot}\|_2^2 > t_0(\delta_1) \right\} \leq N \left( \frac{1}{2p} \right)^{(2+\delta_1) \log_{2p}(Np)} = \frac{1}{N^{1+\delta_1} p^{2+\delta_1}}. \tag{24}
$$

**Step 2: decomposing $W$ by truncation.** *Next, given $\delta_1 > 0$, we decompose the random matrix $\boldsymbol{W}$ as follows:*

$$
\boldsymbol{W} = \boldsymbol{W}^\circ(\delta_1) + \boldsymbol{W}^\times(\delta_1)
$$

*where for each $i \in [N]$,*

$$
\boldsymbol{W}^\circ(\delta_1)_{i,\cdot} = \boldsymbol{W}_{i,\cdot} \mathbb{1}\left\{ \|\boldsymbol{W}_{i,\cdot}\|_2^2 \leq t_0(\delta_1) \right\} \quad \text{and} \quad \boldsymbol{W}^\times(\delta_1)_{i,\cdot} = \boldsymbol{W}_{i,\cdot} \mathbb{1}\left\{ \|\boldsymbol{W}_{i,\cdot}\|_2^2 > t_0(\delta_1) \right\}.
$$

*Then it follows that*

$$
\|\boldsymbol{W}\| \leq \|\boldsymbol{W}^\circ(\delta_1)\| + \left\|\boldsymbol{W}^\times(\delta_1)\right\| \leq \|\boldsymbol{W}^\circ(\delta_1)\| + \left\|\boldsymbol{W}^\times(\delta_1)\right\|_F. \tag{25}
$$

**Step 3: bounding $\|\boldsymbol{W}^{\circ}(\delta_1)\|$ and $\|\boldsymbol{W}^{\times}(\delta_1)\|_F$.** *We define two events for conditioning:*

$$E_1(\delta_1) := \left\{ \|\boldsymbol{W}^{\circ}(\delta_1)\| \le \|\mathbb{E}\boldsymbol{W}^T\boldsymbol{W}\|^{1/2} + \sqrt{\frac{1+\delta_1}{c}t_0(\delta_1)\log(Np)} \right\}, \tag{26}$$

$$E_2(\delta_1) := \left\{ \|\boldsymbol{W}^{\times}(\delta_1)\|_F = 0 \right\}. \tag{27}$$

*First, given $\delta_1 > 0$, we let $\Sigma^{\circ}(\delta_1) = \mathbb{E}\boldsymbol{W}^{\circ}(\delta_1)^T\boldsymbol{W}^{\circ}(\delta_1)$. By definition of $\boldsymbol{W}^{\circ}(\delta_1)$, we have $\|\boldsymbol{W}_{i,\cdot}\|_2 \le \sqrt{t_0(\delta_1)}$ for all $i \in [N]$. Then it follows that for every $s \ge 0$,*

$$\|\boldsymbol{W}^{\circ}(\delta_1)\| \le \|\Sigma^{\circ}(\delta_1)\|^{1/2} + s\sqrt{t_0(\delta_1)}$$

*with probability at least $1 - p\exp(-cs^2)$ (see Theorem 5.44 of [51] and Eqs. (5.32) and (5.33) in reference, and replacing the common second moment $\Sigma = \mathbb{E}\boldsymbol{W}_{i,\cdot}^T\boldsymbol{W}_{i,\cdot}$ with the average second moment for all rows, $\Sigma = \frac{1}{N}\sum_{i=1}^N \mathbb{E}\boldsymbol{W}_{i,\cdot}^T\boldsymbol{W}_{i,\cdot}$, i.e., redefining $\Sigma$). Note that $\|\Sigma^{\circ}(\delta_1)\| = \|\mathbb{E}\boldsymbol{W}^{\circ}(\delta_1)^T\boldsymbol{W}^{\circ}(\delta_1)\| \le \|\mathbb{E}\boldsymbol{W}^T\boldsymbol{W}\|$. Now we define $\tilde{E}_1(s)$ parameterized by $s > 0$ as*

$$\tilde{E}_1(s; \delta_1) := \left\{ \|\boldsymbol{W}^{\circ}(\delta_1)\| > \|\mathbb{E}\boldsymbol{W}^T\boldsymbol{W}\|^{1/2} + s\sqrt{t_0(\delta_1)} \right\}.$$

*If we pick $s = \left(\frac{1+\delta_1}{c}\log(Np)\right)^{1/2}$, then $E_1(\delta_1) = \tilde{E}_1(s; \delta_1)$ and*

$$\mathbb{P}\left(E_1(\delta_1)^c\right) \le p\exp(-cs^2) = p\exp\left(-(1+\delta_1)\log(Np)\right) = \frac{1}{N^{1+\delta_1}p^{\delta_1}}.$$

*Next, we observe that $\|\boldsymbol{W}^{\times}(\delta_1)\|_F = 0$ if and only if $\boldsymbol{W}^{\times}(\delta_1) = 0$. If $\boldsymbol{W}^{\times}(\delta_1) \ne 0$, then $\max_{i \in [n]}\|\boldsymbol{W}_{i,\cdot}\|_2^2 > t_0(\delta_1)$. Therefore,*

$$\mathbb{P}\left(E_2^c\right) \le \frac{1}{N^{1+\delta_1}p^{2+\delta_1}}$$

*by the analysis in Step 1; see (24).*

**Step 4: concluding the proof.** *For any given $\delta_1 > 0$,*

$$\mathbb{P}\left( \|\boldsymbol{W}\| > \|\mathbb{E}\boldsymbol{W}^T\boldsymbol{W}\|^{1/2} + \sqrt{\frac{1+\delta_1}{c}t_0(\delta_1)\log(Np)} \,\middle|\, E_1(\delta_1) \cap E_2(\delta_1) \right) = 0.$$

*by (25), (26), and (27). By the law of total probability and the union bound,*

$$\mathbb{P}\left( \|\boldsymbol{W}\| > \|\mathbb{E}\boldsymbol{W}^T\boldsymbol{W}\|^{1/2} + \sqrt{\frac{1+\delta_1}{c}t_0(\delta_1)\log(Np)} \right)$$

$$\le \mathbb{P}\left( \|\boldsymbol{W}\| > \|\mathbb{E}\boldsymbol{W}^T\boldsymbol{W}\|^{1/2} + \sqrt{\frac{1+\delta_1}{c}t_0(\delta_1)\log(Np)} \,\middle|\, E_1(\delta_1) \cap E_2(\delta_1) \right)$$

$$+ \mathbb{P}\left(E_1(\delta)^c\right) + \mathbb{P}\left(E_2(\delta)^c\right)$$

$$\le \frac{1}{N^{1+\delta_1}p^{\delta_1}} + \frac{1}{N^{1+\delta_1}p^{2+\delta_1}}$$

$$\le \frac{2}{N^{1+\delta_1}p^{\delta_1}}.$$

*This completes the proof.*

### G.4 Proof of Lemma G.1

**Proof G.3** *When Property 2.4 holds, then*

$$\mathbb{E}(\boldsymbol{Z} - \rho\boldsymbol{A})^T(\boldsymbol{Z} - \rho\boldsymbol{A}) = \rho(1-\rho)diag(\boldsymbol{A}^T\boldsymbol{A}) + \rho^2\mathbb{E}(\boldsymbol{X} - \boldsymbol{A})^T(\boldsymbol{X} - \boldsymbol{A})$$

*by [48, Lemma A.2]. Applying triangle inequality, we have*

$$\|\mathbb{E}(\boldsymbol{Z} - \rho\boldsymbol{A})^T(\boldsymbol{Z} - \rho\boldsymbol{A})\| \le \rho(1-\rho)\|diag(\boldsymbol{A}^T\boldsymbol{A})\| + \rho^2\|\mathbb{E}(\boldsymbol{X} - \boldsymbol{A})^T(\boldsymbol{X} - \boldsymbol{A})\|$$

$$\le \rho(1-\rho)\max_{j \in [p]}\|\boldsymbol{A}_{\cdot,j}\|_2^2 + \rho^2\|\mathbb{E}\boldsymbol{H}^T\boldsymbol{H}\|.$$

### G.5 Proof of Lemma G.2

#### G.5.1 Auxiliary Lemmas

**Lemma G.3** *Suppose that $X \in \mathbb{R}^n$ and $P \in \{0,1\}^n$ are random vectors. Then for any $\alpha \geq 1$,*

$$\|X \circ P\|_{\psi_\alpha} \leq \|X\|_{\psi_\alpha}.$$

**Proof G.4** *Given a deterministic binary vector $P_0 \in \{0,1\}^n$, let $I_{P_0} = \{i \in [n] : Q_i = 1\}$. Observe that*

$$X \circ P_0 = \sum_{i \in I_{P_0}} e_i e_i^T X.$$

*Here, $\circ$ denotes the Hadamard product (entrywise product) of two matrices. By definition of the $\psi_\alpha$-norm,*

$$\|X\|_{\psi_\alpha} = \sup_{u \in \mathbb{S}^{n-1}} \|u^T X\|_{\psi_\alpha} = \sup_{u \in \mathbb{S}^{n-1}} \inf \left\{ t > 0 : \mathbb{E}_X \left[ \exp\left(|u^T X|^\alpha / t^\alpha\right) \right] \leq 2 \right\}.$$

*Let $u_0 \in \mathbb{S}^{n-1}$ denote the maximum-achieving unit vector (such $u_0$ exists because $\inf\{\cdots\}$ is continuous with respect to $u$ and $\mathbb{S}^{n-1}$ is compact). Then,*

$$
\begin{aligned}
\|X \circ P\|_{\psi_\alpha} &= \sup_{u \in \mathbb{S}^{n-1}} \|u^T X \circ P\|_{\psi_\alpha} \\
&= \sup_{u \in \mathbb{S}^{n-1}} \inf \left\{ t > 0 : \mathbb{E}_{X,P} \left[ \exp\left(|u^T X \circ P|^\alpha / t^\alpha\right) \right] \leq 2 \right\} \\
&= \sup_{u \in \mathbb{S}^{n-1}} \inf \left\{ t > 0 : \mathbb{E}_P \left[ \mathbb{E}_X \left[ \exp\left(|u^T X \circ P|^\alpha / t^\alpha\right) \,\middle|\, P \right] \right] \leq 2 \right\} \\
&= \sup_{u \in \mathbb{S}^{n-1}} \inf \left\{ t > 0 : \mathbb{E}_P \left[ \mathbb{E}_X \left[ \exp\left(\left|u^T \sum_{i \in I_P} e_i e_i^T X\right|^\alpha / t^\alpha\right) \,\middle|\, P \right] \right] \leq 2 \right\} \\
&= \sup_{u \in \mathbb{S}^{n-1}} \inf \left\{ t > 0 : \mathbb{E}_P \left[ \mathbb{E}_X \left[ \exp\left(\left|\left(\sum_{i \in I_P} e_i e_i^T u\right)^T X\right|^\alpha / t^\alpha\right) \,\middle|\, P \right] \right] \leq 2 \right\}.
\end{aligned}
$$

*For any $u \in \mathbb{S}^{n-1}$ and $P_0 \in \{0,1\}^n$, observe that*

$$\mathbb{E}_X \left[ \exp\left(\left|\left(\sum_{i \in I_P} e_i e_i^T u\right)^T X\right|^\alpha / t^\alpha\right) \,\middle|\, P = P_0 \right] \leq \mathbb{E}_X \left[ \exp\left(|u_0^T X|^\alpha / t^\alpha\right) \right].$$

*Therefore, taking supremum over $u \in \mathbb{S}^{n-1}$, we obtain*

$$\|X \circ P\|_{\psi_\alpha} \leq \|X\|_{\psi_\alpha}.$$

**Lemma G.4** *Let $X$ be a mean-zero, $\psi_\alpha$-random variable for some $\alpha \geq 1$. Then for $|\lambda| \leq \frac{1}{C\|X\|_{\psi_\alpha}}$,*

$$\mathbb{E} \exp(\lambda X) \leq \exp\left(C\lambda^2 \|X\|_{\psi_\alpha}^2\right).$$

**Proof G.5** *See [52], Section 2.7.*

**Lemma G.5** *Let $X_1, \ldots, X_n$ be independent random variables with mean zero. For $\alpha \geq 1$,*

$$\left\| \sum_{i=1}^n X_i \right\|_{\psi_\alpha} \leq C \left( \sum_{i=1}^n \|X_i\|_{\psi_\alpha}^2 \right)^{1/2}.$$

**Proof G.6** *Immediate by Lemma G.4.*

### G.5.2 Proof of Lemma G.2

**Proof G.7** *Let $\boldsymbol{P} \in \{0, 1\}^{N \times p}$ denote a random matrix whose entries are i.i.d. random variables that take value 1 with probability $\rho$ and 0 otherwise. Note that $\boldsymbol{Z}_{i,\cdot} = \boldsymbol{X}_{i,\cdot} \circ \boldsymbol{P}_{i,\cdot}$ when Property 2.4 is assumed and $\star$ is identified with 0. By triangle inequality,*

$$
\begin{aligned}
\|\boldsymbol{Z}_{i,\cdot} - \rho \boldsymbol{A}_{i,\cdot}\|_{\psi_\alpha} &= \|\boldsymbol{X}_{i,\cdot} \circ \boldsymbol{P}_{i,\cdot} - \rho \boldsymbol{A}_{i,\cdot}\|_{\psi_\alpha} \\
&= \|(\boldsymbol{X}_{i,\cdot} \circ \boldsymbol{P}_{i,\cdot}) - (\boldsymbol{A}_{i,\cdot} \circ \boldsymbol{P}_{i,\cdot}) - \rho \boldsymbol{A}_{i,\cdot} + (\boldsymbol{A}_{i,\cdot} \circ \boldsymbol{P}_{i,\cdot})\|_{\psi_\alpha} \\
&\leq \|(\boldsymbol{X}_{i,\cdot} - \boldsymbol{A}_{i,\cdot}) \circ \boldsymbol{P}_{i,\cdot}\|_{\psi_\alpha} + \|(\boldsymbol{A}_{i,\cdot} \circ \boldsymbol{P}_{i,\cdot}) - \rho \boldsymbol{A}_{i,\cdot}\|_{\psi_\alpha}.
\end{aligned}
$$

*By definition of $\boldsymbol{X}$, Property 2.5, and Lemma G.3, we have that*

$$
\|(\boldsymbol{X}_{i,\cdot} - \boldsymbol{A}_{i,\cdot}) \circ \boldsymbol{P}_{i,\cdot}\|_{\psi_\alpha} \leq \|\boldsymbol{X}_{i,\cdot} - \boldsymbol{A}_{i,\cdot}\|_{\psi_\alpha} = \|\eta_{i,\cdot}\|_{\psi_\alpha} \leq CK_\alpha.
$$

*Moreover, Property 2.1 and the i.i.d. property of $\boldsymbol{P}_{ij}$ for different $j$ gives*

$$
\begin{aligned}
\left\|(\boldsymbol{A}_{i,\cdot} \circ \boldsymbol{P}_{i,\cdot}) - \rho \boldsymbol{A}_{i,\cdot}\right\|_{\psi_\alpha} &= \sup_{u \in \mathbb{S}^{p-1}} \left\|\sum_{j=1}^{p} u_j \boldsymbol{A}_{i,j}(\boldsymbol{P}_{i,j} - \rho)\right\|_{\psi_\alpha} \\
&\leq \sup_{u \in \mathbb{S}^{p-1}} \left(\sum_{j=1}^{p} u_j^2 \|\boldsymbol{A}_{i,j}(\boldsymbol{P}_{i,j} - \rho)\|_{\psi_\alpha}^2\right)^{1/2} \\
&\leq \left(\sup_{u \in \mathbb{S}^{p-1}} \sum_j u_j^2 \max_{j \in [p]} |\boldsymbol{A}_{i,j}|^2\right)^{1/2} \|\boldsymbol{P}_{1,1} - \rho\|_{\psi_\alpha} \\
&\leq \Gamma \|\boldsymbol{P}_{1,1} - \rho\|_{\psi_\alpha}.
\end{aligned}
$$

*The first inequality follows from Lemma G.5, the second inequality is immediate, and the last inequality follows from Property 2.1. Lastly, $\|\boldsymbol{P}_{1,1} - \rho\|_{\psi_\alpha} \leq C$ because $\boldsymbol{P}_{1,1} - \rho$ is a bounded random variable in $[-\rho, 1 - \rho]$.*

## H  Proof of Lemma 5.1

### H.1  Outline

To bound the error in estimation of HSVT, $\boldsymbol{Z}^{HSVT,k}$ with thresholding at $k$th singular value, and underlying covariate matrix $\boldsymbol{A}$ with respect to $\|\cdot\|_{2,\infty}$ matrix norm, we shall start by presenting Lemma H.1 which bounds $\|\boldsymbol{Z}^{HSVT,k} - \boldsymbol{A}\|_{2,\infty}$ as a function of few quantities. Next, we bound these quantities with high probability in our setting through help of sequence of results including the spectral norm bound stated in Theorem G.1. We conclude the proof of Lemma 5.1 and subsequently proofs of helper results on our way.

**Notation.**  Consider a matrix $\boldsymbol{B} \in \mathbb{R}^{N \times p}$ such that $\boldsymbol{B} = \sum_{i=1}^{N \wedge p} \sigma_i(\boldsymbol{B}) x_i y_i^T$. With a specific choice of $\lambda \geq 0$, we can define a function $\varphi_\lambda^{\boldsymbol{B}} : \mathbb{R}^N \to \mathbb{R}^N$ as follows: for any vector $w \in \mathbb{R}^N$,

$$
\varphi_\lambda^{\boldsymbol{B}}(w) = \sum_{i=1}^{N \wedge p} \mathbb{1}(\sigma_i(\boldsymbol{B}) \geq \lambda) x_i x_i^T w. \tag{28}
$$

Note that $\varphi_\lambda^{\boldsymbol{B}}$ is a linear operator and it depends on the tuple $(\boldsymbol{B}, \lambda)$; more precisely, the singular values and the left singular vectors of $\boldsymbol{B}$, as well as the threshold $\lambda$. If $\lambda = 0$, then we will adopt the shorthand notation: $\varphi^{\boldsymbol{B}} = \varphi_0^{\boldsymbol{B}}$.

**Lemma H.1** *Suppose that*

1. *$\|\boldsymbol{Z} - \rho \boldsymbol{A}\| \leq \Delta$ for some $\Delta \geq 0$,*

2. *$\frac{1}{\varepsilon} \rho \leq \widehat{\rho} \leq \varepsilon \rho$ for some $\varepsilon \geq 1$.*

Let $\widehat{A} = Z^{HSVT,k}$, $A^k = HSVT_{\tau_k}(A)$ and $E = A - A^k$. Then for any $j \in [p]$,

$$\left\|\widehat{A}_{\cdot,j} - A_{\cdot,j}\right\|_2^2 \le \frac{4\varepsilon^2}{\rho^2} \frac{\Delta^2}{\rho^2(\tau_k - \tau_{k+1})^2} \|Z_{\cdot,j} - \rho A_{\cdot,j}\|_2^2$$
$$+ \frac{4\varepsilon^2}{\rho^2}\left\|\varphi^{A^k}(Z_{\cdot,j} - \rho A_{\cdot,j})\right\|_2^2 + 2(\varepsilon - 1)^2\|A_{\cdot,j}\|_2^2.$$
$$+ \frac{2\Delta^2}{\rho^2(\tau_k - \tau_{k+1})^2}\left\|A_{\cdot,j}^k\right\|_2^2 + 2\left\|E_{\cdot,j}\right\|_2^2.$$

**High probability events for conditioning.** We define the following four events:

$$\mathcal{E}_1 := \left\{\|Z - \rho A\| \le \sqrt{N\rho}\sqrt{\rho\gamma^2 + (1-\rho)\Gamma^2}\right.$$
$$\left. + 2C(\alpha)\sqrt{p}(K_\alpha + \Gamma)\left(1 + 9\log(Np)\right)^{\frac{1}{\alpha}}\sqrt{\log(Np)}\right\}$$

$$\mathcal{E}_2 := \left\{\left(1 - \sqrt{\frac{20\log(Np)}{Np\rho}}\right)\rho \le \widehat{\rho} \le \frac{1}{1 - \sqrt{\frac{20\log(Np)}{Np\rho}}}\rho\right\}$$

$$\mathcal{E}_3 := \left\{\max_{j\in[p]}\left\|Z_{\cdot,j} - \rho A_{\cdot,j}\right\|_2^2 \le 11C(K_\alpha + \Gamma)^2 N \log^{\frac{2}{\alpha}}(Np)\right\}$$

$$\mathcal{E}_4 := \left\{\max_{j\in[p]}\left\|\varphi^{A^k}(Z_{\cdot,j} - \rho A_{\cdot,j})\right\|_2^2 \le 11C(K_\alpha + \Gamma)^2 r \log^{\frac{2}{\alpha}}(Np)\right\}.$$

Here, $C(\alpha)$ is the same absolute constant that appears in Theorem G.1, and $C > 0$ is an absolute constant. The proof of Lemmas H.2, H.3, H.4, and H.5 can be found in Appendix H.5.

**Observation 1:** $\mathcal{E}_1$ **occurs with high probability.**

**Lemma H.2** *Suppose that Properties 2.1, 2.5 for $\alpha \ge 1$, and 2.4 hold. Then for any $\delta_1 > 0$,*

$$\|Z - \rho A\| \le \sqrt{N\rho}\sqrt{\rho\gamma^2 + (1-\rho)\Gamma^2}$$
$$+ C(\alpha)\sqrt{1 + \delta_1}\sqrt{p}(K_\alpha + \Gamma)\left(1 + (2 + \delta_1)\log(Np)\right)^{\frac{1}{\alpha}}\sqrt{\log(Np)}$$

*with probability at least $1 - \frac{2}{N^{1+\delta_1}p^{\delta_1}}$; $C(\alpha) > 0$ is an absolute constant that depends only on $\alpha$.*

**Remark H.1** *By letting $\delta_1 = 10$ in Lemma H.2, we have that $\mathbb{P}\left(\mathcal{E}_1^c\right) \le \frac{2}{N^{10}p^{10}}$.*

**Observation 2:** $\mathcal{E}_2$ **occurs with high probability.**

**Lemma H.3** *Suppose that Property 2.4 holds. Then for any $\varepsilon > 1$,*

$$\mathbb{P}\left(\frac{1}{\varepsilon}\rho \le \widehat{\rho} \le \varepsilon\rho\right) \ge 1 - 2\exp\left(-\frac{(\varepsilon - 1)^2}{2\varepsilon^2}Np\rho\right).$$

**Remark H.2** *Let $\varepsilon = \left(1 - \sqrt{\frac{20\log(Np)}{Np\rho}}\right)^{-1}$ in Lemma H.3. Then, $\mathbb{P}\left(\mathcal{E}_2^c\right) \le \frac{2}{N^{10}p^{10}}$.*

**Observation 3:** $\mathcal{E}_3$ **and** $\mathcal{E}_4$ **occur with high probability.**

**Lemma H.4** *Suppose Properties 2.1, 2.5, and 2.4 hold. Then,*

$$\mathbb{P}\left(\mathcal{E}_3^c\right) \le \frac{2}{N^{10}p^{10}}.$$

**Lemma H.5** *Suppose properties 2.1, 2.5, and 2.4 hold. Then,*

$$\mathbb{P}\left(\mathcal{E}_4^c\right) \le \frac{2}{N^{10}p^{10}}.$$

## H.2 Completing Proof of Lemma 5.1

**Proof H.1** *Recall that our goal is to establish*

$$\mathbb{E}[\|\boldsymbol{Z}^{HSVT,k} - \boldsymbol{A}\|_{2,\infty}^2] \le \frac{C(K_\alpha^2 + \Gamma^2)}{\rho^2}\Big(k + \frac{N\Delta^2}{\rho^2(\tau_k - \tau_{k+1})^2}\Big)\log^{\frac{2}{\alpha}} Np + 2\|\boldsymbol{A}^k - \boldsymbol{A}\|_{2,\infty}^2,$$

*where $C > 0$ is a universal constant. To that end, define $E \triangleq \mathcal{E}_1 \cap \mathcal{E}_2 \cap \mathcal{E}_3 \cap \mathcal{E}_4$. By Lemmas H.2, H.3, H.4 and H.5, it follows that*

$$\mathbb{P}\left(E^c\right) \le \mathbb{P}\left(\mathcal{E}_1^c \cup \mathcal{E}_2^c \cup \mathcal{E}_3^c \cup \mathcal{E}_4^c\right) \le \frac{8}{N^{10}p^{10}}.$$

*Observe (with $\widehat{\boldsymbol{A}} = \boldsymbol{Z}^{HSVT,k}$),*

$$\mathbb{E}[\|\widehat{\boldsymbol{A}} - \boldsymbol{A}\|_{2,\infty}^2] = \mathbb{E}\max_{j\in[p]}\left\|\widehat{\boldsymbol{A}}_{\cdot,j} - \boldsymbol{A}_{\cdot,j}\right\|_2^2$$

$$= \mathbb{E}\left[\max_{j\in[p]}\left\|\widehat{\boldsymbol{A}}_{\cdot,j} - \boldsymbol{A}_{\cdot,j}\right\|_2^2 \cdot \mathbb{1}(E)\right] + \mathbb{E}\left[\max_{j\in[p]}\left\|\widehat{\boldsymbol{A}}_{\cdot,j} - \boldsymbol{A}_{\cdot,j}\right\|_2^2 \cdot \mathbb{1}(E^c)\right]. \quad (29)$$

*In the rest of the proof, we upper bound the two terms in (29) separately.*

**Upper bound on the first term in (29).** *Under event E, from Lemma H.1, we have*

$$\max_{j\in[p]}\left\|\widehat{\boldsymbol{A}}_{\cdot,j} - \boldsymbol{A}_{\cdot,j}\right\|_2^2 \le \frac{C(K_\alpha + \Gamma)^2}{\rho^2}\left(\frac{\Delta^2 N}{\rho^2(\tau_r - \tau_{r+1})^2} + r\right)\log^{\frac{2}{\alpha}}(Np) + 2\max_{j\in[p]}\|\boldsymbol{E}_{\cdot,j}\|_2^2.$$

*where $C > 0$ is an absolute constant. To see this, note that $\varepsilon^2 \le 10$ since $\rho \ge \frac{64\log(Np)}{Np}$; $\|\boldsymbol{A}_j^k\|_2^2 \le \|\boldsymbol{A}_j\|_2^2 \le \Gamma^2 N$ due to argument similar to the contraction property of HSVT operator cf. Lemma H.8 and Property 2.1. Since $\mathbb{P}\left(E\right) \le 1$, it follows that*

$$\mathbb{E}\left[\max_{j\in[p]}\left\|\widehat{\boldsymbol{A}}_{\cdot,j} - \boldsymbol{A}_{\cdot,j}\right\|_2^2 \cdot \mathbb{1}(E)\right] \le \frac{C(K_\alpha + \Gamma)^2}{\rho^2}\left(\frac{\Delta^2 N}{\rho^2(\tau_r - \tau_{r+1})^2} + r\right)\log^{\frac{2}{\alpha}}(Np) + 2\max_{j\in[p]}\|\boldsymbol{E}_{\cdot,j}\|_2^2.$$
$$(30)$$

**Upper bound on the second term in (29).** *To begin with, we note that for any $j \in [p]$,*

$$\left\|\widehat{\boldsymbol{A}}_{\cdot,j} - \boldsymbol{A}_{\cdot,j}\right\|_2 \le \left\|\widehat{\boldsymbol{A}}_{\cdot,j}\right\|_2 + \|\boldsymbol{A}_{\cdot,j}\|_2$$

*by triangle inequality. By the model assumption, the covariates are bounded (Property 2.1) and $\|\boldsymbol{A}_{\cdot,j}\|_2 \le \Gamma\sqrt{N}$ for all $j \in [p]$. By definition, for any $j \in [p]$,*

$$\widehat{\boldsymbol{A}}_{\cdot,j} = \frac{1}{\rho}HSVT_\lambda(\boldsymbol{Z})_{\cdot,j}$$

*for a given threshold $\lambda = s_k$, the kth singular value of $\boldsymbol{Z}$. Therefore,*

$$\|\widehat{\boldsymbol{A}}_{\cdot,j}\|_2 = \frac{1}{\rho}\|HSVT_\lambda(\boldsymbol{Z})_{\cdot,j}\|_2 \overset{(a)}{\le} Np\|HSVT_\lambda(\boldsymbol{Z})_{\cdot,j}\|_2 \overset{(b)}{\le} Np\|\boldsymbol{Z}_{\cdot,j}\|_2.$$

*Here, (a) follows from $\widehat{\rho} \ge \frac{1}{Np}$; and (b) follows from Lemma H.8 – the HSVT operator is a contraction on the columns.*

$$\max_{j\in[p]}\|\widehat{\boldsymbol{A}}_{\cdot,j} - \boldsymbol{A}_{\cdot,j}\|_2 \le \max_{j\in[p]}\|\widehat{\boldsymbol{A}}_{\cdot,j}\|_2 + \max_{j\in[p]}\|\boldsymbol{A}_{\cdot,j}\|_2$$

$$\le Np\max_{j\in[p]}\|\boldsymbol{Z}_{\cdot,j}\|_2 + \Gamma\sqrt{N}$$

$$\le (N^{\frac{3}{2}}p + \sqrt{N})\Gamma + N^{\frac{3}{2}}p\max_{ij}|\eta_{ij}|$$

$$\le 2N^{\frac{3}{2}}p\Big(\Gamma + \max_{ij}|\eta_{ij}|\Big) \quad (31)$$

*because* $\max_{j\in[p]} \|\boldsymbol{Z}_{\cdot,j}\|_2 \leq \sqrt{N} \max_{i,j}|Z_{ij}| \leq \sqrt{N} \max_{i,j}|A_{ij}+\eta_{ij}| \leq \sqrt{N}(\Gamma + \max_{i,j}|\eta_{ij}|)$.
*Now we apply Cauchy-Schwarz inequality on* $\mathbb{E}[\max_{j\in[p]}\|\widehat{\boldsymbol{A}}_{\cdot,j}-\boldsymbol{A}_{\cdot,j}\|_2^2 \cdot \mathbb{1}(E^c)]$ *to obtain*

$$
\begin{aligned}
\mathbb{E}\Big[\max_{j\in[p]}\|\widehat{\boldsymbol{A}}_{\cdot,j}-\boldsymbol{A}_{\cdot,j}\|_2^2 \cdot \mathbb{1}(E^c)\Big] &\leq \mathbb{E}\Big[\max_{j\in[p]}\|\widehat{\boldsymbol{A}}_{\cdot,j}-\boldsymbol{A}_{\cdot,j}\|_2^4\Big]^{\frac{1}{2}} \cdot \mathbb{E}\Big[\mathbb{1}(E^c)\Big]^{\frac{1}{2}} \\
&= \mathbb{E}\Big[\max_{j\in[p]}\|\widehat{\boldsymbol{A}}_{\cdot,j}-\boldsymbol{A}_{\cdot,j}\|_2^4\Big]^{\frac{1}{2}} \cdot \mathbb{P}\left(E^c\right)^{\frac{1}{2}} \\
&\overset{(a)}{\leq} 4N^3 p^2 \mathbb{E}\Big[\big(\Gamma+\max_{ij}|\eta_{ij}|\big)^4\Big]^{\frac{1}{2}} \cdot \mathbb{P}\left(E^c\right)^{\frac{1}{2}} \\
&\overset{(b)}{\leq} 8\sqrt{2}N^3 p^2 \big(\Gamma^4 + \mathbb{E}[\max_{ij}|\eta_{ij}|^4]\big)^{\frac{1}{2}} \cdot \mathbb{P}\left(E^c\right)^{\frac{1}{2}} \\
&\overset{(c)}{\leq} 8\sqrt{2}N^3 p^2 \big(\Gamma^2 + \mathbb{E}[\max_{ij}|\eta_{ij}|^4]^{\frac{1}{2}}\big) \cdot \mathbb{P}\left(E^c\right)^{\frac{1}{2}}. \qquad (32)
\end{aligned}
$$

*Here, (a) follows from* (31)*; and (b) follows from Jensen's inequality:*

$$
\begin{aligned}
\mathbb{E}\Big[\big(\Gamma+\max_{ij}|\eta_{ij}|\big)^4\Big] &= \mathbb{E}\Big[\Big(\frac{1}{2}(2\Gamma+2\max_{ij}|\eta_{ij}|)\Big)^4\Big] \leq \mathbb{E}\Big[\frac{1}{2}\big((2\Gamma)^4+(2\max_{ij}|\eta_{ij}|)^4\big)\Big] \\
&= 8\mathbb{E}\Big[\Gamma^4+\max_{ij}|\eta_{ij}|^4\Big] = 8\Big(\Gamma^4+\mathbb{E}[\max_{ij}|\eta_{ij}|^4]\Big);
\end{aligned}
$$

*and (c) follows from the trivial inequality:* $\sqrt{A+B} \leq \sqrt{A}+\sqrt{B}$ *for any* $A, B \geq 0$.

*Now it remains to find an upper bound for* $\mathbb{E}[\max_{ij}|\eta_{ij}|^4]$. *Note that for any* $\alpha > 0$ *and* $\theta \geq 1$, $\eta_{ij}$ *being a* $\psi_\alpha$-*random variable implies that* $|\eta_{ij}|^\theta$ *is a* $\psi_{\alpha/\theta}$-*random variable. With the choice of* $\theta = 4$, *we have that*

$$
\mathbb{E}\max_{ij}|\eta_{ij}|^4 \leq C' K_\alpha^4 \log^{\frac{4}{\alpha}}(Np) \qquad (33)
$$

*for some* $C' > 0$ *by Lemma D.5 (also see Remark D.1). Inserting* (33) *to* (32) *yields*

$$
\begin{aligned}
\mathbb{E}\Big[\max_{j\in[p]}\|\widehat{\boldsymbol{A}}_{\cdot,j}-\boldsymbol{A}_{\cdot,j}\|_2^2 \cdot \mathbb{1}(\mathcal{E}^c)\Big] &\leq 8\sqrt{2}N^3 p^2 \Big(\Gamma^2 + C'^{1/2}K_\alpha^2 \log^{\frac{2}{\alpha}}(Np)\Big) \cdot \mathbb{P}\left(E^c\right)^{\frac{1}{2}} \\
&\overset{(a)}{\leq} 32\Big(\Gamma^2 + C'^{1/2}K_\alpha^2 \log^{\frac{2}{\alpha}}(Np)\Big)\frac{1}{N^2 p^2}, \qquad (34)
\end{aligned}
$$

*where (a) follows from recalling that* $\mathbb{P}\left(E^c\right) \leq 8/N^{10}p^{10}$.

**Concluding the Proof.** *Thus, combining* (30) *and* (34) *in* (29) *and noticing that term in* (34) *is smaller order term than that in* (30)*, by defining appropriate constant* $C > 0$*, we obtain the desired bound:*

$$
\begin{aligned}
\mathbb{E}[\|\widehat{\boldsymbol{A}}-\boldsymbol{A}\|_{2,\infty}^2] \leq &\frac{C(K_\alpha+\Gamma)^2}{\rho^2}\left(\frac{\Delta^2 N}{\rho^2(\tau_r-\tau_{r+1})^2}+r\right)\log^{\frac{2}{\alpha}}(Np) + 2\max_{j\in[p]}\|\boldsymbol{E}_{\cdot,j}\|_2^2 \\
&+ \frac{C}{N^2 p^2}\Big(\Gamma^2 + K_\alpha^2 \log^{\frac{2}{\alpha}}(Np)\Big),
\end{aligned}
$$

*with*

$$
\Delta = \sqrt{N\rho}\sqrt{\rho\gamma^2+(1-\rho)\Gamma^2} + 2C(\alpha)\sqrt{p}(K_\alpha+\Gamma)\Big(1+9\log(Np)\Big)^{\frac{1}{\alpha}}\sqrt{\log(Np)}.
$$

### H.3 More on HSVT

#### H.3.1 Interlacing of Singular Values

**Lemma H.6** *Given covariate matrix* $\boldsymbol{A}\mathbb{R}^{N\times p}$ *and its noisy observation with missing values,* $\boldsymbol{Z} \in \mathbb{R}^{N\times p}$*, let* $2\|\boldsymbol{Z}-\rho\boldsymbol{A}\| < \rho(\tau_k-\tau_{k+1})$ *where* $\tau_i$ *is ith singular value of* $\boldsymbol{A}$ *for* $i \in [N]$*. Then,*

$$
s_{k+1} \leq \rho\tau_{k+1}+\|\boldsymbol{Z}-\rho\boldsymbol{A}\| \;<\; \rho\tau_k-\|\boldsymbol{Z}-\rho\boldsymbol{A}\| \;\leq s_k,
$$

*where* $s_i$ *is the ith singular value of* $\boldsymbol{Z}$ *for* $i \in [N]$.

**Proof H.2** *We may write*

$$\boldsymbol{Z} = \rho\boldsymbol{A} + (\boldsymbol{Z} - \rho\boldsymbol{A}).$$

*Recall that $s_i$ are the singular values of $\boldsymbol{Z}$. Then, from Weyl's inequality as in Lemma D.6 the result follows immediately.*

### H.3.2 Column Operator Induced by HSVT

**Lemma H.7** *Let $\boldsymbol{B} \in \mathbb{R}^{N \times p}$ and $\lambda \geq 0$ be given. Then for any $j \in [p]$,*

$$\varphi_\lambda^B(\boldsymbol{B}_{\cdot,j}) = \mathrm{HSVT}_\lambda(\boldsymbol{B})_{\cdot,j}.$$

**Proof H.3** *By (28) and the orthonormality of the left singular vectors,*

$$
\begin{aligned}
\varphi_\lambda^B(\boldsymbol{B}_{\cdot,j}) &= \sum_{i=1}^{N\wedge p} \mathbb{1}(\sigma_i(\boldsymbol{B}) \geq \lambda) x_i x_i^T \boldsymbol{B}_{\cdot,j} = \sum_{i=1}^{N\wedge p} \mathbb{1}(\sigma_i(\boldsymbol{B}) \geq \lambda) x_i x_i^T \Big( \sum_{i'=1}^{N\wedge p} \sigma_{i'}(\boldsymbol{B}) x_{i'} y_{i'} \Big)_{\cdot,j} \\
&= \sum_{i,i'=1}^{N\wedge p} \sigma_{i'}(\boldsymbol{B}) \mathbb{1}(\sigma_i(\boldsymbol{B}) \geq \lambda) x_i x_i^T x_{i'} (y_{i'})_j = \sum_{i,i'=1}^{N\wedge p} \sigma_{i'}(\boldsymbol{B}) \mathbb{1}(\sigma_i(\boldsymbol{B}) \geq \lambda) x_i \delta_{ii'} (y_{i'})_j \\
&= \sum_{i=1}^{N\wedge p} \mathbb{1}(\sigma_i(\boldsymbol{B}) \geq \lambda^*) \sigma_i x_i (y_i)_j \\
&= HSVT_\lambda(\boldsymbol{B})_{\cdot,j}.
\end{aligned}
$$

*This completes the proof.*

**Remark H.3** *Suppose we have missing data. Then the estimator $\widehat{\boldsymbol{A}}$ has the following representation:*

$$\widehat{\boldsymbol{A}} = \frac{1}{\widehat{\rho}} \mathrm{HSVT}_{\lambda^*}(\boldsymbol{Z}) = \frac{1}{\widehat{\rho}} \sum_{i=1}^{N\wedge p} s_i \mathbb{1}(s_i \geq \lambda^*) \cdot u_i v_i^T.$$

*By Lemma H.7, we note that*

$$\widehat{\boldsymbol{A}}_{\cdot,j} = \frac{1}{\widehat{\rho}} \varphi_{\lambda^*}^Z(\boldsymbol{Z}_{\cdot,j}). \tag{35}$$

### H.3.3 HSVT Operator is a Contraction

**Lemma H.8** *Let $\boldsymbol{B} \in \mathbb{R}^{N \times p}$ and $\lambda \geq 0$ be given. Then for any $j \in [p]$,*

$$\left\| \mathrm{HSVT}_\lambda(\boldsymbol{B})_{\cdot,j} \right\|_2 \leq \|\boldsymbol{B}_{\cdot,j}\|_2.$$

**Proof H.4** *By (28) and Lemma H.7, we have*

$$
\begin{aligned}
\left\| HSVT_\lambda(\boldsymbol{B})_{\cdot,j} \right\|_2^2 &= \left\| \varphi_\lambda^B(\boldsymbol{B}_{\cdot,j}) \right\|_2^2 = \left\| \sum_{i=1}^{N\wedge p} \mathbb{1}(\sigma_i(\boldsymbol{B}) \geq \lambda) \cdot x_i x_i^T \cdot \boldsymbol{B}_{\cdot,j} \right\|_2^2 \\
&\stackrel{(a)}{=} \sum_{i=1}^{N\wedge p} \left\| \mathbb{1}(\sigma_i(\boldsymbol{B}) \geq \lambda) \cdot x_i x_i^T \cdot \boldsymbol{B}_{\cdot,j} \right\|_2^2 \leq \sum_{i=1}^{N\wedge p} \left\| x_i x_i^T \cdot \boldsymbol{B}_{\cdot,j} \right\|_2^2 \\
&\stackrel{(b)}{=} \left\| \sum_{i=1}^{N\wedge p} x_i x_i^T \cdot \boldsymbol{B}_{\cdot,j} \right\|_2^2 = \|\boldsymbol{B}_{\cdot,j}\|_2^2.
\end{aligned}
$$

*Note that (a) and (b) use the orthonormality of the left singular vectors.*

### H.4 Proof of Lemma H.1

**Proof H.5** *First, we recall three conditions assumed in the Lemma that will be used in the proof:*

1. $\|Z - \rho A\| \leq \Delta$ *for some* $\Delta \geq 0$.
2. $\frac{1}{\varepsilon}\rho \leq \widehat{\rho} \leq \varepsilon\rho$ *for some* $\varepsilon \geq 1$.

*We will use notation* $\lambda^* = s_k$, *the kth singular value of* $Z$ *for simplicity. We prove our Lemma in three steps.*

**Step 1.** *Fix a column index* $j \in [p]$. *Observe that*

$$\widehat{A}_{\cdot,j} - A_{\cdot,j} = \left(\widehat{A}_{\cdot,j} - \varphi_{\lambda^*}^{Z}(A_{\cdot,j})\right) + \left(\varphi_{\lambda^*}^{Z}(A_{\cdot,j}) - A_{\cdot,j}\right).$$

*By choice,* $\mathrm{rank}(\widehat{A}) = k$. *By definition (see (28)), we have that* $\varphi_{\lambda^*}^{Z} : \mathbb{R}^N \to \mathbb{R}^N$ *is the projection operator onto the span of the top* $k$ *left singular vectors of* $Z$, *namely,* $\mathrm{span}\{u_1, \ldots, u_k\}$. *Therefore,*

$$\varphi_{\lambda^*}^{Z}(A_{\cdot,j}) - A_{\cdot,j} \in \mathrm{span}\{u_1, \ldots, u_k\}^{\perp}$$

*and by (35) (using Lemma H.7),*

$$\widehat{A}_{\cdot,j} - \varphi_{\lambda^*}^{Z}(A_{\cdot,j}) = \frac{1}{\widehat{\rho}}\varphi_{\lambda^*}^{Z}(Z_{\cdot,j}) - \varphi_{\lambda^*}^{Z}(A_{\cdot,j}) \in \mathrm{span}\{u_1, \ldots, u_k\}.$$

*Hence,* $\langle \widehat{A}_{\cdot,j} - \varphi_{\lambda^*}^{Z}(A_{\cdot,j}), \varphi_{\lambda^*}^{Z}(A_{\cdot,j}) - A_{\cdot,j}\rangle = 0$ *and*

$$\left\|\widehat{A}_{\cdot,j} - A_{\cdot,j}\right\|_2^2 = \left\|\widehat{A}_{\cdot,j} - \varphi_{\lambda^*}^{Z}(A_{\cdot,j})\right\|_2^2 + \left\|\varphi_{\lambda^*}^{Z}(A_{\cdot,j}) - A_{\cdot,j}\right\|_2^2 \tag{36}$$

*by the Pythagorean theorem. It remains to bound the terms on the right hand side of (36).*

**Step 2.** *We begin by bounding the first term on the right hand side of (36). Again applying Lemma H.7, we can rewrite*

$$\widehat{A}_{\cdot,j} - \varphi_{\lambda^*}^{Z}(A_{\cdot,j}) = \frac{1}{\widehat{\rho}}\varphi_{\lambda^*}^{Z}(Z_{\cdot,j}) - \varphi_{\lambda^*}^{Z}(A_{\cdot,j}) = \varphi_{\lambda^*}^{Z}\left(\frac{1}{\widehat{\rho}}Z_{\cdot,j} - A_{\cdot,j}\right)$$

$$= \frac{1}{\widehat{\rho}}\varphi_{\lambda^*}^{Z}(Z_{\cdot,j} - \rho A_{\cdot,j}) + \frac{\rho - \widehat{\rho}}{\widehat{\rho}}\varphi_{\lambda^*}^{Z}(A_{\cdot,j}).$$

*Using the Parallelogram Law (or, equivalently, combining Cauchy-Schwartz and AM-GM inequalities), we obtain*

$$\left\|\widehat{A}_{\cdot,j} - \varphi_{\lambda^*}^{Z}(A_{\cdot,j})\right\|_2^2 = \left\|\frac{1}{\widehat{\rho}}\varphi_{\lambda^*}^{Z}(Z_{\cdot,j} - \rho A_{\cdot,j}) + \frac{\rho - \widehat{\rho}}{\widehat{\rho}}\varphi_{\lambda^*}^{Z}(A_{\cdot,j})\right\|_2^2$$

$$\leq 2\left\|\frac{1}{\widehat{\rho}}\varphi_{\lambda^*}^{Z}(Z_{\cdot,j} - \rho A_{\cdot,j})\right\|_2^2 + 2\left\|\frac{\rho - \widehat{\rho}}{\widehat{\rho}}\varphi_{\lambda^*}^{Z}(A_{\cdot,j})\right\|_2^2$$

$$\leq \frac{2}{\widehat{\rho}^2}\left\|\varphi_{\lambda^*}^{Z}(Z_{\cdot,j} - \rho A_{\cdot,j})\right\|_2^2 + 2\left(\frac{\rho - \widehat{\rho}}{\widehat{\rho}}\right)^2\|A_{\cdot,j}\|_2^2$$

$$\leq \frac{2\varepsilon^2}{\rho^2}\left\|\varphi_{\lambda^*}^{Z}(Z_{\cdot,j} - \rho A_{\cdot,j})\right\|_2^2 + 2(\varepsilon - 1)^2\|A_{\cdot,j}\|_2^2. \tag{37}$$

*because Condition 2 implies* $\frac{1}{\widehat{\rho}} \leq \frac{\varepsilon}{\rho}$ *and* $\left(\frac{\rho - \widehat{\rho}}{\widehat{\rho}}\right)^2 \leq (\varepsilon - 1)^2$.

*Note that the first term of (37) can further be decomposed (using the Parallelogram Law and recalling* $A = A^k + E$, *we have*

$$\left\|\varphi_{\lambda^*}^{Z}(Z_{\cdot,j} - \rho A_{\cdot,j})\right\|_2^2$$
$$\leq 2\left\|\varphi_{\lambda^*}^{Z}(Z_{\cdot,j} - \rho A_{\cdot,j}) - \varphi^{A^k}(Z_{\cdot,j} - \rho A_{\cdot,j})\right\|_2^2 + 2\left\|\varphi^{A^k}(Z_{\cdot,j} - \rho A_{\cdot,j})\right\|_2^2. \tag{38}$$

We now bound the first term on the right hand side of (38) separately. First, we apply the Davis-Kahan $\sin\Theta$ Theorem (see [31, 54]) to arrive at the following inequality:

$$\|\mathcal{P}_{u_1,\ldots,u_k} - \mathcal{P}_{\mu_1,\ldots,\mu_k}\|_2 \le \frac{\|\boldsymbol{Z} - \rho\boldsymbol{A}\|}{\rho\tau_k - \rho\tau_{k+1}} \le \frac{\Delta}{\rho(\tau_k - \tau_{k+1})} \tag{39}$$

where $\mathcal{P}_{u_1,\ldots,u_k}$ and $\mathcal{P}_{\mu_1,\ldots,\mu_k}$ denote the projection operators onto the span of the top $k$ left singular vectors of $\boldsymbol{Z}$ and $\boldsymbol{A}^k$, respectively. We utilized Condition 1 to bound $\|\boldsymbol{Z} - \rho\boldsymbol{A}\|_2 \le \Delta$. Then it follows that

$$\left\|\varphi_{\lambda^*}^{\boldsymbol{Z}}(\boldsymbol{Z}_{\cdot,j} - \rho\boldsymbol{A}_{\cdot,j}) - \varphi^{\boldsymbol{A}^k}(\boldsymbol{Z}_{\cdot,j} - \rho\boldsymbol{A}_{\cdot,j})\right\|_2 \le \|\mathcal{P}_{u_1,\ldots,u_k} - \mathcal{P}_{\mu_1,\ldots,\mu_k}\|_2 \|\boldsymbol{Z}_{\cdot,j} - \rho\boldsymbol{A}_{\cdot,j}\|_2$$

$$\le \frac{\Delta}{\rho(\tau_k - \tau_{k+1})}\|\boldsymbol{Z}_{\cdot,j} - \rho\boldsymbol{A}_{\cdot,j}\|_2.$$

Combining the inequalities together, we have

$$\left\|\widehat{\boldsymbol{A}}_{\cdot,j} - \varphi_{\lambda^*}^{\boldsymbol{Z}}(\boldsymbol{A}_{\cdot,j})\right\|_2^2 \le \frac{4\varepsilon^2}{\rho^2}\frac{\Delta^2}{\rho^2(\tau_k - \tau_{k+1})^2}\|\boldsymbol{Z}_{\cdot,j} - \rho\boldsymbol{A}_{\cdot,j}\|_2^2$$

$$+ \frac{4\varepsilon^2}{\rho^2}\left\|\varphi^{\boldsymbol{A}^k}(\boldsymbol{Z}_{\cdot,j} - \rho\boldsymbol{A}_{\cdot,j})\right\|_2^2 + 2(\varepsilon - 1)^2\|\boldsymbol{A}_{\cdot,j}\|_2^2. \tag{40}$$

**Step 3.** We now bound the second term of (36). Recalling $\boldsymbol{A} = \boldsymbol{A}^k + \boldsymbol{E}$ and using (39)

$$\left\|\varphi_{\lambda^*}^{\boldsymbol{Z}}(\boldsymbol{A}_{\cdot,j}) - \boldsymbol{A}_{\cdot,j}\right\|_2^2 = \left\|\varphi_{\lambda^*}^{\boldsymbol{Z}}(\boldsymbol{A}_{\cdot,j}^k + \boldsymbol{E}_{\cdot,j}) - \boldsymbol{A}_{\cdot,j}^k - \boldsymbol{E}_{\cdot,j}\right\|_2^2$$

$$\le 2\left\|\varphi_{\lambda^*}^{\boldsymbol{Z}}(\boldsymbol{A}_{\cdot,j}^k) - \boldsymbol{A}_{\cdot,j}^k\right\|_2^2 + 2\left\|\varphi_{\lambda^*}^{\boldsymbol{Z}}(\boldsymbol{E}_{\cdot,j}) - \boldsymbol{E}_{\cdot,j}\right\|_2^2$$

$$= 2\left\|\varphi_{\lambda^*}^{\boldsymbol{Z}}(\boldsymbol{A}_{\cdot,j}^k) - \varphi^{\boldsymbol{A}^k}(\boldsymbol{A}_{\cdot,j}^k)\right\|_2^2 + 2\left\|\varphi_{\lambda^*}^{\boldsymbol{Z}}(\boldsymbol{E}_{\cdot,j}) - \boldsymbol{E}_{\cdot,j}\right\|_2^2$$

$$\le 2\left\|\mathcal{P}_{u_1,\ldots,u_k} - \mathcal{P}_{\mu_1,\ldots,\mu_k}\right\|^2\left\|\boldsymbol{A}_{\cdot,j}^k\right\|_2^2 + 2\left\|\boldsymbol{E}_{\cdot,j}\right\|_2^2$$

$$\le \frac{2\Delta^2}{\rho^2(\tau_k - \tau_{k+1})^2}\left\|\boldsymbol{A}_{\cdot,j}^k\right\|_2^2 + 2\left\|\boldsymbol{E}_{\cdot,j}\right\|_2^2. \tag{41}$$

Inserting (40) and (41) back to (36) completes the proof.

## H.5 Proof of $\mathcal{E}_1, \mathcal{E}_2, \mathcal{E}_3, \mathcal{E}_4$ Being High-probability Events

### H.5.1 Proof of Lemma H.2

**Proof H.6** Observe that $\|\boldsymbol{A}_{\cdot,j}\|_2^2 \le N\Gamma^2$ when Property 2.1 holds, and $\|\mathbb{E}\boldsymbol{H}^T\boldsymbol{H}\| \le N\gamma^2$ when Property 2.5 holds. By Theorem G.1, we know that for any $\delta_1 > 0$,

$$\|\boldsymbol{Z} - \rho\boldsymbol{A}\| \le \sqrt{N\rho}\sqrt{\rho\gamma^2 + (1-\rho)\Gamma^2}$$

$$+ C(\alpha)\sqrt{1 + \delta_1}\sqrt{p}(K_\alpha + \Gamma)\left(1 + (2 + \delta_1)\log(Np)\right)^{\frac{1}{\alpha}}\sqrt{\log(Np)}$$

with probability at least $1 - \frac{2}{N^{1+\delta_1}p^{\delta_1}}$.

### H.5.2 Proof of Lemma H.3

**Proof H.7** Recall that $\widehat{\rho} = \frac{1}{Np}\sum_{i=1}^N\sum_{j=1}^p\mathbb{1}(Z_{ij} \ne \star) \vee \frac{1}{Np}$. By the binomial Chernoff bound, for $\varepsilon > 1$,

$$\mathbb{P}\left(\widehat{\rho} > \varepsilon\rho\right) \le \exp\left(-\frac{(\varepsilon - 1)^2}{\varepsilon + 1}Np\rho\right), \quad \text{and}$$

$$\mathbb{P}\left(\widehat{\rho} < \frac{1}{\varepsilon}\rho\right) \le \exp\left(-\frac{(\varepsilon - 1)^2}{2\varepsilon^2}Np\rho\right).$$

By the union bound,

$$\mathbb{P}\left(\frac{1}{\varepsilon}\rho \le \widehat{\rho} \le \varepsilon\rho\right) \ge 1 - \mathbb{P}\left(\widehat{\rho} > \varepsilon\rho\right) - \mathbb{P}\left(\widehat{\rho} < \frac{1}{\varepsilon}\rho\right).$$

Noticing $\varepsilon + 1 < 2\varepsilon < 2\varepsilon^2$ for all $\varepsilon > 1$ completes the proof.

### H.5.3 Two Helper Lemmas for the Proof of Lemmas H.4 and H.5

**Lemma H.9** *Assume Properties 2.1, 2.5, and 2.4 hold. Then for any $\alpha \geq 1$ with which Property 2.5 holds,*

$$\|\boldsymbol{Z}_{\cdot,j} - \rho \boldsymbol{A}_{\cdot,j}\|_{\psi_\alpha} \leq C(K_\alpha + \Gamma), \qquad \forall j \in [p]$$

*where $C > 0$ is an absolute constant.*

**Proof H.8** *Observe that*

$$
\begin{aligned}
\|\boldsymbol{Z}_{\cdot,j} - \rho \boldsymbol{A}_{\cdot,j}\|_{\psi_\alpha} &= \sup_{u \in \mathbb{S}^{N-1}} \left\| u^T (\boldsymbol{Z}_{\cdot,j} - \rho \boldsymbol{A}_{\cdot,j}) \right\|_{\psi_\alpha} \\
&= \sup_{u \in \mathbb{S}^{N-1}} \left\| u^T (\boldsymbol{Z} - \rho \boldsymbol{A}) e_j \right\|_{\psi_\alpha} \\
&= \sup_{u \in \mathbb{S}^{N-1}} \left\| \sum_{i=1}^{n} u_i (\boldsymbol{Z}_{i,\cdot} - \rho \boldsymbol{A}_{i,\cdot}) e_j \right\|_{\psi_\alpha} \\
&\overset{(a)}{\leq} C \sup_{u \in \mathbb{S}^{N-1}} \left( \sum_{i=1}^{n} u_i^2 \|(\boldsymbol{Z}_{i,\cdot} - \rho \boldsymbol{A}_{i,\cdot}) e_j\|_{\psi_\alpha}^2 \right)^{1/2} \\
&\leq C \max_{i \in [N]} \|\boldsymbol{Z}_{i,\cdot} - \rho \boldsymbol{A}_{i,\cdot}\|_{\psi_\alpha},
\end{aligned}
$$

*where (a) follows from Lemma G.5. Then the conclusion follows from Lemma G.2.*

**Lemma H.10** *Let $W_1, \ldots, W_n$ be a sequence of $\psi_\alpha$-random variables for some $\alpha \geq 1$. For any $t \geq 0$,*

$$\mathbb{P}\left( \sum_{i=1}^{n} W_i^2 > t \right) \leq 2 \sum_{i=1}^{n} \exp\left( -\left( \frac{t}{n\|W_i\|_{\psi_\alpha}^2} \right)^{\alpha/2} \right).$$

**Proof H.9** *Note that $\sum_{i=1}^{n} W_i^2 > t$ implies that there exists at least one $i \in [n]$ with $W_i^2 > \frac{t}{n}$. By the union bound,*

$$\mathbb{P}\left( \sum_{i=1}^{n} W_i^2 > t \right) \leq \sum_{i=1}^{n} \mathbb{P}\left( W_i^2 > \frac{t}{n} \right) \leq \sum_{i=1}^{n} \mathbb{P}\left( |W_i| > \sqrt{\frac{t}{n}} \right) \leq \sum_{i=1}^{n} 2 \exp\left( -\left( \frac{t}{n\|W_i\|_{\psi_\alpha}^2} \right)^{\alpha/2} \right).$$

### H.5.4 Proof of Lemma H.4

**Proof H.10** *Fix $j \in [p]$. Let $e_i \in \mathbb{R}^N$ denote the $i$-th canonical basis of $\mathbb{R}^N$ (column vector representation). Note that*

$$\left\| \boldsymbol{Z}_{\cdot,j} - \rho \boldsymbol{A}_{\cdot,j} \right\|_2^2 = \sum_{i=1}^{N} \left( e_i^T (\boldsymbol{Z}_{\cdot,j} - \rho \boldsymbol{A}_{\cdot,j}) \right)^2$$

*and $e_i^T (\boldsymbol{Z}_{\cdot,j} - \rho \boldsymbol{A}_{\cdot,j})$ is a $\psi_\alpha$-random variable with $\left\| e_i^T (\boldsymbol{Z}_{\cdot,j} - \rho \boldsymbol{A}_{\cdot,j}) \right\|_{\psi_\alpha} \leq \|\boldsymbol{Z}_{\cdot,j} - \rho \boldsymbol{A}_{\cdot,j}\|_{\psi_\alpha}$. By Lemma H.9, $\|\boldsymbol{Z}_{\cdot,j} - \rho \boldsymbol{A}_{\cdot,j}\|_{\psi_\alpha} \leq C(K_\alpha + \Gamma)$ for all $j \in [p]$. By Lemma H.10 and the union bound,*

$$
\begin{aligned}
\mathbb{P}\left( \mathcal{E}_3^c \right) &\leq \sum_{j=1}^{p} \mathbb{P}\left( \left\| \boldsymbol{Z}_{\cdot,j} - \rho \boldsymbol{A}_{\cdot,j} \right\|_2^2 > 11 C^2 (K_\alpha + \Gamma)^2 N \log^{\frac{2}{\alpha}}(Np) \right) \\
&\leq 2 \sum_{j=1}^{p} \sum_{i=1}^{N} \exp\left( -11 \log(Np) \right) \\
&= \frac{2}{N^{10} p^{10}}.
\end{aligned}
$$

### H.5.5 Proof of Lemma H.5

**Proof H.11** *Recall that* $rank(\boldsymbol{A}^k) = k$. *We write*

$$\left\| \varphi^{\boldsymbol{A}^k} (\boldsymbol{Z}_{\cdot,j} - \rho \boldsymbol{A}_{\cdot,j}) \right\|_2^2 = \sum_{i=1}^{k} \left( u_i^T (\boldsymbol{Z}_{\cdot,j} - \rho \boldsymbol{A}_{\cdot,j}) \right)^2,$$

*where* $u_1, \ldots, u_k$ *denote the left singular vectors of* $\boldsymbol{A}^k$. *The proof has the same structure with that of Lemma H.4 with* $u_1, \ldots, u_k$ *in place of* $e_1, \ldots, e_n$.

## I  Proof of Corollary 5.1

Corollary 5.1 follows from Lemma 5.1 and Theorem 5.1. The key step is simplification of bound on $\Delta$, as stated in (42), which we briefly discuss here. To that end, since $\alpha \geq 1$,

$$\Delta^2 \leq C_1(\alpha) \Big( N\rho(\gamma^2 + \Gamma^2) + p(K_\alpha + \Gamma)^2 \log^3(Np) \Big) \tag{42}$$
$$\leq C_2(\alpha)(1 + \gamma + \Gamma + K_\alpha)^2 (N\rho + p) \log^3(Np).$$

for some constants $C_1(\alpha), C_2(\alpha)$, which may depend on $\alpha$. Using this bound, replacing in Lemma 5.1 and subsequently in Theorem 5.1 with $n = \Theta(N)$, we obtain the desired result of Corollary 5.1.

## J  Proof of Theorem 4.1

The proof of Theorem 4.1 follows from Corollary 5.1 by observing that $\phi = 0$, and for $k = r$, $\boldsymbol{A}^k = \boldsymbol{A}$ and $\tau_{k+1} = \tau_{r+1} = 0$.

## K  Proof of Theorem 4.2

The proof of Theorem 4.2 follows the standard approach in terms of establishing generalization error bounds using Rademacher complexity (cf. [16] and references therein). We note two important contributions: (1) relating our notion of generalization error to the standard definitions; (2) arguing that the Rademacher complexity of our matrix estimation regression algorithm (using HSVT) can be identified with the Rademacher complexity of regression with $\ell_0$-regularization.

**Outline.** We start by introducing some useful notation. We define a conditioning event of relevance and show that this event occurs with high probability. Lemma K.3 then bounds the expected generalization error in terms of the Rademacher complexity of the class of squared loss functions for linear predictors. Due to Lemma 4.1, we analyze the Rademacher complexity of squared loss functions under $r$-sparse linear predictors, which is summarized in Lemma K.4. Using these, we conclude the proof of Theorem 4.2.

**Notation, Setup.** We consider PCR with parameter $k = r$ for some $r \geq 1$. Recall that the training sample set $\Omega \subset [N]$ with $|\Omega| = n$. The PCR with parameter $r$ is equivalent to Linear Regression with pre-processing of noisy covariates using HSVT as argued in Proposition 3.1. Let $\widehat{\boldsymbol{A}} = \boldsymbol{Z}^{\text{HSVT},r}$ and $\widehat{\beta} = \beta^{\text{HSVT},r}$.

**Model Class.** We now state our model class of consideration for the purposes of generalization:

$$\mathcal{F} = \{\beta \in \mathbb{R}^p : \|\beta\|_2 \leq B, \ \|\beta\|_0 \leq r\},$$

where $B > 0$ is an absolute constant. We will now justify the above model class of interest.

As aforementioned, the goal of this work is to analyze the prediction properties of the PCR algorithm; hence, for the purposes of generalization (and in line with standard assumptions on generalization properties of linear regression algorithms), we begin by restricting the hypothesis class of candidate regression vectors $\mathcal{F}_{\text{PCR}}$ to have bounded $\ell_2$-norm, i.e., $\mathcal{F}_{\text{PCR}} = \{\beta \in \mathbb{R}^r : \|\beta\|_2 \leq B\}$. For any $\beta^{\text{PCR},r} \in \mathcal{F}_{\text{PCR}}$, we highlight that

$$\widehat{Y}^{\text{PCR}} = \boldsymbol{Z}^{\text{PCR},r} \cdot \beta^{\text{PCR},r} = \widetilde{\boldsymbol{Z}} \cdot \boldsymbol{V}_r \cdot \beta^{\text{PCR},r} = \boldsymbol{Z}_r \cdot \boldsymbol{V}_r \cdot \beta^{\text{PCR},r} = \widehat{Y}^{\text{HSVT}}. \tag{43}$$

Recall that $\boldsymbol{Z}_r = \boldsymbol{U}_r \boldsymbol{S}_r \boldsymbol{V}_r^T$, where $\boldsymbol{U}_r = [u_1, \ldots, u_r], \boldsymbol{V}_r = [v_1, \ldots, v_r]$, and $\boldsymbol{S}_r = \text{diag}(s_1, \ldots, s_r)$ denote the top $r$ left and right singular vectors, and singular values, respectively (as defined in Section 3.1); hence, $\boldsymbol{Z}^{\text{HSVT},r} = \boldsymbol{Z}_r$. This allows us to rewrite (43) as

$$\widehat{Y}^{\text{HSVT}} = \boldsymbol{Z}^{\text{HSVT},r} \cdot \boldsymbol{V}_r \cdot \beta^{\text{PCR},r} = \boldsymbol{Z}^{\text{HSVT},r} \cdot \beta^{\text{HSVT},r},$$

where $\beta^{\text{HSVT},r} = \boldsymbol{V}_r \cdot \beta^{\text{PCR},r} \in \mathbb{R}^p$. Using the orthonormality property of the vectors in $\boldsymbol{V}_r$, we obtain the following $\ell_2$-bound for any $\beta^{\text{HSVT},r}$:

$$\begin{aligned}
\|\beta^{\text{HSVT},r}\|_2^2 &= \|\boldsymbol{V}_r \cdot \beta^{\text{PCR},r}\|_2^2 \\
&= \|\sum_{j=1}^{r} \beta_j^{\text{PCR},r} \cdot v_j\|_2^2 \\
&= \sum_{j=1}^{r} (\beta_j^{\text{PCR},r})^2 \cdot \|v_j\|_2^2 \\
&= \sum_{j=1}^{r} (\beta_j^{\text{PCR},r})^2 \\
&= \|\beta^{\text{PCR},r}\|_2^2 \ \leq B^2.
\end{aligned}$$

Thus, we consider the collection of candidate vectors $\beta^{\text{HSVT},r} = \boldsymbol{V}_r \cdot \beta^{\text{PCR},r} \in \mathbb{R}^p$ such that $\|\beta^{\text{HSVT},r}\|_2 \leq B$.

Further, by definition, recall that $\boldsymbol{Z}^{\text{HSVT},r}$ has rank $r$. Then by Proposition 4.1, for any $\boldsymbol{Z}^{\text{HSVT},r}$ and corresponding $\beta^{\text{HSVT},r} \in \mathbb{R}^p$, there exists an $r$-sparse vector $\beta' \in \mathbb{R}^p$ such that

$$\boldsymbol{Z}^{\text{HSVT},r} \cdot \beta^{\text{HSVT},r} = \boldsymbol{Z}^{\text{HSVT},r} \cdot \beta'.$$

Therefore, we consider the collection of candidate vectors $\beta' \in \mathbb{R}^p$ that are $r$-sparse, i.e., $\|\beta'\|_0 \leq r$. In other words, for analyzing properties of PCR with parameter $r$, or equivalently Linear regression with covariate pre-processing using HSVT with rank $r$ thresholding, we can restrict our model class to linear predictors with sparsity $r$.

Given the above two observations, we will consider the family of regression vectors defined by $\mathcal{F}$, which have bounded $\ell_2$-norm and are $r$-sparse.

**Generalization error and Rademacher complexity.** For any hypothesis $\beta \in \mathcal{F}$ and training set $\Omega$, the empirical error is

$$\widehat{\mathcal{E}}_\Omega(\beta) = \frac{1}{n} \sum_{\omega \in \Omega} \left( \widehat{\boldsymbol{A}}_{\omega,\cdot} \beta - \boldsymbol{A}_{\omega,\cdot} \beta^* \right)^2. \tag{44}$$

Similarly, we define the test error as

$$\mathcal{E}(\beta) = \frac{1}{N} \sum_{i=1}^{N} \left( \widehat{\boldsymbol{A}}_{i,\cdot} \beta - \boldsymbol{A}_{i,\cdot} \beta^* \right)^2. \tag{45}$$

The generalization error is defined as the supremum of the gap between (44) and (45) over $\mathcal{F}$. Precisely, for training set $\Omega$,

$$\phi(\Omega) = \sup_{\beta \in \mathcal{F}} \left( \mathcal{E}(\beta) - \widehat{\mathcal{E}}_\Omega(\beta) \right).$$

The notion of Radamacher complexity has been very effective for establishing generalization error. To begin with, Radamacher complexity of a set $A \subset \mathbb{R}^n$ is defined as

$$R(A) = \mathbb{E}_\sigma \left[ \sup_{a \in A} \frac{1}{n} \sum_{i=1}^{n} \sigma_i a_i \right],$$

where $\sigma_1, \ldots, \sigma_n$ are i.i.d. Radamacher variables with uniform distribution of $\{-1, 1\}$ and the expectation above is with respect to their randomness. This has been naturally extended for setting of

prediction problem as follows: given a collection of real-valued response variables and covariates, say $(Y_i, X_i)$, $i \in [n]$, collection of real-valued functions or hypothesis $\mathcal{G}$ that map covariates to real values, and loss function $L : \mathbb{R}^2 \to [0, \infty)$ that measures the error or loss in prediction for a given function, define

$$R_S(\mathcal{G}) = \mathbb{E}_\sigma \left[ \sup_{g \in \mathcal{G}} \frac{1}{n} \sum_{i=1}^n \sigma_i g(X_i) \right], \quad R_S(L \circ \mathcal{G}) = \mathbb{E}_\sigma \left[ \sup_{g \in \mathcal{G}} \frac{1}{n} \sum_{i=1}^n \sigma_i L(Y_i, g(X_i)) \right].$$

In our setting, the covariates that predictor uses are de-noised rows of $\widehat{A}$, defined as $\widehat{\mathcal{A}} = \{\widehat{A}_{1,.}, \ldots, \widehat{A}_{N,.}\}$. We use linear functions as predictors with hypothesis or function classes of interest are $\mathcal{F}_B^\infty$, $\mathcal{F}_r^0$. The loss function of interest is quadratic function: $\ell(y, y') = (y - y')^2$. The ideal response variable of our interest are $A_{i,.}\beta^*$ for $i \in [N]$. Given that, our algorithm observes (noisy) response variables in the index set $\Omega$, we shall use the sample set $\{(A_{\omega,.}\beta^*, \widehat{A}_{\omega,.}) : \omega \in \Omega\}$. It turns out that the appropriate adaptation of the Radamacher complexity for our setting is as follows:

$$R_n(\mathcal{F}) = \mathbb{E}_{\sigma, \Omega} \left[ \sup_{\beta \in \mathcal{F}} \left( \frac{1}{n} \sum_{\omega \in \Omega} \sigma_\omega \widehat{A}_{\omega,.}\beta \right) \right], \quad R_n(\ell \circ \mathcal{F}) = \mathbb{E}_{\sigma, \Omega} \left[ \sup_{\beta \in \mathcal{F}} \left( \frac{1}{n} \sum_{\omega \in \Omega} \sigma_\omega \ell(A_{\omega,.}\beta^*, \widehat{A}_{\omega,.}\beta) \right) \right],$$

where $\mathbb{E}_\Omega$ represents average with respect to uniformly at random selection of $\Omega \subset [N]$ of size $n$.

**High probability event.** We define the following event:

$$\mathcal{E}_5 = \left\{ \phi(\Omega) \leq \mathbb{E}_\Omega \left[\phi(\Omega)\right] + \sqrt{\frac{8 \cdot C(\widehat{A}) \cdot \log(Np)}{n}} \right\},$$

where $C(\widehat{A}) = 2 \left[ (rB \cdot \|\widehat{A}\|_{\max})^2 + (\Gamma\|\beta^*\|_1)^2 \right]$.

**Helpful Lemmas.** We now state a series of lemmas that will help us prove Theorem 4.2.

**Lemma K.1** *Let Property 2.1 hold. Then, for any $\beta \in \mathcal{F}$,*

$$\max_{i \in [N]} \ell(A_{i,.}\beta^*, \widehat{A}_{i,.}\beta) \leq C(\widehat{A}).$$

**Lemma K.2** *Let Properties 2.1 and 2.5 hold. Then,*

$$\mathbb{P}(\mathcal{E}_5^c) \leq \frac{2}{N^{10}p^{10}}.$$

**Lemma K.3** *Let $\phi(\Omega)$ be defined as in (51). Let $\Omega$ be random subset of $[N]$ of size $n$ that is chosen uniformly at random. Then,*

$$\mathbb{E}_\Omega \left[\phi(\Omega)\right] \leq 2R_n(\ell \circ \mathcal{F}).$$

**Lemma K.4** *Let $\operatorname{rank}(\widehat{A}) = r$. Then,*

$$R_n(\mathcal{F}) \leq \frac{rB}{\sqrt{n}} \cdot \|\widehat{A}\|_{\max}.$$

**Lemma K.5  Lipschitz composition of Rademacher averages.**
*Suppose $\{\phi_i\}, \{\psi_i\}$, $i = 1, \ldots, n$, are two sets of functions on $\Theta$ such that for each $i$ and $\theta, \theta' \in \Theta$, $|\phi_i(\theta) - \phi_i(\theta')| \leq |\psi_i(\theta) - \psi_i(\theta')|$. Then, for all functions $c : \Theta \to \mathbb{R}$,*

$$\mathbb{E} \left[ \sup_{\theta \in \Theta} \left\{ c(\theta) + \sum_{i=1}^n \sigma_i \phi_i(\theta) \right\} \right] \leq \mathbb{E} \left[ \sup_{\theta \in \Theta} \left\{ c(\theta) + \sum_{i=1}^n \sigma_i \psi_i(\theta) \right\} \right],$$

*where $\sigma_i$ are Rademacher random variables.*

**Completing The Proof of Theorem 4.2.**    Now we are ready to complete the proof of Theorem 4.2.

**Proof K.1** *The testing error for PCR with parameter $r$ or equivalent Linear Regression with covariate pre-processing using HSVT with thresholding done at $r$th singular value, is*

$$MSE(\widehat{Y}) = \frac{1}{N}\mathbb{E}\left[\sum_{i=1}^{N}\left(\widehat{Y}_i - \boldsymbol{A}_{i,\cdot}\beta^*\right)^2\right] = \mathbb{E}\left[\mathcal{E}(\widehat{\beta})\right].$$

*And, for a given training set $\Omega$, the training error is*

$$MSE_\Omega(\widehat{Y}) = \frac{1}{|\Omega|}\mathbb{E}\left[\sum_{i\in\Omega}\left(\widehat{Y}_i - \boldsymbol{A}_{i,\cdot}\beta^*\right)^2\right] = \mathbb{E}\left[\mathcal{E}_\Omega(\widehat{\beta})\right]. \tag{46}$$

*We shall consider the training set $\Omega$ being chosen uniformly at random amongst subset of $[N]$ of size $n$. Our interest is in bounding $\mathbb{E}\left[\mathcal{E}(\widehat{\beta})\right]$ in terms of $\mathbb{E}_\Omega\mathbb{E}\left[\mathcal{E}_\Omega(\widehat{\beta})\right]$ where randomness in the data generation as well as $\Omega$. Let $E \triangleq \mathcal{E}_1 \cap \mathcal{E}_2 \cap \mathcal{E}_3 \cap \mathcal{E}_4 \cap \mathcal{E}_5$. Then,*

$$\mathbb{E}\left[\mathcal{E}(\widehat{\beta})\right] = \mathbb{E}\left[\mathcal{E}(\widehat{\beta}) \cdot \mathbb{1}(E)\right] + \mathbb{E}\left[\mathcal{E}(\widehat{\beta}) \cdot \mathbb{1}(E^c)\right]. \tag{47}$$

*We will bound each term on the right-hand side of (47) separately.*

**Upper bound on first term in** (47) **.** *Given any $\Omega$, observe that*

$$\mathcal{E}(\widehat{\beta}) \leq \widehat{\mathcal{E}}_\Omega(\widehat{\beta}) + \sup_{\beta\in\mathcal{F}}\left(\mathcal{E}(\beta) - \widehat{\mathcal{E}}_\Omega(\beta)\right) = \widehat{\mathcal{E}}_\Omega(\widehat{\beta}) + \phi(\Omega). \tag{48}$$

*Further, under $E$ (and hence $\mathcal{E}_5$),*

$$\phi(\Omega) \leq \mathbb{E}_\Omega\left[\phi(\Omega)\right] + \sqrt{\frac{8 \cdot C(\widehat{\boldsymbol{A}}) \cdot \log(Np)}{n}} \leq 2R_n(\ell \circ \mathcal{F}) + \sqrt{\frac{8 \cdot C(\widehat{\boldsymbol{A}}) \cdot \log(Np)}{n}},$$

*where the second inequality follows from Lemma K.3. Using Lemma K.1, we have for any $\beta \in \mathcal{F}$,*

$$\max_{i\in[N]}|\ell'(\boldsymbol{A}_{i,\cdot}\beta^*, \widehat{\boldsymbol{A}}_{i,\cdot}\beta)| \leq 2\sqrt{C(\widehat{\boldsymbol{A}})},$$

*where $\ell'(\cdot,\cdot)$ denotes the derivative of the loss function with respect to our estimate. Since our loss function of interest has bounded first derivative, the Lipschitz constant of $\ell(\cdot,\cdot)$ is bounded by $2C(\widehat{\boldsymbol{A}})^{1/2}$; hence, applying a corollary of Lemma K.5 for Lipschitz functions and using Lemma K.4 yields the following inequality:*

$$R_n(\ell \circ \mathcal{F}) \leq 2\sqrt{C(\widehat{\boldsymbol{A}})} \cdot R_n(\mathcal{F}) \leq 2rB \cdot \sqrt{\frac{C(\widehat{\boldsymbol{A}})}{n}} \cdot \|\widehat{\boldsymbol{A}}\|_{\max}.$$

*Plugging the above results into (48), we obtain*

$$\mathbb{E}_\Omega\mathbb{E}\left[\mathcal{E}(\widehat{\beta}) \cdot \mathbb{1}(E)\right] \leq \mathbb{E}_\Omega\mathbb{E}\left[\widehat{\mathcal{E}}_\Omega(\widehat{\beta}) \cdot \mathbb{1}(E)\right] + \mathbb{E}_\Omega\mathbb{E}\left[\phi(\Omega) \cdot \mathbb{1}(E)\right]$$

$$\leq \mathbb{E}_\Omega\mathbb{E}\left[\widehat{\mathcal{E}}_\Omega(\widehat{\beta})\right] + \mathbb{E}\left[4rB \cdot \sqrt{\frac{C(\widehat{\boldsymbol{A}})}{n}} \cdot \|\widehat{\boldsymbol{A}}\|_{\max} + 2\sqrt{2} \cdot \sqrt{\frac{C(\widehat{\boldsymbol{A}}) \cdot \log(Np)}{n}}\right]$$

$$= \mathbb{E}_\Omega\mathbb{E}\left[\widehat{\mathcal{E}}_\Omega(\widehat{\beta})\right] + \frac{4\sqrt{2} \cdot (rB)^2}{\sqrt{n}} \cdot \mathbb{E}\left[\|\widehat{\boldsymbol{A}}\|_{\max}^2\right] + \frac{4rB\Gamma\|\beta^*\|_1}{\sqrt{n}} \cdot \mathbb{E}\left[\|\widehat{\boldsymbol{A}}\|_{\max}\right]$$

$$+ 4\left(rB \cdot \mathbb{E}\left[\|\widehat{\boldsymbol{A}}\|_{\max}\right] + \Gamma\|\beta^*\|_1\right) \cdot \sqrt{\frac{\log(Np)}{n}}.$$

*By (46), the first term on the right-hand side of the above corresponds to $\mathbb{E}_\Omega[MSE_\Omega(\widehat{Y})]$; hence,*

$$\mathbb{E}\left[\mathcal{E}(\widehat{\beta}) \cdot \mathbb{1}(E)\right] \leq \mathbb{E}_\Omega[MSE_\Omega(\widehat{Y})] + C_1 \cdot (rB)^2 \cdot \Gamma\|\beta^*\|_1 \cdot \mathbb{E}\left[\|\widehat{\boldsymbol{A}}\|_{\max}^2\right] \cdot \sqrt{\frac{\log(Np)}{n}}, \tag{49}$$

*where $C_1$ is a universal positive constant.*

**Upper bound on second term in** (47) **.** *We begin with the following trivial bound on the expected prediction error: since $\widehat{\beta} \in \mathcal{F}$, Lemma K.1 gives*

$$\mathcal{E}(\widehat{\beta}) = \frac{1}{N} \sum_{i=1}^{N} \left( \widehat{\boldsymbol{A}}_{i,\cdot}\widehat{\beta} - \boldsymbol{A}_{i,\cdot}\beta^* \right)^2 \leq \max_{i \in [N]} \left( \widehat{\boldsymbol{A}}_{i,\cdot}\widehat{\beta} - \boldsymbol{A}_{i,\cdot}\beta^* \right)^2 \leq C(\widehat{\boldsymbol{A}}).$$

*Further, by a simple application of DeMorgan's Law and the union bound, we have*

$$\mathbb{P}(E^c) \leq \sum_{q=1}^{5} \mathbb{P}(\mathcal{E}_q^c) \leq \frac{9}{N^{10}p^{10}}.$$

*By Cauchy-Schwarz inequality, the following inequality holds:*

$$\mathbb{E}\left[ \mathcal{E}(\widehat{\beta}) \cdot \mathbb{1}(E^c) \right]^2 \leq \mathbb{E}\left[ \mathcal{E}(\widehat{\beta})^2 \right] \cdot \mathbb{E}\left[ \mathbb{1}(E^c) \right] = \mathbb{E}\left[ \mathcal{E}(\widehat{\beta})^2 \right] \cdot \mathbb{P}(E^c).$$

*Putting everything together yields*

$$\begin{aligned}
\mathbb{E}\left[ \mathcal{E}(\widehat{\beta}) \cdot \mathbb{1}(E^c) \right] &\leq \mathbb{E}\left[ C(\widehat{\boldsymbol{A}})^2 \right]^{1/2} \cdot \mathbb{P}(E^c)^{1/2} \\
&\leq 2\sqrt{2} \left( (rB)^4 \cdot \mathbb{E}\left[ \|\widehat{\boldsymbol{A}}\|_{\max}^4 \right] + (\Gamma\|\beta^*\|_1)^4 \right)^{1/2} \cdot \mathbb{P}(E^c)^{1/2} \\
&\leq 2\sqrt{2} \left( (rB)^2 \cdot \mathbb{E}\left[ \|\widehat{\boldsymbol{A}}\|_{\max}^4 \right]^{1/2} + (\Gamma\|\beta^*\|_1)^2 \right) \cdot \mathbb{P}(E^c)^{1/2} \\
&\leq C_2 \left( (rB)^2 \cdot \mathbb{E}\left[ \|\widehat{\boldsymbol{A}}\|_{\max}^4 \right]^{1/2} + (\Gamma\|\beta^*\|_1)^2 \right) \cdot \frac{1}{N^5p^5},
\end{aligned} \tag{50}$$

*where $C_2$ is an absolute constant.*

**Concluding the proof.** *Plugging* (49) *and* (50) *into* (47) *gives the following bound:*

$$\begin{aligned}
MSE(\widehat{Y}) &\leq \mathbb{E}_{\Omega}[MSE_{\Omega}(\widehat{Y})] + C_1 \cdot (rB)^2 \cdot \Gamma\|\beta^*\|_1 \cdot \mathbb{E}\left[ \|\widehat{\boldsymbol{A}}\|_{\max}^2 \right] \cdot \sqrt{\frac{\log(Np)}{n}} \\
&\quad + C_2 \left( (rB)^2 \cdot \mathbb{E}\left[ \|\widehat{\boldsymbol{A}}\|_{\max}^4 \right]^{1/2} + (\Gamma\|\beta^*\|_1)^2 \right) \cdot \frac{1}{N^5p^5}.
\end{aligned}$$

*Let $C_3 = C_2 B^2 \Gamma\|\beta^*\|_1$. Then,*

$$MSE(\widehat{Y}) \leq \mathbb{E}_{\Omega}[MSE_{\Omega}(\widehat{Y})] + C_3 \cdot r^2 \cdot \mathbb{E}\left[ \|\widehat{\boldsymbol{A}}\|_{\max}^2 \right] \cdot \sqrt{\frac{\log(Np)}{n}}.$$

*This completes the proof.*

**Lemma K.6** *Let $\Omega = \{i_1, \dots, i_n\}$ and $\Omega' = \{i_1, \dots, i'_j, \dots, i_n\}$ such that $\Omega$ and $\Omega'$ differ only in their $j$-th elements. Let*

$$\phi(\Omega) = \sup_{\beta \in \mathcal{F}} \left( \mathcal{E}(\beta) - \widehat{\mathcal{E}}_{\Omega}(\beta) \right). \tag{51}$$

*Then for any $\beta \in \mathcal{F}$,*

$$|\phi(\Omega) - \phi(\Omega')| \leq \frac{C(\widehat{\boldsymbol{A}})}{n}.$$

**Proof K.2** *Here, we will show that*

$$\phi(\Omega) = \sup_{\beta \in \mathcal{F}} \left( \mathcal{E}(\beta) - \widehat{\mathcal{E}}_{\Omega}(\beta) \right)$$

*satisfies the conditions necessary to invoke McDiarmid's Inequality. We begin by noting that for any*
*real-valued functions $f_1, f_2$, $\sup_x f_1(x) - \sup_x f_2(x) \leq \sup_x (f_1(x) - f_2(x))$. Hence,*

$$
\begin{aligned}
\phi(\Omega) - \phi(\Omega') &= \sup_{\beta \in \mathcal{F}} \left( \mathcal{E}(\beta) - \widehat{\mathcal{E}}_\Omega(\beta) \right) - \sup_{\beta \in \mathcal{F}} \left( \mathcal{E}(\beta) - \widehat{\mathcal{E}}_{\Omega'}(\beta) \right) \\
&\leq \sup_{\beta \in \mathcal{F}} \left( \mathcal{E}(\beta) - \widehat{\mathcal{E}}_\Omega(\beta) - \mathcal{E}(\beta) + \widehat{\mathcal{E}}_{\Omega'}(\beta) \right) \\
&= \sup_{\beta \in \mathcal{F}} \left( \widehat{\mathcal{E}}_{\Omega'}(\beta) - \widehat{\mathcal{E}}_\Omega(\beta) \right) \leq \frac{C(\widehat{\boldsymbol{A}})}{n},
\end{aligned}
$$

*where the final equality follows from Lemma K.1 since $\Omega$ and $\Omega'$ differ by only one element. Using*
*a similar argument, we can prove that $\phi(\Omega') - \phi(\Omega) \leq C(\widehat{\boldsymbol{A}})/n$, and therefore $|\phi(\Omega) - \phi(\Omega')| \leq$*
*$C(\widehat{\boldsymbol{A}})/n$.*

## K.1 Proof of Lemma K.1

**Proof K.3** *Observe that for any $i \in [N]$ and $\beta \in \mathcal{F}$,*

$$
\ell(\boldsymbol{A}_{i,.}\beta^*, \widehat{\boldsymbol{A}}_{i,.}\beta) = (\widehat{\boldsymbol{A}}_{i,.}\beta - \boldsymbol{A}_{i,.}\beta^*)^2 \leq 2(\widehat{\boldsymbol{A}}_{i,.}\beta)^2 + 2(\boldsymbol{A}_{i,.}\beta^*)^2.
$$

*Recall that every candidate vector $\beta \in \mathcal{F}$ has the following properties: $\|\beta\|_0 \leq r$ and $\|\beta\|_2 \leq B$.*
*Hence, it follows that for any $i \in [N]$,*

$$
|\widehat{\boldsymbol{A}}_{i,.}\widehat{\beta}| \leq r \cdot \|\beta\|_\infty \cdot \max_{j \in [p]} |\widehat{A}_{ij}| \ \leq r \cdot \|\beta\|_2 \cdot \|\widehat{\boldsymbol{A}}\|_{\max} \ \leq rB \cdot \|\widehat{\boldsymbol{A}}\|_{\max}.
$$

*Further, By Property 2.1 and Holder's inequality, we have for any $i \in [N]$,*

$$
|\boldsymbol{A}_{i,.}\beta^*| \leq \|\boldsymbol{A}_{i,.}\|_\infty \|\beta^*\|_1 \leq \Gamma \|\beta^*\|_1.
$$

*The desired result then follows from an immediate application of the above results.*

## K.2 Proof of Lemma K.2

**Proof K.4** *By Lemma K.1, we know that for any $i \in [N]$ and $\beta \in \mathcal{F}$, $\ell(\boldsymbol{A}_{i,.}\beta^*, \widehat{\boldsymbol{A}}_{i,.}\beta) \in [0, C(\widehat{\boldsymbol{A}})]$.*
*Lemma K.6 then allows us to apply McDiarmid's Inequality (Lemma D.4), which gives*

$$
\mathbb{P}\{\phi(\Omega) - \mathbb{E}_\Omega \phi(\Omega) \geq t_1\} \leq \exp\left( -t_1^2 n / C(\widehat{\boldsymbol{A}}) \right).
$$

*Setting $t_1 = \sqrt{10 \cdot C(\widehat{\boldsymbol{A}}) \cdot \log(Np)/n}$ completes the proof.*

## K.3 Proof of Lemma K.3

**Proof K.5** *Let $\Omega' = \{i'_1, \ldots, i'_n\}$ be a "ghost sample", i.e., $\Omega'$ is an independent set of $n$ locations*
*sampled uniformly at random and without replacement from $[N]$. Observe that $\mathcal{E}(\beta) = \mathbb{E}_{\Omega'}[\widehat{\mathcal{E}}_{\Omega'}(\beta)]$*
*and $\widehat{\mathcal{E}}_\Omega(\beta) = \mathbb{E}_{\Omega'}[\widehat{\mathcal{E}}_\Omega(\beta)]$. Thus,*

$$
\begin{aligned}
\mathbb{E}_\Omega \phi(\Omega) &= \mathbb{E}_\Omega \left[ \sup_{\beta \in \mathcal{F}} \left( \mathcal{E}(\beta) - \widehat{\mathcal{E}}_\Omega(\beta) \right) \right] \\
&= \mathbb{E}_\Omega \left[ \sup_{\beta \in \mathcal{F}} \left( \mathbb{E}_{\Omega'} \left[ \widehat{\mathcal{E}}_{\Omega'}(\beta) - \widehat{\mathcal{E}}_\Omega(\beta) \right] \right) \right] \\
&\leq \mathbb{E}_{\Omega, \Omega'} \left[ \sup_{\beta \in \mathcal{F}} \left( \widehat{\mathcal{E}}_{\Omega'}(\beta) - \widehat{\mathcal{E}}_\Omega(\beta) \right) \right] \\
&= \mathbb{E}_{\Omega, \Omega'} \left[ \sup_{\beta \in \mathcal{F}} \frac{1}{n} \sum_{k=1}^n \left( \ell(\boldsymbol{A}_{i'_k}\beta^*; \widehat{\boldsymbol{A}}_{i'_k}\beta) - \ell(\boldsymbol{A}_{i_k}\beta^*; \widehat{\boldsymbol{A}}_{i_k}\beta) \right) \right],
\end{aligned}
$$

*where the inequality follows by the convexity of the supremum function and Jensen's Inequality.*

*To proceed, we will use the ghost sampling technique. Recall that the entries of $\Omega$ and $\Omega'$ were drawn uniformly at random from $[N]$. As a result, $\ell(\boldsymbol{A}_{i'_k}\beta^*; \widehat{\boldsymbol{A}}_{i'_k}\beta) - \ell(\boldsymbol{A}_{i_k}\beta^*; \widehat{\boldsymbol{A}}_{i_k}\beta)$ and $\ell(\boldsymbol{A}_{i_k}\beta^*; \widehat{\boldsymbol{A}}_{i_k}\beta) - \ell(\boldsymbol{A}_{i'_k}\beta^*; \widehat{\boldsymbol{A}}_{i'_k}\beta)$ have the same distribution. Further, since $\sigma_k$ takes value $1$ and $-1$ with equal probability, we have*

$$\mathbb{E}_{\Omega,\Omega'}\left[\sup_{\beta\in\mathcal{F}}\frac{1}{n}\sum_{k=1}^{n}\left(\ell(\boldsymbol{A}_{i'_k}\beta^*; \widehat{\boldsymbol{A}}_{i'_k}\beta) - \ell(\boldsymbol{A}_{i_k}\beta^*; \widehat{\boldsymbol{A}}_{i_k}\beta)\right)\right]$$

$$= \mathbb{E}_{\sigma,\Omega,\Omega'}\left[\sup_{\beta\in\mathcal{F}}\frac{1}{n}\sum_{k=1}^{n}\sigma_k\left(\ell(\boldsymbol{A}_{i'_k}\beta^*; \widehat{\boldsymbol{A}}_{i'_k}\beta) - \ell(\boldsymbol{A}_{i_k}\beta^*; \widehat{\boldsymbol{A}}_{i_k}\beta))\right)\right].$$

*Combining the above relation with the fact that the supremum of a sum is bounded above by the sum of supremums, we obtain*

$$\mathbb{E}_{\Omega}\phi(\Omega) \leq \mathbb{E}_{\sigma,\Omega,\Omega'}\left[\sup_{\beta\in\mathcal{F}}\frac{1}{n}\sum_{k=1}^{n}\sigma_k\left(\ell(\boldsymbol{A}_{i'_k}\beta^*; \widehat{\boldsymbol{A}}_{i'_k}\beta) - \ell(\boldsymbol{A}_{i_k}\beta^*; \widehat{\boldsymbol{A}}_{i_k}\beta)\right)\right]$$

$$\leq \mathbb{E}_{\sigma,\Omega,\Omega'}\left[\sup_{\beta\in\mathcal{F}}\frac{1}{n}\sum_{k=1}^{n}\sigma_k\ell(\boldsymbol{A}_{i'_k}\beta^*; \widehat{\boldsymbol{A}}_{i'_k}\beta) + \sup_{\beta\in\mathcal{F}}\frac{1}{n}\sum_{k=1}^{n}-\sigma_k\ell(\boldsymbol{A}_{i_k}\beta^*; \widehat{\boldsymbol{A}}_{i_k}\beta)\right]$$

$$= \mathbb{E}_{\sigma,\Omega}\left[\sup_{\beta\in\mathcal{F}}\frac{1}{n}\sum_{k=1}^{n}\sigma_k\ell(\boldsymbol{A}_{i_k}\beta^*; \widehat{\boldsymbol{A}}_{i_k}\beta)\right] + \mathbb{E}_{\sigma,\Omega'}\left[\sup_{\beta\in\mathcal{F}}\frac{1}{n}\sum_{k=1}^{n}\sigma_k\ell(\boldsymbol{A}_{i'_k}\beta^*; \widehat{\boldsymbol{A}}_{i'_k}\beta)\right]$$

$$= 2\cdot R_n(\ell\circ\mathcal{F}),$$

*where the second to last equality holds because $\sigma_k$ is a symmetric random variable.*

## K.4 Proof of Lemma K.4

**Proof K.6** *Let $I_\beta = \{i\in[p] : \beta_i\neq 0\}$ denote the index set for the nonzero elements of $\beta\in\mathbb{R}^p$. For any vector $v\in\mathbb{R}^p$, we denote $v_{I_\beta}$ as the vector that retains only its values in $I_\beta$ and takes the value $0$ otherwise. Then,*

$$R_n(\mathcal{F}) = \mathbb{E}_{\sigma,\Omega}\left[\sup_{\beta\in\mathcal{F}}\left(\frac{1}{n}\sum_{i=1}^{n}\sigma_i\langle\alpha_i,\beta\rangle\right)\right]$$

$$= \frac{1}{n}\mathbb{E}_{\sigma,\Omega}\left[\sup_{\beta\in\mathcal{F}}\left(\langle\sum_{i=1}^{n}\sigma_i\alpha_i,\beta\rangle\right)\right]$$

$$= \frac{1}{n}\mathbb{E}_{\sigma,\Omega}\left[\sup_{\beta\in\mathcal{F}}\left(\sum_{j\in I_\beta}\beta_j\left(\sum_{i=1}^{n}\sigma_i\alpha_i\right)_j\right)\right]$$

$$\overset{(a)}{\leq} \frac{1}{n}\mathbb{E}_{\sigma,\Omega}\left[\sup_{\beta\in\mathcal{F}}\left\|\beta_{I_\beta}\right\|_2\cdot\left\|\left(\sum_{i=1}^{n}\sigma_i\alpha_i\right)_{I_\beta}\right\|_2\right]$$

$$\overset{(b)}{\leq} \frac{\sqrt{r}B}{n}\mathbb{E}_{\sigma,\Omega}\left[\left\|\left(\sum_{i=1}^{n}\sigma_i\alpha_i\right)_{I_\beta}\right\|_2\right]$$

$$\overset{(c)}{\leq} \frac{\sqrt{r}B}{n}\left(\mathbb{E}_{\sigma,\Omega}\left[\left(\sum_{i=1}^{n}\sigma_i\alpha_i\right)_{I_\beta}^T\left(\sum_{k=1}^{n}\sigma_k\alpha_k\right)_{I_\beta}\right]\right)^{1/2}$$

$$= \frac{\sqrt{r}B}{n}\left(\mathbb{E}_{\Omega}\left[\sum_{i=1}^{n}\left\|(\alpha_i)_{I_\beta}\right\|_2^2\right]\right)^{1/2}$$

$$\leq \frac{\sqrt{r}B}{n}\left(nr\max_{i\in[n]}\left\|(\alpha_i)_{I_\beta}\right\|_\infty^2\right)^{1/2}$$

$$= \frac{rB}{\sqrt{n}} \cdot \max_{i \in [n]} \|\alpha_i\|_\infty.$$

*Note that (a) makes use of the Cauchy-Schwartz Inequality, (b) follows from the boundedness assumption in the Lemma statement, and (c) applies Jensen's Inequality. The proof is complete after observing that $\|\widehat{A}\|_{\max} = \max_{i \in [n]} \|\alpha_i\|_\infty$.*

## L  Examples

### L.1  Embedded Random Gaussian Features.

**Analysis for the Example.**  In this subsection, we show that $s_r(A) = \Omega(\sqrt{N})$ and $\Gamma = O\left(\sqrt{\frac{r \log(Np)}{p}}\right)$ with high probability.

**Lemma L.1** *Suppose that $r \le \frac{\sqrt{p}}{4\sqrt{2}\log p} + 1$ and let $R \in \mathbb{R}^{r \times p}$ be a random matrix with independent entries such that $R_{ij} = \frac{1}{\sqrt{p}}$ with probability $\frac{1}{2}$ and $R_{ij} = -\frac{1}{\sqrt{p}}$ with probability $\frac{1}{2}$. With probability at least $1 - \frac{1}{p^2}$, for all $v \in \mathbb{R}^r$,*

$$\frac{1}{2}\|v\|_2^2 \le \|vR\|_2^2 \le \frac{3}{2}\|v\|_2^2.$$

**Remark L.1** *Lemma L.1 implies that given $r \le 1 + \frac{\sqrt{p}}{4\sqrt{2}\log p}$, the right multiplication of $R$ defines a quasi-isometric embedding from $\mathbb{R}^r$ to $\mathbb{R}^p$ with high probability. More precisely, with probability at least $1 - \frac{1}{p^2}$, the following inequalities are true:*

$$\frac{1}{2}\|v\|_2^2 \le \|vR\|_2^2 \le \frac{3}{2}\|v\|_2^2, \quad \forall v \in \mathbb{R}^r, \qquad and \qquad \frac{2}{3}\|w\|_2^2 \le \|Rw\|_2^2 \le 2\|w\|_2^2, \quad \forall w \in rowspan(R).$$

**Remark L.2** *By Remark L.1, with probability at least $1 - \frac{1}{p^2}$,*

$$s_r(A) = \sup_{\substack{W \subset \mathbb{R}^p \\ \dim W = r}} \inf_{w \in W} \frac{\|Aw\|_2}{\|w\|_2} = \sup_{\substack{W \subset \mathbb{R}^p \\ \dim W = r}} \inf_{w \in W} \frac{\|\tilde{A}Rw\|_2}{\|w\|_2} = \inf_{w \in \text{rowspace} R} \frac{\|\tilde{A}Rw\|_2}{\|w\|_2}$$

$$\ge \sqrt{\frac{2}{3}} \inf_{w \in \text{rowspace} R} \frac{\|\tilde{A}Rw\|_2}{\|Rw\|_2} = \sqrt{\frac{2}{3}} \inf_{v \in \mathbb{R}^r} \frac{\|\tilde{A}v\|_2}{\|v\|_2} = \sqrt{\frac{2}{3}} s_r(\tilde{A}).$$

**Lemma L.2 (Spectral properties of $\tilde{A}$)** *Let $\tilde{A} \in \mathbb{R}^{N \times r}$ be a random matrix whose entries are i.i.d. standard Gaussian random variable.*

1. *With probability at least $1 - 2\exp\left(-2\sqrt{Nr}\right)$, $\text{rank}(\tilde{A}) = r$ and*

$$\frac{s_1(\tilde{A})}{s_r(\tilde{A})} \le \left(\frac{N^{1/4} + r^{1/4}}{N^{1/4} - r^{1/4}}\right)^2.$$

2. *With probability at least $1 - \exp\left(-\frac{Nr}{8}\right)$,*

$$\|\tilde{A}\|_F^2 > \frac{Nr}{2}.$$

**Remark L.3** *Lemma L.2 implies that with probability at least $1 - 2\exp\left(-2\sqrt{Nr}\right) - \exp\left(-\frac{Nr}{8}\right)$,*

$$s_r(\tilde{A})^2 \ge \left[1 + (r-1)\frac{s_1(\tilde{A})^2}{s_r(\tilde{A})^2}\right]^{-1} \|\tilde{A}\|_F^2 \ge \left[1 + (r-1)\left(\frac{N^{1/4} + r^{1/4}}{N^{1/4} - r^{1/4}}\right)^4\right]^{-1} \frac{Nr}{2}.$$

**Lemma L.3 (Structural properties of $A$)** *Let $A \in \mathbb{R}^{N \times p}$ be a matrix generated as above. With probability at least $1 - \frac{2}{N^2 p}$,*

$$\max_{i,j} |A_{ij}| \le 4\sqrt{\frac{r \log(Np)}{p}}.$$

**Proof of Proposition 4.2**

**Proof L.1** *Proof is immediate from Lemmas L.1, L.2, L.3 and Theorem 4.1 (along with Remarks L.2 and L.3).*

### L.1.1  Proof of Lemma L.1

**Proof L.2** *For $i \in [r]$, let $\boldsymbol{R}_i$ denote the $i$-th row of $\boldsymbol{R}$. Observe that $\|\boldsymbol{R}_i\|_2 = 1$ for all $i \in [r]$. Also, note that for $i \neq j \in [r]$, $\langle \boldsymbol{R}_i, \boldsymbol{R}_j \rangle = \frac{1}{p} \sum_{k=1}^{p} \boldsymbol{R}_{ik} \boldsymbol{R}_{jk}$ is a sum of $p$ independent binary random variables; $\boldsymbol{R}_{ik} \boldsymbol{R}_{jk} = 1$ with probability $\frac{1}{2}$ and $-1$ with probability $\frac{1}{2}$. Therefore, $\mathbb{E} \langle \boldsymbol{R}_i, \boldsymbol{R}_j \rangle = 0$. By Hoeffding's inequality for bounded random variables,*

$$\mathbb{P}\left( |\langle \boldsymbol{R}_i, \boldsymbol{R}_j \rangle| > t \right) \leq 2 \exp\left( -\frac{pt^2}{2} \right).$$

*Letting $t = \frac{2\sqrt{2 \log p}}{\sqrt{p}}$, we can conclude that for any pair of $i \neq j \in [r]$, $|\langle \boldsymbol{R}_i, \boldsymbol{R}_j \rangle| \leq \frac{2\sqrt{2 \log p}}{\sqrt{p}}$ with probability at least $1 - \frac{2}{p^4}$. There are $\binom{r}{2} \leq \frac{r^2}{2}$ such pairs and $r \leq p$. Thus, applying the union bound, we know that $|\langle \boldsymbol{R}_i, \boldsymbol{R}_j \rangle| \leq \frac{2\sqrt{2 \log p}}{\sqrt{p}}$ for all pairs $i \neq j$ with probability at least $1 - \frac{1}{p^2}$.*

*Now we observe that*

$$
\begin{aligned}
\|v\boldsymbol{R}\|_2^2 &= \left\langle \sum_{i=1}^{r} v_i \boldsymbol{R}_i, \sum_{i=1}^{r} v_i \boldsymbol{R}_i, \right\rangle \\
&= \sum_{i=1}^{r} v_i^2 \|\boldsymbol{R}_i\|_2^2 + \sum_{i=1}^{r} \sum_{j \neq i} v_i v_j \langle \boldsymbol{R}_i, \boldsymbol{R}_j \rangle \\
&\leq \sum_{i=1}^{r} v_i^2 \|\boldsymbol{R}_i\|_2^2 + \sum_{i=1}^{r} \sum_{j \neq i} |v_i v_j| |\langle \boldsymbol{R}_i, \boldsymbol{R}_j \rangle|.
\end{aligned}
$$

*With probability at least $1 - \frac{1}{p^2}$,*

$$
\begin{aligned}
\|v\boldsymbol{R}\|_2^2 &\leq \sum_{i=1}^{r} v_i^2 \|\boldsymbol{R}_i\|_2^2 + \sum_{i=1}^{r} \sum_{j \neq i} |v_i v_j| \frac{2\sqrt{2 \log p}}{\sqrt{p}} \\
&\overset{(a)}{\leq} \sum_{i=1}^{r} v_i^2 + (r-1) \sum_{i=1}^{r} v_i^2 \frac{2\sqrt{2 \log p}}{\sqrt{p}} \\
&\leq \|v\|_2^2 \left( 1 + \frac{2(r-1)\sqrt{2 \log p}}{\sqrt{p}} \right)
\end{aligned}
$$

*where (a) follows from that $\|\boldsymbol{R}_i\|_2^2 = 1$ for all $i \in [r]$ and the Cauchy-Schwarz inequality ($2|v_i v_j| \leq v_i^2 + v_j^2$). By the same argument, $\|v\boldsymbol{R}\|_2^2 \geq \|v\|_2^2 \left( 1 - \frac{2(r-1)\sqrt{2 \log p}}{\sqrt{p}} \right)$.*

*Lastly, we note that $\frac{2(r-1)\sqrt{2 \log p}}{\sqrt{p}} \leq \frac{1}{2}$ if and only if $r \leq \frac{\sqrt{p}}{4\sqrt{2 \log p}} + 1$ to complete the proof.*

### L.1.2  Proof of Lemma L.2

**Proof L.3** *Since $u_1, \ldots, u_r$ are orthonormal, the row rank of $U$ is $r$. Thus the column rank of $u$ is also $r$. By the generation process described above, $\|\boldsymbol{A}x\|_2 = \|\tilde{\boldsymbol{A}}Ux\|_2$ and we can observe that $\sigma_k(\boldsymbol{A}) = \sigma_k(\tilde{\boldsymbol{A}})$ for all $k \in [r]$.*

**Proof of Claim 1**  *By [51, Corollary 5.35], for any $t \geq 0$, we have*

$$\sqrt{N} - \sqrt{r} - t \leq s_{min}(\tilde{\boldsymbol{A}}) \leq s_{min}(\tilde{\boldsymbol{A}}) \leq \sqrt{N} + \sqrt{r} + t,$$

*with probability at least $1 - 2\exp\left( -t^2/2 \right)$. Choosing $t = 2(Nr)^{1/4}$ concludes the proof.*

**Proof of Claim 2** *Observe that $\|\tilde{\boldsymbol{A}}\|_F^2 = \sum_{i,j} \tilde{\boldsymbol{A}}_{ij}^2$. We can easily observe that $\mathbb{E}\|\tilde{\boldsymbol{A}}\|_F^2 = Nr$. By Bernstein's inequality, it follows that for every $t \geq 0$,*

$$\mathbb{P}\{\|\tilde{\boldsymbol{A}}\|_F^2 - \mathbb{E}\|\tilde{\boldsymbol{A}}\|_F^2 \leq -t\} \leq \exp\left(-\frac{1}{2}\min\left\{\frac{t^2}{Nr}, t\right\}\right).$$

*With $t = \frac{Nr}{2}$, we have*

$$\mathbb{P}\{\|\tilde{\boldsymbol{A}}\|_F^2 \leq \frac{Nr}{2}\} \leq \exp\left(-\frac{Nr}{8}\right).$$

### L.1.3  Proof of Lemma L.3

**Proof L.4** *Note that $A_{ij} \sim \mathcal{N}(0, \sum_{k=1}^r R_{kj}^2)$. For each $j$, $A_{ij}$ and $A_{i'j}$ are independent. Therefore, $\mathbb{E}[\max_i |A_{ij}|] \leq 2\|\boldsymbol{R}_{\cdot,j}\|\sqrt{\log N}$. By the concentration of Lipschitz function,*

$$\mathbb{P}\left(|\max_i|A_{ij}| - \mathbb{E}[\max_i|A_{ij}|]| \geq t\right) \leq 2\exp\left(-\frac{t^2}{2\|\boldsymbol{R}_{\cdot,j}\|^2}\right).$$

*Letting $t = 2\|\boldsymbol{R}_{\cdot,j}\|\sqrt{\log(Np)}$, it follows for each $j \in [p]$ that $\mathbb{P}\left(|\max_i|A_{ij}| \geq 4\|\boldsymbol{R}_{\cdot,j}\|\sqrt{\log(Np)}\right) \leq \frac{2}{N^2p^2}$. Note that $\|\boldsymbol{R}_{\cdot,j}\| = \sqrt{\frac{r}{p}}$ for all $j \in [p]$. Taking union bound over $j \in [p]$, we achieve*

$$\max_{i,j}|A_{ij}| \leq 4\max_{j\in[p]}\|\boldsymbol{R}_{\cdot,j}\|\sqrt{\log(Np)} = 4\sqrt{\frac{r\log(Np)}{p}}$$

*with probability at least $1 - \frac{2}{N^2p}$.*

### L.2  Geometrically Decaying Singular Values.

### L.2.1  Proof of Lemma 5.2

**Proof L.5** *For $(i,j) \in [N] \times [p]$, we have $\boldsymbol{A}_{ij} = \sum_{k=1}^{N\wedge p} \Sigma_{kk}U_{ik}V_{jk}$. Thus,*

$$\begin{aligned}
|A_{ij}| &= \left|\sum_{k=1}^{N\wedge p}\Sigma_{kk}U_{ik}V_{jk}\right| \leq \sum_{k=1}^{N\wedge p}\Sigma_{kk}|U_{ik}||V_{jk}| \\
&\overset{(a)}{\leq} \sum_{k=1}^{N\wedge p}\Sigma_{11}\theta^{k-1}\frac{1}{\sqrt{Np}} = \Sigma_{11}\frac{1-\theta^{N\wedge p}}{1-\theta}\frac{1}{\sqrt{Np}} \\
&\overset{(b)}{\leq} \frac{C}{1-\theta}.
\end{aligned}$$

*Here, (a) follows from that $|U_{ik}| = \frac{1}{\sqrt{N}}$, $|V_{jk}| = \frac{1}{\sqrt{p}}$, and $\Sigma_{kk} \leq \Sigma_{11}\theta^{k-1}$; and (b) follows from the assumption $\Sigma_{11} = C\sqrt{Np}$ and that $1 - \theta^{N\wedge p} \leq 1$.*

## M  Proof of Proposition 5.1

**Proof M.1** *We wish to argue that for some constant $C_2 > 0$, PCR with parameter $k = C_2\frac{\log\log(np)}{\log(1/\theta)}$ the following bound on training error holds:*

$$\mathrm{MSE}_\Omega(\widehat{Y}) \leq \frac{2C_2\sigma^2\log\log np}{n\log(1/\theta)} + \frac{C'(\alpha,\theta)(1+\gamma+\Gamma+K_\alpha)^4\|\beta^*\|_1^2\log^{5+2C_2}np}{n\rho^4} + \frac{C''\|\beta^*\|_1^2}{\log^{2C_2}np} + 20\|\phi\|_\infty^2,$$

*where $C'(\alpha,\theta) > 0$ constant dependent on $\alpha, \theta$ and $C'' > 0$ a universal constant. We wish to utilize bound in Corollary 5.1 to derive this result. To that end, the bound of Corollary 5.1 is*

$$\mathrm{MSE}_\Omega(\widehat{Y}) \leq \frac{4\sigma^2k}{n} + \frac{C(\alpha)(1+\gamma+\Gamma+K_\alpha)^4\|\beta^*\|_1^2\log^2 np}{n\rho^2}\left(\frac{(n^2\rho+np)\log^3 np}{\rho^2(\tau_k-\tau_{k+1})^2} + k\right)$$

$$+ \frac{6\|\beta^*\|_1^2}{n}\|A^k - A\|_{2,\infty}^2 + 20\|\phi\|_\infty^2,$$

where $C(\alpha) > 0$ is a constant that may depend on $\alpha \geq 1$. Let us evaluate each of the four terms in the right hand side of (6) to reach the desired (7).

**First term.** *Due to choice of* $k = C_2 \frac{\log\log(np)}{\log(1/\theta)}$ *it immediately follows that it is*

$$\frac{4\sigma^2 k}{n} = \frac{4\sigma^2 C_2 \log\log(np)}{n \log(1/\theta)}.$$

**Second term.** *For this, consider*

$$\frac{n^2\rho + np}{\rho^2(\tau_k - \tau_{k+1})^2} \log^3(np) + k = \frac{n(n\rho + p)\log^3 np}{\rho^2\tau_1^2\theta^{2k-2}(1-\theta)^2} + k$$

$$\leq \frac{np\log^3 np}{\rho^2 C_1^2 Np \log^{2C_2} np(1-\theta)^2\theta^{-2}} + \frac{C_2\log\log np}{\log(1/\theta)}$$

$$\leq C_3(\theta)\frac{\log^{3+2C_2} np}{\rho^2}$$

where we have used the fact that $\tau_i = \tau_1\theta^{i-1}$ for $i \geq 1$, $\tau_1 = C_1\sqrt{Np}$, $n\rho \leq N \leq p$ and $C_3(\theta) > 0$ is a constant that depends on $\theta$.

**Third term.** *The goal is to bound* $\|A^k - A\|_{2,\infty}^2$. *With notation* $E = A - A^k$, *this is equivalent to bounding* $\max_{j\in[p]}\|E_{\cdot,j}\|_2^2$. *With* $A = \sum_{i=1}^N \tau_i\mu_i\nu_i^T$ *where* $\mu_i \in \mathbb{R}^N$, $\nu_i \in \mathbb{R}^p$ *for* $i \in [N]$, *for any* $j \in [p]$, *we have*

$$\frac{1}{n}\|E_{\cdot,j}\|^2 = \frac{1}{n}\left\|\left(\sum_{i=k+1}^N \tau_i\mu_i\nu_i^T\right)e_j\right\|^2 = \frac{1}{n}\left\|\sum_{i=k+1}^N \tau_i\mu_i(\nu_i^T e_j)\right\|^2$$

$$\stackrel{(a)}{=} \frac{1}{n}\sum_{i=k+1}^N \tau_i^2(\nu_i^T e_j)^2$$

$$\stackrel{(b)}{\leq} \frac{1}{n}\sum_{i=k+1}^N \tau_1^2\theta^{2(i-1)}(\nu_i^T e_j)^2$$

$$\stackrel{(c)}{\leq} \frac{C_1 Np}{n}\sum_{i=k+1}^N \theta^{2(i-1)}(\nu_i^T e_j)^2$$

$$\stackrel{(d)}{\leq} \frac{C_1 Np}{np}\sum_{i=k+1}^N \theta^{2(i-1)}$$

$$\stackrel{(e)}{\leq} C\theta^{2k} \stackrel{(f)}{\leq} \frac{C}{\log^{2C_2}(np)}$$

*Here, (a) follows from the orthonormality of the (left) singular vectors; (b) follows from* $\tau_i = \tau_1\theta^{i-1}$; *(c) follows from* $\tau_1 = C_1\sqrt{Np}$; *(d) 'incoherence' property of singular vector, i.e.* $\nu_i^T e_j = O(1/\sqrt{p})$ *for all* $i, j \in [p]$; *(e) follows from property of geometric series for some absolute constant* $C > 0$; *and (f) follows from choice of* $k$.

**Concluding the proof.** *The final term is repeat of* $20\|\phi\|_\infty^2$. *Therefore, putting all of the above together, the proof concludes.*

## Footnotes

[1]In the control systems literature, such $f$ are known as Linear Time Invariant (LTI) systems.

[2]Note that an auxiliary benefit of our setup is that it allows for a significant fraction of the query response to be masked, in addition to to the Laplacian noise corruption.

[3]Column vector representation

[4]Recall that $\boldsymbol{W}_{i,\cdot}$ is a row vector and hence $\boldsymbol{W}_{i,\cdot} \boldsymbol{W}_{i,\cdot}^T$ is a scalar.