[Reviews · NeurIPS 2019]

Reviewer 1



I have read the authors' comments and believe all concerns are adequately addressed. -------------- Original Comments: Originality: The results are new and apply to a couple very well known methods. The previous work seems rigorously explored. Quality: I think the quality is high. I believe the analysis is technically sound, but I have only reviewed some of the proofs. I think the work is complete and I have made a couple comments about addressing weaknesses of the presentation. Clarity: The paper is very well written, but it definitely needs a conclusions section. Significance: High. These are new theoretical results for a couple problems that have been of interest for a long time. The authors present the first (to the authors’ knowledge and mine) training and generalization* error bounds for Principal Component Regression under noisy and missing data assumptions as well as model mismatch. The results rely on a (new to me) connection between PCR and Hard Singular Value Thresholding, for which a new error bound is derived for matrix completion via HSVT. From my reading of the paper, the results only seem to hold in the transductive semi-supervised setting (where HSVT and PCA steps are done with training AND testing covariates, but don’t utilize test response variables). Notes: Lines 186-195: I think I understand the argument, but I am not sure PCR is usually applied in practice by considering training and testing covariates together in the PCA step. It may be more fitting to describe your approach and results as being in a semi-supervised setting. Specifically, the approach you have described here fits into the transductive semi-supervised learning setting, where you want to infer the response variables for the specific unlabeled data you are using rather than a general rule that maps the covariates to the response. I don’t think this invalidates the work in this paper by any means (I think it is quite good), but I do think this point should be stated very clearly in the paper, since it is substantially different from the usual supervised learning setting we may expect with vanilla linear regression or PCR. This interpretation may also lead to some additional compelling applications for section 6. Proposition 3.1: I think there is a typo in your supplemental material E.1 where you write T instead of S. Proof of Thm 5.1: I think you are missing a term in beta hat corresponding to the model mismatch, though it won’t change the result since in (16) the independence assumption and a zero mean assumption on epsilon will zero that term out. The rest of the proof I have gone through reasonably closely and I believe it is correct. I have not had the opportunity to go through the proofs of other theorems closely, but the results seem believable given Theorem 5.1 Writing comments: In general the writing is superb. I did notice at least one grammatical error, so another proof reading is in order. Example: Line 193: “it is a natural to allow”

Reviewer 2



The paper gives the first analyses of the PCR under models that are commonly encountered in practice. It would be good that the relevant community know about this work. The proofs look serious though I did not check all of them. The paper is written well in general. The following are some minor points: Line 43: ‘cf.’ does not mean ‘see’ but ‘compare’. Hence it is better to say ‘see [36]’. There are more occurrences of misuse of ‘cf.’ in the paper Line 49: what does ‘faster’ mean? Please be clearer. There are more occurrences of unclear meaning of ‘faster’. Line 93: better to say ‘than the Frobenius norm’ or ‘the average of …’ Line 120: do not use citation as the subject of a sentence Line 156: covariates matrix -> covariate matrix Line 179: better to say explicitly that gamma depends on both alpha and p Line 216: there is no ‘orthonormal matrix’, only ‘orthogonal matrix’. This mistake also occurs on Line 362. Line 245: rth -> r-th Line 364: need a full stop after ‘<= C/(1-theta)’. Line 935: insert an \in between A and R^{N\times p} A style comment: Why is the text all italic in the proofs?

Reviewer 3



This paper presents a complete theoretical analysis of Principal Component Regression under noisy, missing, and mixed valued covariates settings. Equivalence between PCR and Linear Regression with covariate pre-processing via Hard Singular Value Thresholding is established. Several general error bounds are proposed under those settings. Overall, the paper is well organized, and the presentation is clear. Detail comments and suggestions are as following. - For theorem 5.1, are there any ways of estimating the degree of corrupted error or mismatch error? Is there any threshold that these two terms will dominate the bound? - How will regularization affect the proposed theorem, will it helps reducing impact of the last two terms in theorem 5.1? - How would proposed theorems extend to generalized linear models, such as logistic regression, Poisson regression? - Understand that the limitation of space, but it would be nice to have a simple experiment section even with synthetic dataset to help understand derived bound under each of these settings. I have read author's rebuttal and I will keep my original scores.

[Author Response · NeurIPS 2019]

# On Robustness of Principal Component Regression: Author Response

We begin by thanking all reviewers for their extremely encouraging and helpful responses. We intend to incorporate their feedback into our revision as best as possible. Below we respond to specific points raised by each reviewer.

**Typos (Reviewer 1 and 2):** We appreciate the thorough read throughs, and will fix all the typos and grammatical errors mentioned.

**Transductive Semi-Supervised Setting (Reviewer 1):** We agree that the fact we do PCR on both the training and testing covariates should be more explicitly placed in the context of transductive semi-supervised learning. In particular, we will detail the setting of transductive semi-supervised learning in Section 2.3 (Problem Setup).

**Conclusion Section (Reviewer 1):** We agree a conclusion section will be helpful for a reader to contextualize our results, but did not include it due to space constraints. However we will include one in our final draft and make the following points: (i) our work addresses a long-standing problem of demonstrating PCR is robust to noisy, sparse, and mixed valued covariates - in particular, we provide non-asymptotic bounds for both training and testing error for these settings; (ii) we establish a simple, but powerful equivalence between PCR and linear regression with covariate pre-processing via HSVT, and provide a novel error analysis of matrix estimation via HSVT with respect to the $\|\cdot\|_{2,\infty}$-norm; (iii) we formally connect our results with important applications to demonstrate the broad meaning of "noisy covariates": (a) synthetic control (measurement noise); (b) differentially-private regression (noise added by design); (c) mixed covariates ("structural" noise).

**Interpretation of Theoretical Error Bounds (Reviewer 2 and 3):** We have strived to interpret our major theorem results (Thm 4.2 & Thm 5.1) by: (i) providing examples of natural generating processes for $A$ (Section 4.2 and 5.2) and the error bounds associated with them; (ii) explaining the necessity of the terms in the error bounds (e.g. lines 273-277; lines 308-312; lines 330-335). However, as the reviewers suggest, providing information-theoretic lower bounds for the training/testing error is indeed interesting future work (for example, we believe training error scaling as $\sim \sigma^2 r/n$, as in Proposition 4.2, should be tight).

**Effect of Regularization (Reviewer 3):** We believe additional regularization in the regression step will not have significant impact. The reason is that HSVT covariate pre-processing, very pleasingly, already performs implicit $\ell_0$-regularization (see Proposition 4.1 and proof of Lemma K.4).

**Extension to Non-Linear Models (Reviewer 3):** We agree with the reviewer, that extending our results for PCR to non-linear models is an important direction to pursue. We believe a path forward to do so is by "linearizing" the relationship between the responses and covariates, i.e., embedding the covariates in a higher-dimensional space. For example, by using a polynomial basis, we have $f(x_i) = \sum_{j=1}^{p} \beta^j x_i^j + \phi_i$, where $\phi_i$ is the model mismatch error (see Section 5 of the paper).

**Experiments (Reviewer 3):** Due to space constraints, we did not include any experiments, but plan to do so in a longer version of our exposition. Please refer to the following empirical evaluations of PCR in the context of time series analysis and synthetic control: Figures (3, 7, 10) of [1] and (2, 5) of [2], and (2-6) of [3]. Their empirical results support our theoretical guarantees.

[1] M. Amjad, D. Shah, and D. Shen. "Robust synthetic control". *Journal of Machine Learning Research*, 19:1–51, 2018.

[2] M. Amjad, V. Mishra, D. Shah, and D. Shen. "mRSC: Multi-dimensional Robust Synthetic Control". *Proceedings of the ACM on Measurement and Analysis of Computing Systems*, 2019.

[3] A. Agarwal, M. Amjad, D. Shah, and D. Shen. "Model Agnostic Time Series Analysis via Matrix Estimation". *Proceedings of the ACM on Measurement and Analysis of Computing Systems*, 2019.



[Meta-Review · NeurIPS 2019]

Reviewers are all in agreement that this is an obvious accept.